ARTICLES
# Brain-enriched RagB isoforms regulate the dynamics of mTORC1 activity through GATOR1 inhibition

Gianluca Figlia[1,2], Sandra Müller[1,2], Anna M. Hagenston[3], Susanne Kleber[4], Mykola Roiuk[1,2], Jan-Philipp Quast[1,2], Nora ten Bosch[1], Damian Carvajal Ibañez[2,4], Daniela Mauceri[3,5], Ana Martin-Villalba[4,5] and Aurelio A. Teleman [1,2] ✉

**Mechanistic target of rapamycin complex 1 (mTORC1) senses nutrient availability to appropriately regulate cellular anabolism and catabolism. During nutrient restriction, different organs in an animal do not respond equally, with vital organs being relatively spared. This raises the possibility that mTORC1 is differentially regulated in different cell types, yet little is known about this mechanistically. The Rag GTPases, RagA or RagB bound to RagC or RagD, tether mTORC1 in a nutrient-dependent manner to lysosomes where mTORC1 becomes activated. Although the RagA and B paralogues were assumed to be functionally equivalent, we find here that the RagB isoforms, which are highly expressed in neurons, impart mTORC1 with resistance to nutrient starvation by inhibiting the RagA/B GTPase-activating protein GATOR1. We further show that high expression of RagB isoforms is observed in some tumours, revealing an alternative strategy by which cancer cells can retain elevated mTORC1 upon low nutrient availability.**

Mechanistic target of rapamycin complex 1 (mTORC1) senses and integrates inputs from nutrients and growth factors to promote cell growth by stimulating cellular anabolic processes while inhibiting catabolic ones[1,2]. As a consequence, mTORC1 ensures that cell growth and metabolism are in line with the availability of biomolecular building blocks and the overall growth status of the organism. In metazoans, mTORC1 has different functions in different tissues. mTORC1 regulates synaptic transmission and myelination in the nervous system[3,4], trophism in the skeletal muscle[5] and triglyceride synthesis and adipocyte differentiation in the adipose tissue[6], to name a few. Such diversity of processes regulated by mTORC1 raises the question of whether the mTORC1 pathway is identically regulated in all tissues, or whether it can be tuned to the specific needs of a tissue or cell type. Consistent with the latter possibility, fluctuations of nutrient levels have different growth outcomes in different tissues. When nutrients become limiting to an organism, most tissues and organs enter a low-metabolic or catabolic state, while a few tissues and organs that provide vital functions to the organism, such as the brain or the heart, are notably spared. As an extreme example, severe nutrient restriction during intra-uterine growth causes the central nervous system to grow at the expense of other organs and results in small newborns with large heads, a process known as brain sparing[7]. Likewise, the brain is spared during acute hypoxia by re-routing cardiac output away from peripheral vascular beds[8]. As mTORC1 plays a central role in regulating cellular growth and metabolism, one could imagine that its activity is differentially modulated in response to nutrient limitation in different tissues and cell types. To our knowledge, such molecular mechanisms are not known.

A sophisticated molecular machinery has evolved that allows mTORC1 to sense nutrients. At the heart of the nutrient sensing capacity is a group of GTPases belonging to the Ras superfamily named the Rag GTPases in metazoa, and Gtr proteins in yeast[9–12]. Rag GTPases have two peculiar features that make them exquisitely dynamic, as expected for a system that has to respond rapidly to changing nutrient levels. First, unlike most other GTPases, the Rag GTPases do not associate with membranes directly through a lipid modification, but rather interact with a protein complex ('Ragulator') that is located on the lysosomal surface, the signalling hub of the mTORC1 pathway[13]. This allows the Rag GTPases to shuttle on and off the lysosome according to their nucleotide loading state[14], which could regulate the amplitude of the response to nutrients. Second, unlike most other Ras-like GTPases, they function as obligate heterodimers, consisting in mammals of a RagA or RagB protein heterodimerizing with either RagC or RagD through complementary C-terminal roadblock domains (CRD). The two subunits of the heterodimer affect each other's GTP binding and hydrolysis rate[15]. Such inter-subunit crosstalk causes the dimer to quickly lock in one of two opposite configurations (GTP–GDP or GDP–GTP) according to nutrient levels, thus conferring rapidity and robustness to nutrient sensing.

As is often the case for small GTPases, the intrinsic GTP hydrolysis of the Rag GTPases is slow ($k_{cat} = 2.2 \times 10^{-4} – 3.0 \times 10^{-3}\,min^{-1}$) (ref. [15]). To allow swift responses as nutrient levels change, GTPase-activating proteins (GAPs) interact with the Rag GTPases to accelerate their hydrolysis rate, in an amino-acid-regulated fashion. The GATOR1 complex, composed of DEPDC5, Nprl2 and Nprl3, is the GAP for RagA/B[16]. Its catalytic mechanism entails stabilization of the reaction intermediate through insertion of an

[1]Signal Transduction in Cancer and Metabolism, German Cancer Research Center (DKFZ), Heidelberg, Germany. [2]Heidelberg University, Heidelberg, Germany. [3]Department of Neurobiology, Interdisciplinary Center for Neurosciences (IZN), Heidelberg University, INF 366, Heidelberg, Germany. [4]Molecular Neurobiology, German Cancer Research Center (DKFZ), Heidelberg, Germany. [5]These authors contributed equally: Daniela Mauceri, Ana Martin-Villalba. ✉e-mail: a.teleman@dkfz.de

arginine finger provided by Nprl2 in a low-affinity interaction with RagA/B[17,18]. In addition, a high-affinity interaction between RagA/B and the DEPDC5 subunit with no direct GAP function has been reported and proposed to somehow diminish the overall GAP activity of GATOR1 (ref. [17]), although it is unclear under which circumstances this 'inhibitory binding mode' is biologically relevant. When amino acids are low, GATOR1 associates with RagA/B, causing them to switch to the GDP-bound state. In contrast, when amino acids are abundant, amino-acid sensors inhibit GATOR1 by acting on the upstream GATOR2 complex[19,20] or on GATOR1 itself[21], causing RagA/B to stay bound to GTP. The GAPs for RagC/D are the folliculin complex, composed of FLCN and FNIP1/2 (ref. [22]), and LARS1, which is specific to RagD[23]. Interestingly, despite being a RagC/D GAP, FLCN–FNIP1/2 interacts extensively with RagA/B and is affected by the nucleotide loading state of RagA/B, thus placing GATOR1 in a hierarchically higher position in relaying nutrient signals to the Rag GTPases[24–26]. Upon GTP binding, or conversely when GTP is hydrolysed, three regions named switch I, interswitch and switch II undergo major structural re-arrangements[27,28], causing the interactions with some Rag effectors to be lost and interactions with other effectors to be established. In particular, when RagA/B are bound to GTP and RagC/D to GDP, the dimer adopts a conformation competent to bind the mTORC1 subunit Raptor[27,28], thus recruiting the whole complex to the lysosomal surface and bringing it in close proximity to the mTORC1 activator Rheb.

Mammalian Rag GTPases come in a range of different isoforms and paralogous genes. In contrast to yeast or fruit fly, where only one RagA/B gene and one RagC/D gene exist, the mammalian Rag GTPases include two pairs of highly similar paralogues, RagA/B and RagC/D. Moreover, in addition to the main RagB isoform, which we name here RagB[short], a brain-specific alternative splicing isoform of RagB, RagB[long], with unknown function has been described[11], yielding a total of potentially six distinct Rag dimers. Why such a high degree of redundancy has evolved is unclear, as is its functional relevance. Until now, it has been assumed that RagA and RagB are functionally equivalent, as are RagC and RagD[9,10]. Interestingly, deletion of RagA causes embryonic lethality in mice, while mice devoid of the RagB isoforms are viable[29], indicating that they can be compensated for by RagA, or that they have more refined, tissue-specific functions.

In this Article, we show that the RagB isoforms RagB[short] and RagB[long], which are highly expressed in neurons, change the dynamics of mTORC1 activity, causing it to persist despite low amino acid levels. This effect relies on two distinct mechanisms of GATOR1 inhibition. RagB[short] inhibits GATOR1 by binding it in the 'inhibitory mode' via DEPDC5, while RagB[long] acts as a Rag isoform with low affinity for GTP and high affinity for the GATOR1 subunits Nprl2/3, thus titrating away the GAP activity of GATOR1. We further show that aberrantly high expression of the RagB isoforms is observed in a subset of tumour samples, providing an alternative strategy whereby mTORC1 activity in cancer cells can acquire resistance to low nutrient levels.

## Results

**The Rag isoforms are differentially expressed in tissues.** The Rag GTPases are central components of the molecular machinery regulating mTORC1 in response to nutrients. Unclear is whether RagA is functionally equivalent to RagB, and RagC to RagD, and whether different combinations of Rag proteins cause different responses of mTORC1 to nutrients. Transcriptomic data show that the relative expression of Rag GTPases varies in different human tissues, suggesting that different repertoires of Rag GTPases may be present in different cell types (Fig. 1a). Similarly, we also detected differential levels of RagA, RagB and RagC proteins in mouse tissues (Fig. 1b) (no working antibody for mouse RagD is commercially available). In particular, the main RagB isoform (hereafter named RagB[short]) is expressed at low levels in most tissues, but at higher levels in the brain, where additionally a longer splice isoform with unknown function is expressed (hereafter named RagB[long]). RagC levels are lower in skeletal muscle and heart, where RagD is highest according to transcriptomic data (Fig. 1a), indicating that RagD could be the predominant isoform in these tissues.

Although RagA/B and RagC/D proteins are almost identical in the GTPase and CRD domains, they diverge substantially in the most N- and/or C-terminal regions (Fig. 1c,d). RagB[short] and RagB[long] have a 33-amino acid N-terminal extension that is absent in RagA. Additionally, RagB[long] contains a stretch of 28 amino acids encoded by exon 4 that is inserted in the switch I region, which changes conformation upon GTP binding and is responsible for effector binding. Analogously, two poorly structured N- and C-terminal extensions present in RagC and RagD exhibit only 25% and 39% similarity, respectively, between the two paralogues. Together, these differences and the non-homogeneous tissue distribution of the four Rag GTPases raise the possibility that they might have specific functions in certain cell types and/or conditions.

**RagB isoforms are more resistant to amino-acid removal than RagA.** To study if the Rag isoforms differ functionally, we used HEK293T cells, which are often employed to study mTOR signalling. We generated RagA and RagB double-knockout HEK293T cells (RagABKO) (Extended Data Fig. 1a–c), which we then reconstituted with either RagA, RagB[short] or RagB[long] to yield cells containing

**Fig. 1 | RagA/B paralogues determine distinct mTORC1 responses. a**, Transcript levels of the Rag isoforms in healthy human tissues (gtexportal.org). *n*, biological replicates. **b**, Western blot for RagA, RagB and RagC in mouse tissues. Calnexin is the loading control. The experiment was repeated once. **c,d**, Domain organization of the Rag isoforms. Numbering indicates amino-acid positions in the human sequence. Percentages represent similarity of each domain between Rag paralogues. Ex4 is the sequence encoded by exon 4 of the *Rragb* gene. **e–h**, S6K1, TFEB and 4EBP1 phosphorylation in control or RagABKO cells stably transfected with a control protein (FLAG–metap2) or with the indicated Rag isoforms. Cells were incubated in amino-acid-rich medium or starved of amino acids for 30 min: representative example (**e**) and quantification of three independent experiments, with unstarved control cells set to 1 (**f–h**). Bar height indicates average, and error bars represent standard deviation; *n* = 3 biological replicates. Two-way ANOVA and Sidak's post-hoc test. **i,j**, S6K1 phosphorylation upon loss of RagA, RagB or both: representative example (**i**) and quantification of three independent experiments, with control cells set to 1 (**j**). Bar height indicates average, and error bars represent standard deviation; *n* = 3 biological replicates. One-way ANOVA and Tukey's post-hoc test. **k–n**, RagA (**k** and **l**) but not RagB (**m** and **n**) loss causes persistent mTORC1 activity. Cells were incubated in amino-acid-rich medium, starved of amino acids for 1 h, or starved for 1 h and re-stimulated with amino acids for 15 min (addback, 'ab'): representative examples (**k** and **m**) and quantification of three independent experiments, with unstarved control cells set to 1 (**l** and **n**). Bar height indicates average, and error bars represent standard deviation; *n* = 3 biological replicates. Two-way ANOVA and Sidak's post-hoc test. **o–p**, The elevated mTORC1 activity in RagAKO cells upon amino-acid removal cannot be rescued by stable overexpression of the RagB isoforms. Cells were incubated in amino-acid-rich medium or starved of amino acids for 30 min: representative example (**o**) and quantification of three independent experiments, with unstarved control cells set to 1 (**p**). Bar height indicates average, error bars represent standard deviation; *n* = 3 biological replicates. Two-way ANOVA and Sidak's post-hoc test. −aa, amino-acid-free DMEM + 10% dFBS. +aa, −aa medium supplemented with 1× amino acids. Exact *P* values are shown in the graphs. Source numerical data and unprocessed blots are available in source data.

only one Rag paralogue (Fig. 1e–h). Likewise, we generated RagC/D double-knockout cells (RagCDKO) (Extended Data Fig. 1d–f) and reconstituted them with either RagC or RagD (Extended Data Fig. 1h–k). This approach yields cells expressing comparable levels of

the different Rag paralogues, thereby revealing effects caused by differences in function rather than expression. Indeed, HEK293T cells endogenously express more RagA than RagB mRNA (Extended Data Fig. 1g) and protein (Extended Data Fig. 1a). Deletion of

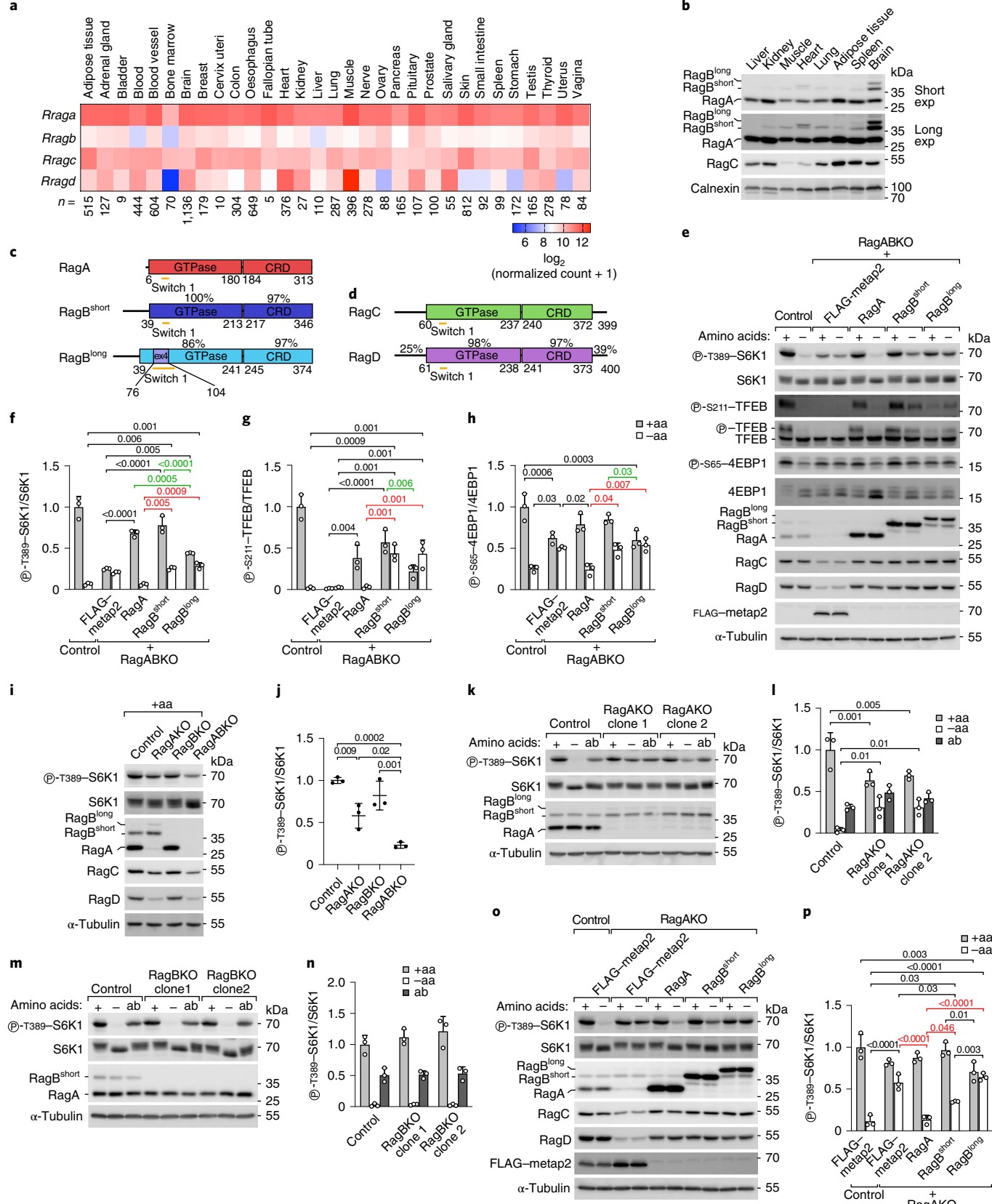

RagA/B caused a decrease in RagC/D protein levels and vice versa (Extended Data Fig. 1a,d), which was rescued upon reconstitution with single Rag paralogues (Fig. 1e and Extended Data Fig. 1h), suggesting Rag monomers are unstable.

We then assessed mTORC1 activity via phosphorylation of S6K1, TFEB and 4EBP1, direct mTORC1 substrates that respond rapidly to changes in nutrient levels[30,31] (Extended Data Fig. 2a,b). Double knockout of RagA/B or RagC/D caused a decrease, but not a complete loss, of mTORC1 activity and rendered the residual mTORC1 activity largely unresponsive to amino-acid withdrawal or re-addition (Extended Data Fig. 1a–f). Consistent with Rag depletion, immunofluorescence experiments showed loss of mTOR accumulation on lysosomes in RagABKO and RagCDKO cells under nutrient-replete conditions (Extended Data Fig. 3a–d). Low levels of lysosomal mTOR and Raptor, however, could still be detected by immunoblotting-purified lysosomes (lyso-IP) from RagA/B or RagC/D double knockouts (Extended Data Fig. 3e–g), probably contributing to the residual S6K phosphorylation in double knockouts. These results are consistent with previous work showing that, in the absence of all Rag isoforms, mTOR can still be recruited to lysosomes in an Arf-1-dependent manner[32] and that upon amino-acid starvation the inactive Rag GTPases not only release mTORC1 from the lysosome, but also actively recruit factors that inactivate mTORC1, such as the TSC complex, causing persistent mTORC1 activity when all Rag isoforms are missing[33]. We also noticed that RagC/D did not localize to lysosomes in the absence of RagA/B and vice versa (Extended Data Fig. 3e,h–k), suggesting that assembly of functional Rag dimers is necessary not only for their stabilization, but also for their delivery to lysosomes.

Interestingly, re-expression of the three RagA/B isoforms in RagABKO cells yielded distinct patterns of mTORC1 responses. In nutrient-replete conditions, reconstitution with RagA or RagB[short] increased S6K1, 4EBP1 and TFEB phosphorylation to comparable levels, while these were lower in RagB[long]-expressing cells (Fig. 1e–h). Furthermore, while phosphorylation of the three substrates dropped strongly upon amino-acid starvation in RagA-expressing cells, it dropped less strongly in RagB[short]- and RagB[long]-expressing cells, indicating persistent mTORC1 activity. This suggests that RagB[short] and RagB[long] keep mTORC1 more active despite amino-acid removal. All three RagA/B isoforms interact similarly with the RagC/D isoforms in co-immunoprecipitation (co-IP) experiments (Extended Data Fig. 3l–n), indicating that the phenotypic differences between RagA and RagB are not due to differences in RagC/D binding. Reconstitution of RagCDKO cells with RagC versus RagD showed differences in the phosphorylation of TFEB but not S6K1 or 4EBP1 in response to nutrients (Extended Data Fig. 1h–k), suggesting that these two isoforms might have differential effects on a subset of mTORC1 substrates. In this manuscript we focus on the functional differences between RagA and RagB, while the accompanying manuscript by Demetriades and colleagues focuses on RagC versus RagD[34].

The effects of RagA, RagB[short] and RagB[long] on mTOR localization correlated with their effect on mTORC1 activity. Stable transfection of RagABKO cells with RagA rescued mTOR localization to a large extent to wild-type behaviour, with predominantly lysosomal accumulation in nutrient-rich conditions and cytosolic localization upon amino-acid removal (Extended Data Fig. 4a,b). Cells reconstituted with RagB[short] were able to recruit mTOR to lysosomes in nutrient-replete conditions, but still retained significant amounts of mTOR on lysosomes upon amino-acid removal, consistent with the persistent mTORC1 activity observed by western blot (Fig. 1e–h). In contrast, RagB[long] failed to increase the lysosomal localization of mTOR in either immunofluorescence or lyso-IP experiments (Extended Data Fig. 4a–d), suggesting that RagB[long] interacts poorly with mTORC1. Indeed, RagB[long] co-immunoprecipitated substantially less Raptor than RagA or RagB[short], although a weak interaction could still be detected (Extended Data Fig. 4e,f).

We observed the same phenotypic differences between RagA and RagB in a complementary experimental set-up, where we knocked out either RagA or RagB, leaving the cells to express only the other endogenous paralogue. Worth noting, deletion of RagA led to a mild compensatory increase in RagB[short] levels together with the appearance of RagB[long], which is undetectable in control cells (Fig. 1i). RagAKO cells had lower basal mTORC1 activity than control cells, but higher than RagA/B double-knockout cells (Fig. 1i,j), indicating that RagB can partially compensate for loss of RagA. Consistent with the possibility that RagB is more resistant to nutrient removal than RagA, amino-acid removal caused only a mild drop in S6K1 phosphorylation in RagAKO cells, but a complete loss in control and RagBKO cells (Fig. 1k–n). Likewise, mTOR accumulation on lysosomes decreased only mildly in RagAKO cells during amino-acid starvation, in contrast to the complete re-localization to the cytoplasm in control cells (Extended Data Fig. 4g,h). Importantly, the persistent mTORC1 activity in RagAKO cells could be reverted fully by re-expressing RagA, only partially by RagB[short] and not at all by RagB[long] (Fig. 1o,p), confirming that the persistent mTORC1 activity stems from qualitative and not quantitative differences between RagA and the RagB isoforms.

In sum, these results show that the RagA/B isoforms are not functionally redundant: (1) RagB[short] and RagB[long] are more resistant to nutrient withdrawal compared with RagA, and (2) RagB[long] does not bind and recruit mTOR to lysosome as efficiently as the other isoforms.

**RagB[short] and RagB[long] are resistant to GATOR1.** Amino-acid removal activates GATOR1, which acts as a GAP for RagA/B to promote GTP hydrolysis and subsequent release of mTORC1 from the lysosome[16]. One possible explanation why mTORC1 activity

---

**Fig. 2 | RagB[short] and RagB[long] are resistant to GATOR1. a,b**, RagABKO cells expressing each RagA/B isoform were transiently transfected with increasing amounts of GATOR1 plasmids (5 ng, 25 ng or 100 ng of each GATOR1 subunit) or metap2 (100 ng) as negative control: representative example (**a**) and quantification of four independent experiments, with RagA-expressing cells transfected with metap2 set to 1 (**b**). Bar height indicates average, and error bars represent standard deviation; $n = 4$ biological replicates. Two-way ANOVA and Tukey's post-hoc test. **c,d**, Control and RagAKO cells were transiently transfected with high (200 ng) levels of each GATOR1 subunit or metap2 as negative control and treated with amino-acid-rich medium or starved of amino acids for 30 min before lysis: representative example (**c**) and quantification of three independent experiments, with unstarved metap2-transfected control cells set to 1 (**d**). Bar height indicates average, and error bars represent standard deviation; $n = 3$ biological replicates. Two-way ANOVA and Sidak's post-hoc test. **e**, Schematic representation of the two binding interfaces of GATOR1 to RagA/B. **f–h**, Rag interaction with the inhibitory interface (depicted in **f**) is assessed by co-immunoprecipitating the whole GATOR1 complex: representative example (**g**) and quantification of three independent experiments, with the inactive mutant of RagA set to 1 (**h**). Bar height indicates average, and error bars represent standard deviation; $n = 3$ biological replicates. One-way ANOVA and Tukey's post-hoc test. **i–k**, Rag interaction with the GAP interface (depicted in **i**) is assessed through co-IP with the Nprl2/3 dimer in DEPDC5KO cells: representative example (**j**) and quantification of three independent experiments, with the inactive mutant of RagA set to 1 (**k**). Bar height indicates average, and error bars represent standard deviation; $n = 3$ biological replicates. One-way ANOVA and Tukey's post-hoc test. −aa, amino-acid-free DMEM + 10% dFBS. +aa, −aa medium supplemented with 1× amino acids. Exact $P$ values are shown in the graphs. NS, not significant. Source numerical data and unprocessed blots are available in source data.

remains high in RagB-expressing cells upon amino-acid removal is that GATOR1 does not stimulate the RagB isoforms to hydrolyse GTP as efficiently as it does RagA. Alternatively, the RagB isoforms

efficiently hydrolyse GTP to GDP but their non-GTP-bound conformations still bind mTORC1. To test the latter option, we first performed a co-IP experiment between the three RagA/B isoforms

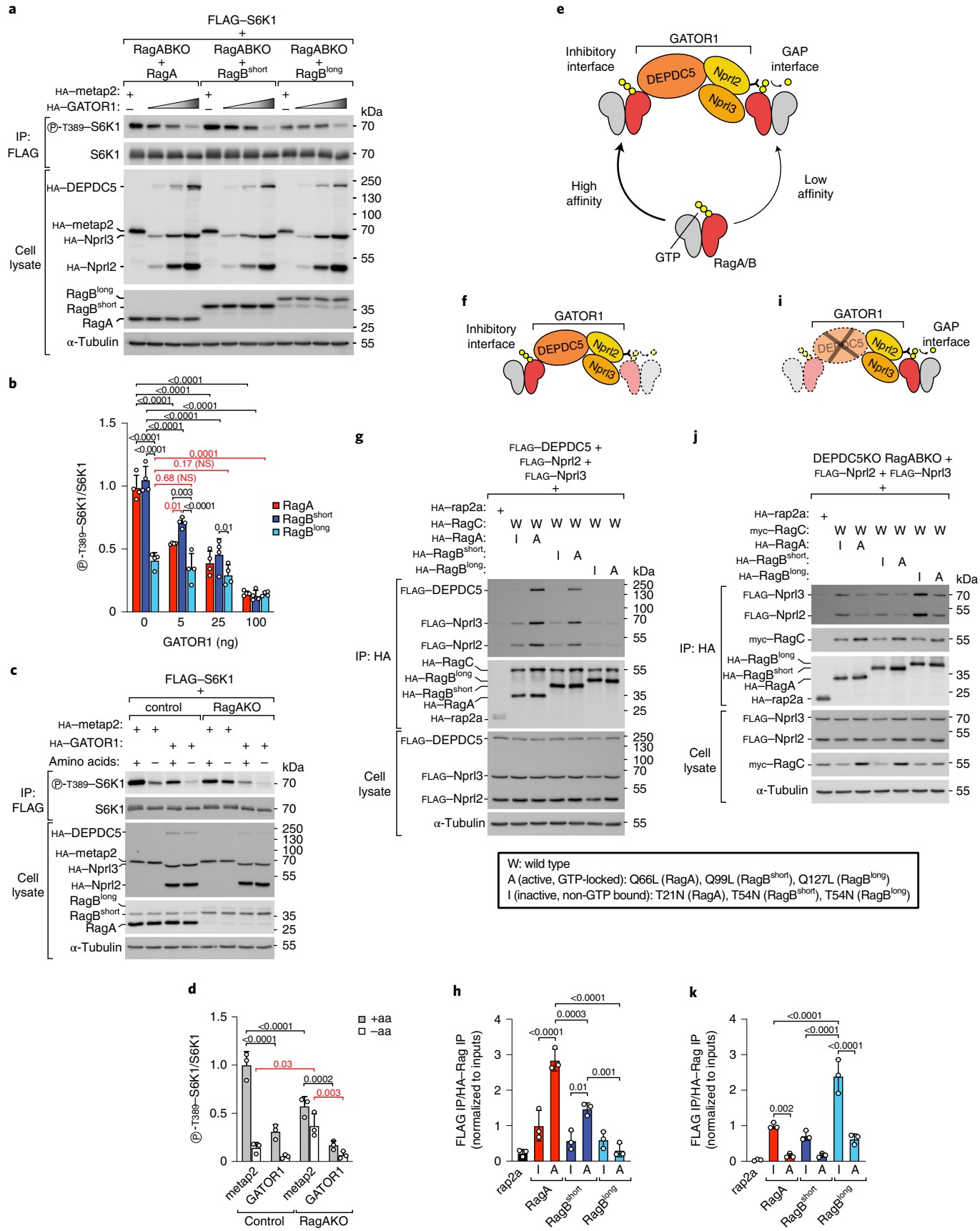

in different nucleotide loading states and the mTORC1 subunit Raptor. GTP-locked RagA and RagB[short] interacted comparably with Raptor, while GTP-locked RagB[long] bound more weakly (Extended Data Fig. 5a,b), consistent with the reduced ability of wild-type RagB[long] to bind mTORC1 (Extended Data Fig. 4a,b,e,f). In contrast, none of the three RagA/B isoforms in the non-GTP-bound state interacted with Raptor (Extended Data Fig. 5a,b), indicating that inactive RagB[short] and RagB[long] should not be retaining mTORC1 on the lysosome. Consistent with these results, strong overexpression of GDP-locked RagA, RagB[short] or RagB[long] rescued mTORC1 activity, enabling it to be low in RagAKO cells upon amino-acid removal (Extended Data Fig. 5c–f). Together, these data indicate that, if the RagB isoforms hydrolyse GTP to GDP, they release mTORC1 and allow it to turn off.

We therefore tested the alternate explanation, that the RagB isoforms are more resistant to GATOR1 than RagA. We first tested this in nutrient-replete conditions by transfecting cells expressing the single RagA or B isoforms with increasing amounts of GATOR1 (Fig. 2a,b). Although high levels of GATOR1 inhibited mTORC1 in all cells, lower levels of GATOR1 overexpression induced a stronger reduction of mTORC1 activity in RagA- than in RagB[short]-expressing cells, while high levels of GATOR1 were required to cause an appreciable drop in S6K1 phosphorylation in RagB[long]-expressing cells (Fig. 2a,b, Extended Data Fig. 6a,b). Together, these data indicate that the RagB isoforms are comparatively resistant to GATOR1.

To confirm that relative resistance to GATOR1 is the cause of high mTORC1 activity in RagB-expressing cells upon amino-acid removal, we performed two epistasis experiments. First, we verified that overexpression of GATOR1 rescues this phenotype. Indeed, GATOR1 overexpression rescued the persistent mTORC1 activity observed in RagAKO cells upon amino-acid deprivation, causing it to decrease to the same level as control cells (Fig. 2c,d). Second, the phenotypic difference between RagA and RagB should be gone in cells lacking GATOR1. To this end, we knocked out the GATOR1 subunit DEPDC5 in RagABKO cells and stably transfected them with RagA, RagB[short], RagB[long] or metap2 as a control. Indeed, cells expressing RagA or RagB[short] in a DEPDC5KO background have high mTORC1 activity upon amino-acid starvation, as expected, but without noticeable differences between these two Rag isoforms (Extended Data Fig. 6c–e), indicating that differential resistance to GATOR1 is the main functional difference between RagA and RagB[short]. Consistent with the low binding of Raptor to RagB[long], mTORC1 activity in RagB[long]-expressing cells remained substantially lower in all nutrient conditions (Extended Data Fig. 6c–e).

In sum, these results suggest that the three RagA/B isoforms have different resistance to GATOR1 in the order RagB[long] > RagB[short] > RagA and that this different resistance to GATOR1 is probably the main functional distinction between RagB[short] and RagA.

**RagB[short] inhibits GATOR1 through binding via DEPDC5.** The relative resistance of the RagB isoforms to GATOR1 suggests that either they are poor substrates of the GATOR1 complex or they actively inhibit GATOR1, or both. The interaction between GATOR1 and the Rag GTPases consists of two binding interfaces (Fig. 2e)[17]. At the GAP interface, the Nprl2/3 subunits of GATOR1 bind with low affinity to RagA/B to provide the arginine finger (R78 of Nprl2) necessary for GTP hydrolysis. Unlike other known GAPs and their target GTPases, GATOR1 and the Rag GTPases also have an additional, high-affinity interaction between the DEPDC5 subunit of GATOR1 and switch I of RagA/B, which does not execute any GAP activity. As expression of a DEPDC5 mutant that does not bind to the Rag GTPases results in stronger mTORC1 suppression than its wild-type counterpart[17], this binding mode is thought to inhibit the GAP activity of GATOR1 and has therefore been named the inhibitory interface. Recent structural studies suggest that binding of GATOR1 to Rags via the inhibitory interface holds GATOR1 in an orientation relative to the lysosomal surface that is unfavourable for acting as a GAP on adjacent Rag molecules[35].

We first tested how the three isoforms interact with GATOR1 at the two binding interfaces. As Rag binding to the GAP interface is approximately 40-fold weaker than binding to the inhibitory interface[17,18], co-IP of the Rag GTPases with the entire GATOR1 complex reflects mainly binding to the inhibitory interface (Fig. 2f–h). In parallel, we co-immunoprecipitated the Rag GTPases with Nprl2/3 from DEPDC5KO cells to exclude binding via the inhibitory interface, thereby specifically assessing the GAP interface (Fig. 2i–k). Interestingly, each RagA/B isoform exhibited a distinct profile of GATOR1 interaction. RagB[short] interacted less than RagA with the inhibitory interface, but interacted similar to RagA with the GAP interface. RagB[long] interacted even less with the GATOR1 inhibitory interface as compared with RagA and RagB[short], but more strongly than the other two isoforms with the GAP interface, when non-GTP bound.

We focused first on the differences between RagA and RagB[short]. To compare how strongly RagA versus RagB[short] inhibit GATOR1 via the inhibitory interface, we compared cells expressing wild-type DEPDC5 with cells expressing a DEPDC5 mutant (Y775A) that does not bind the Rag GTPases on the inhibitory interface[17]. We did this by reconstituting RagA/B–DEPDC5 triple-knockout cells with single Rag isoforms and either wild-type or mutant DEPDC5. As expected, expression of wild-type DEPDC5 in RagA- or RagB[short]-expressing cells caused a reduction in mTORC1 activity, because this reconstitutes the GATOR1 complex (lanes 1–4 in Fig. 3a–d). In RagB[short]-expressing cells, DEPDC5[Y775A] expression led to an even stronger inhibition of mTORC1 compared with wild-type DEPDC5, because the DEPDC5[Y775A] mutant cannot bind RagB[short] protein and thus cannot be inhibited by it[17] (lane 2 versus lane 5, Fig. 3c,d). Thus, the difference between lane 2 and lane 5 of Fig. 3c,d reflects the inhibitory activity of RagB[short] on GATOR1 through DEPDC5 binding. In RagA-expressing cells, however, DEPDC5[Y775A] expression caused the same degree of mTORC1 inhibition as wild-type DEPDC5 (Fig. 3a,b). Hence, although RagA binds DEPDC5, it does not cause GATOR1 inhibition as much as when RagB[short] binds DEPDC5.

**Fig. 3 | RagB[short] inhibits GATOR1 through interaction with DEPDC5. a–d,** Transient expression of increasing amounts (5 ng, 25 ng or 100 ng DNA) of a non-Rag binding mutant of DEPDC5 (Y775A) suppresses mTORC1 more strongly than wild-type DEPDC5 in RagB[short]-expressing cells (**c** and **d**) but not in RagA-expressing cells (**a** and **b**). Cells were subjected to amino-acid starvation (amino-acid-free DMEM + 10% dFBS) for 30 min to activate GATOR1: representative examples (**a** and **c**) and quantifications of three independent experiments, with metap2-transfected cells set to 1 (**b** and **d**). Circle indicates average, and error bars represent standard deviation; $n = 3$ biological replicates. Two-way ANOVA and Sidak's post-hoc test. **e–h,** Transient expression of increasing amounts (5 ng, 25 ng or 100 ng DNA) of wild-type DEPDC5 (**e** and **f**) inhibits mTORC1 activity more strongly in RagA-expressing cells than in RagB[short]-expressing cells, whereas expression of a non-Rag binding mutant of DEPDC5 (Y775A) (**g** and **h**) inhibits mTORC1 activity equally well in the presence of RagA or RagB[short]. Cells were subjected to amino-acid starvation (amino-acid-free DMEM + 10% dFBS) for 30 min to activate GATOR1: representative examples (**e** and **g**) and quantification of three independent experiments, with RagA-expressing cells transfected with metap2 set to 1 (**f** and **h**). Circle indicates average, and error bars represent standard deviation; $n = 3$ biological replicates. Two-way ANOVA and Sidak's post-hoc test. **i,** Schematic representation of GATOR1 binding to RagA or RagB[short] via DEPDC5. DEPDC5 binding to RagB[short] but not to RagA inhibits GATOR1 activity. Exact $P$ values are shown in the graphs. NS, not significant. Source numerical data and unprocessed blots are available in source data.

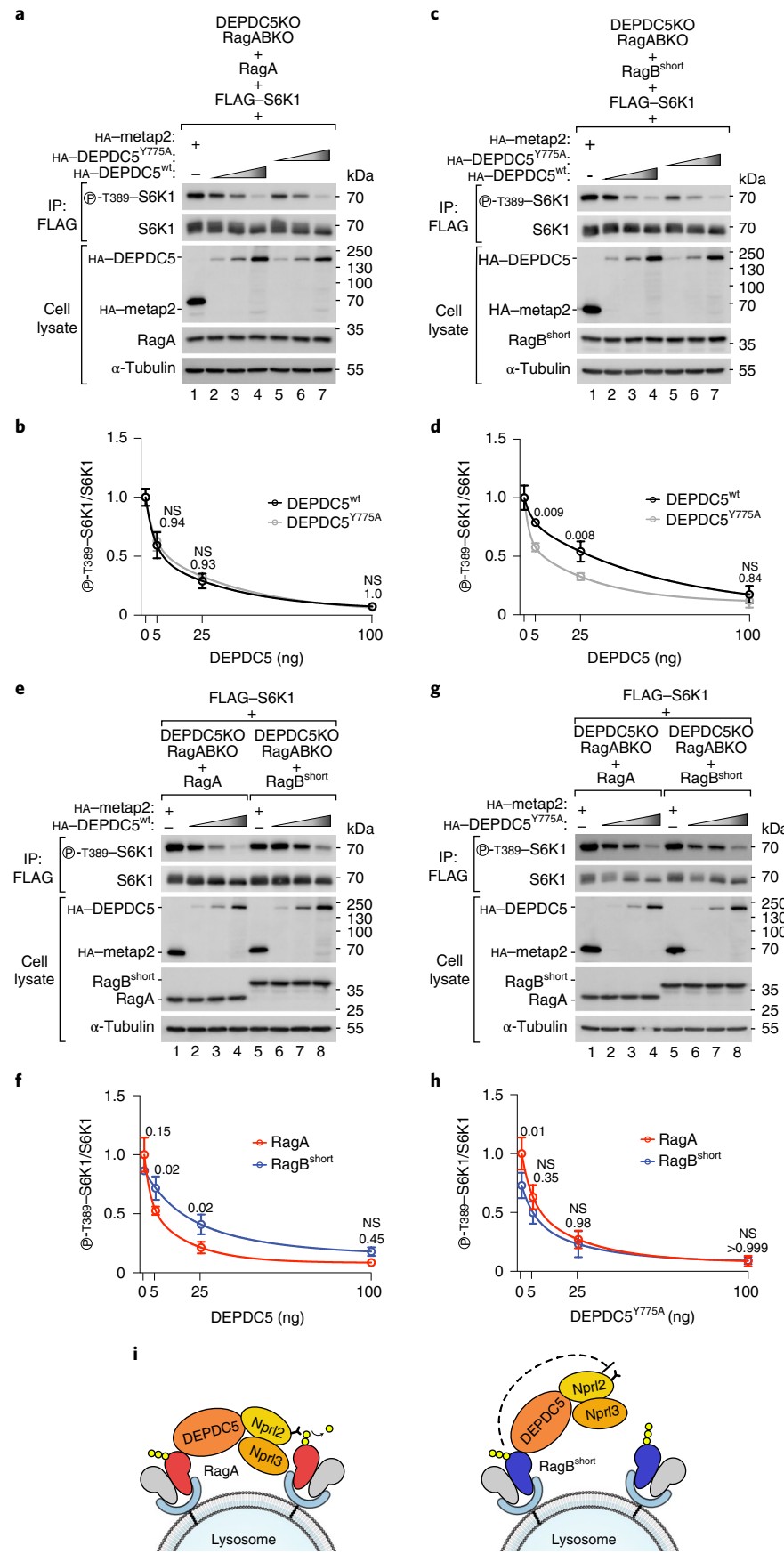

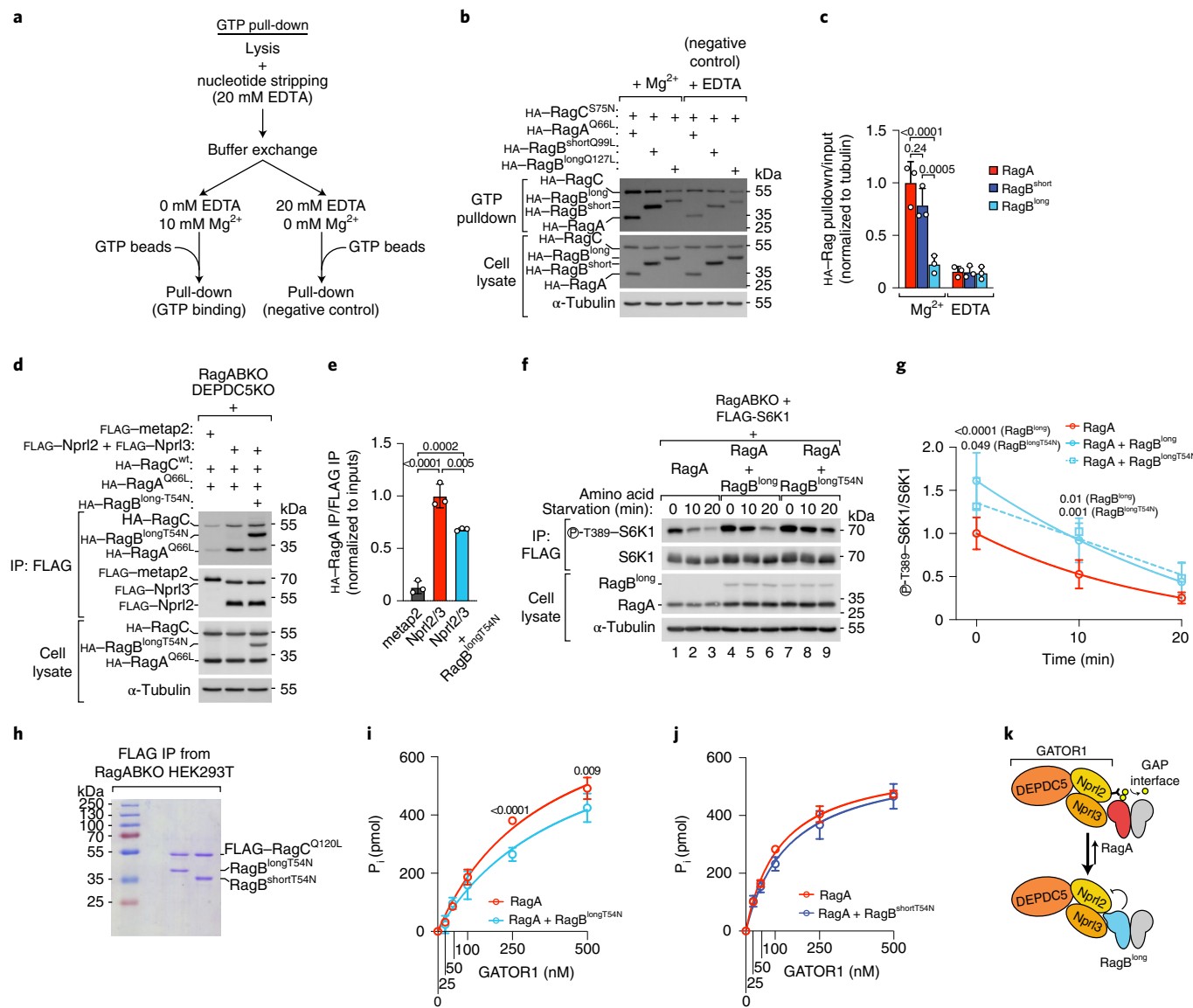

**Fig. 4 | RagB$^{long}$ acts as a 'sponge' for the GAP interface of GATOR1. a**, Schematic representation of the GTP pull-down assay. **b,c**, GTP pull-down of RagA, RagB$^{short}$ or RagB$^{long}$ in the presence of magnesium or with addition of 20 mM EDTA as negative control: representative example (**b**) and quantification of three independent experiments, with RagA Mg$^{2+}$ set to 1 (**c**). Bar height indicates average, and error bars represent standard deviation; $n = 3$ biological replicates. Two-way ANOVA and Sidak's post-hoc test. **d,e**, Co-IP of Nprl2 and Nprl3 with Rag dimers consisting of wild-type RagC and GTP-locked RagA (Q66L) expressed with or without non-GTP-bound (T54N) RagB$^{long}$. The experiment was performed using RagA/B and DEPDC5 triple-knockout cells to assess specifically the binding to the GAP interface of GATOR1: representative example (**d**) and quantification of three independent experiments, with the Nprl2/3 condition set to 1 (**e**). Bar height indicates average, and error bars represent standard deviation; $n = 3$ biological replicates. One-way ANOVA and Tukey's post-hoc test. **f,g**, S6K1 phosphorylation in RagABKO cells transiently transfected with RagA either alone or together with wild-type RagB$^{long}$ or a non-GTP-bound mutant of RagB$^{long}$ (T54N) during amino-acid starvation (amino-acid-free DMEM + 10% dFBS): representative example (**f**) and quantification of six independent experiments, with RagA-expressing cells at timepoint 0 set to 1 (**g**). Circle/square indicates average, and error bars represent standard deviation; $n = 6$ biological replicates. Two-way ANOVA and Tukey's post-hoc test. **h**, Coomassie staining of RagB$^{longT54N}$ and RagB$^{shortT54N}$ proteins purified from RagABKO HEK293T cells as dimers with FLAG-tagged RagC$^{Q120L}$ (1 μg dimer per lane). The experiment was repeated once. **i,j**, Malachite-green GTPase assay with 1 μM of RagA•RagC$^{S75N}$ either alone or mixed with an equimolar amount of RagB$^{longT54N}$•RagC$^{Q120L}$ (**i**) or RagB$^{shortT54N}$•RagC$^{Q120L}$ (**j**) in solution in the presence of the indicated amounts of GATOR1. Quantification of four (**i**) or three (**j**) independent experiments. Circle indicates average, and error bars represent standard deviation; $n = 3$ (**j**) and 4 (**i**) replicates. Two-way ANOVA and Sidak's post-hoc test. **k**, Schematic representation of the mechanism whereby non-GTP-bound RagB$^{long}$ titrates away the GAP interface of GATOR1. Exact $P$ values are shown in the graphs. Source numerical data and unprocessed blots are available in source data.

If this is the case, then the difference between RagA and RagB$^{short}$ should be gone if DEPDC5 cannot bind the Rag proteins. Indeed, re-expression of wild-type DEPDC5 inhibited mTORC1 more strongly in RagA-expressing cells than in RagB$^{short}$-expressing cells (lane 3 versus lane 7, Fig. 3e,f), consistent with RagB$^{short}$ being comparatively resistant to GATOR1, while this difference was gone in cells expressing DEPDC5$^{Y775A}$ (Fig. 3g,h). These results indicate that RagB$^{short}$ is resistant to GATOR1 activity not because it is less sensitive to the GAP activity (indeed, RagB$^{short}$ binds equally well to the GAP side of GATOR1 as RagA, Fig. 2i–k), but rather

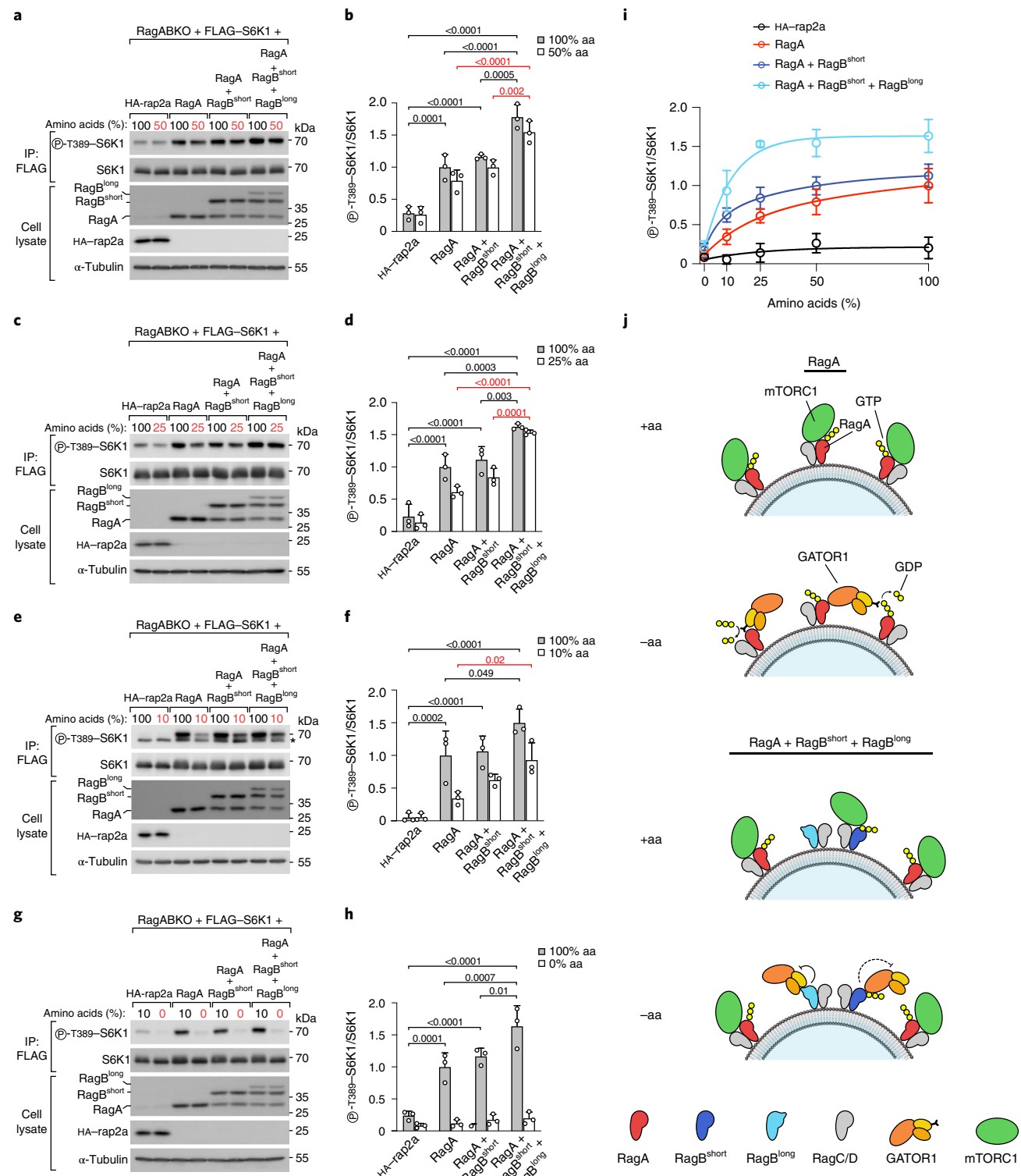

**Fig. 5 | RagB^short and RagB^long have additive effects on mTORC1. a–h,** S6K1 phosphorylation in RagABKO cells transiently transfected with RagA, RagA and RagB^short, or all three RagA/B isoforms and treated with amino-acid-rich medium or medium containing 50% (**a** and **b**), 25% (**c** and **d**), 10% (**e** and **f**) or 0% (**g** and **h**) of the normal amino-acid concentration for 30 min, where cells transfected with HA–rap2a were used as a negative control: representative examples (**a**, **c**, **e** and **g**) and quantification of three independent experiments, with RagA-transfected cells set to 1 (**b**, **d**, **f** and **h**). Asterisk indicates non-specific band. Bar height indicates average, and error bars represent standard deviation; $n = 3$ biological replicates. Two-way ANOVA and Sidak's post-hoc test. **i**, Graph summarizing the data from **a–h** after fitting a hyperbola curve using non-linear regression analysis. Circle indicates average, and error bars represent standard deviation; $n = 12$ biological replicates for the 100% amino-acid condition, $n = 3$ biological replicates for the other conditions. **j**, Schematic diagram of the effects of different RagA/B isoform combinations on mTORC1 signalling in the presence or absence of amino acids. RagB^short and RagB^long cause persistent mTORC1 activation during amino-acid starvation via GATOR1 inhibition due to RagB^short binding to the inhibitory interface and RagB^long binding to the GAP interface. Exact $P$ values are shown in the graphs. Source numerical data and unprocessed blots are available in source data.

because RagB[short], but not RagA, is able to inhibit GATOR1 through DEPDC5 binding (Fig. 3i).

RagB[short] differs from RagA at the N-terminal extension and at five other amino acids, four of which are located in the CRD domain and one in the C-terminal part of the GTPase domain (Extended Data Fig. 6f). To determine which of these features is responsible for the functional difference between RagA and RagB[short], we generated a mutant of RagB[short] lacking the N-terminal extension (ΔN mutant) or a mutant of RagB[short] where the five amino acids are swapped to the RagA version (AQVHS mutant) (Extended Data Fig. 6f). Only removal of the N-terminal extension restored the interaction with GATOR1 to the same level as RagA (Extended Data Fig. 6g,h), as well as the ability to inactivate mTORC1 upon amino-acid removal (Extended Data Fig. 6i,j). This is consistent with the spatial proximity between the N-terminal extension of RagB and the switch I region that mediates GATOR1 binding via DEPDC5.

We aimed to recapitulate these effects in an in vitro GAP assay. We purified GATOR1 from HEK293T cells and the two Rags from bacteria as dimers with a RagC mutant (S75N) that abolishes GTP binding so that only the GTPase activity of the RagA or RagB isoforms would be measured (Extended Data Fig. 7a,b). We used a multiple-turnover GAP assay with malachite green to detect the phosphate released from GTP (Extended Data Fig. 7c), and we immobilized the purified RagA●RagC[S75N] and RagB[short]●RagC[S75N] dimers on the surface of beads, since the spatial orientation of GATOR1 and neighbouring Rag dimers could be important for the inhibitory mechanism through DEPDC5 binding, as discussed above[35]. Using this set-up, we found that also in vitro GATOR1 had reduced activity towards RagB[short] compared with RagA (Extended Data Fig. 7d) and this difference was reduced when using GATOR1 containing DEPDC5[Y775A] (Extended Data Fig. 7e), consistent with the in vivo results (Fig. 3a–h).

In sum, these results suggest that the N-terminal extension of RagB[short] enables it to inhibit GATOR1 via DEPDC5.

**RagB[long] has low affinity for GTP.** We next turned our attention to RagB[long]. RagB[long] binds weakly to Raptor (Extended Data Figs. 4e,f and 5a,b) and to the inhibitory interface of GATOR1 (Fig. 2f–h). As both interactions are enhanced when RagA/B are loaded with GTP, one possible explanation is that RagB[long] has reduced affinity for GTP. Indeed, the 28-amino-acid insertion in RagB[long] resides within the switch I region, which forms part of the GTP-binding pocket. A GTP pull-down assay revealed that RagB[long] interacts much less with GTP than RagA or RagB[short], almost at background levels (Fig. 4a–c). These results are in line with a previous report

in which a radiolabelled-GTP binding assay was employed[11]. As additional confirmation, we also assessed the interaction of the RagA and RagB isoforms with the p18 subunit of Ragulator and the folliculin complex (FLCN–FNIP2), two complexes that sense the nucleotide loading state of the Rag GTPases but do not interact directly with switch I (refs. [24,25,36,37]). As expected, mutations that disrupt GTP binding led to strong interaction of RagA or RagB[short] with both FLCN–FNIP2 and p18, while mutation of the catalytic glutamine causing RagA or RagB[short] to lock into GTP binding abrogated or reduced FLCN–FNIP2 and p18 interaction (Extended Data Fig. 8a–d). In contrast, the analogous mutant of RagB[long] that cannot hydrolyse GTP retained substantial interaction with FLCN–FNIP2 and p18 (Extended Data Fig. 8a–d), consistent with RagB[long] having impaired GTP binding. Likewise, wild-type RagB[long] bound p18 more strongly than RagA or RagB[short] (Extended Data Fig. 8e,f). In sum, all these data indicate that RagB[long] binds GTP less well than RagA or RagB[short].

To understand why RagB[long] has reduced GTP binding, we mutagenized individually all the 28 amino acids encoded by exon 4 of *Rragb*. Among all the mutants screened, we found that only L94A could increase GTP binding, albeit only mildly (Extended Data Fig. 8g,h and data not shown). This suggests that the low affinity of RagB[long] for GTP depends on complex structural features rather than on the identity of one specific residue. Consistent with increased GTP binding, addition of the L94A mutation to the GTP-locking mutant of RagB[long] decreased its affinity for FLCN–FNIP2, but not the neighbouring D96A mutation that does not increase GTP binding (Extended Data Fig. 8i,j). Finally, we tested to what extent the weak binding of RagB[long] to GATOR1 via DEPDC5 depends on its particular switch I sequence or its N-terminal extension. Consistent with the additional 28-amino-acid loop in the switch I sequence of RagB[long] imposing a large constraint, removal of the N-terminal extension of RagB[long] did not visibly improve GATOR1 binding via DEPDC5 (Extended Data Fig. 8k,l). In sum, the 28-amino-acid insertion in RagB[long] impairs both GTP binding and GATOR1 binding via DEPDC5.

Consistent with these in vivo data, in vitro GAP assays revealed that GATOR1 was hardly able to stimulate GTP hydrolysis by RagB[long] purified from HEK293T cells (the yield of RagB[long] purified from bacteria was too low) (Extended Data Fig. 9a–c), whereas it efficiently stimulated GTP hydrolysis by RagA purified from either bacteria or HEK293T cells (Extended Data Fig. 9c,d).

**RagB[long] titrates away the GAP interface of GATOR1.** The low affinity of RagB[long] for GTP and its strong interaction with the GAP

---

**Fig. 6 | RagB determines mTORC1 resistance to starvation in neurons and tumour cells. a–c**, mRNA levels of *Rraga* (**a**), *Rragb* (**b**) and their ratio (**c**) in brain cells (ref. [39]; GEO accession number GSE52564); n = 2 biological replicates; OL, oligodendrocytes; FPKM, fragments per kilobase million. **d**, RagA/B expression in DIV 5 mouse cortical neurons and MEFs, two biological replicates each. GAPDH was loading control. **e,f**, Control (shmCherry) or RagA- or RagB-knockdown DIV 10 mouse cortical neurons were starved of amino acids for 24 h with/without bicuculline (50 μM). −aa, amino-acid-free MEM 1:10 in buffered saline solution; +aa, −aa medium + 1× amino acids (Methods): representative example (**e**) and quantification of three biological replicates, with shmCherry +aa/−bicuculline set to 1 (**f**). Bar height indicates average, and error bars represent standard deviation; n = 3 biological replicates. Two-way ANOVA and Tukey's post-hoc test. **g,h**, RagB but not RagA knockdown blunts bicuculline-stimulated dendritogenesis of hippocampal neurons grown in 10% MEM amino-acid concentration (methods). Bar height indicates average, and error bars represent standard deviation; n = 3 biological replicates. One-way ANOVA and Dunnett's post-hoc test. **i–k**, Violin plot of of *Rraga/Rragb* mRNA levels and their ratio in normal (GTEX) or cancer (TCGA) tissues. $n_{GTEX} = 7,825$ biological replicates, $n_{TCGA} = 10,534$ biological replicates. **l**, Violin plot of the *Rragb/Rraga* ratio in normal brain (GTEX brain) or brain cancers (TCGA brain), compared with all other samples (GTEX non-brain and TCGA non-brain). $n_{GTEX\ non-brain} = 6,688$ biological replicates, $n_{TCGA\ non-brain} = 9,840$ biological replicates, $n_{GTEX\ brain} = 1,136$ biological replicates, $n_{TCGA\ brain} = 694$ biological replicates. **m**, RagA/B expression in HEK293T and various cancer cell lines. The experiment was repeated once. **n–q**, mTORC1 inactivation in EFO21 cells knockdown for RagA (**n** and **o**) or RagB (**p** and **q**) during starvation, where *Renilla* luciferase was negative control: representative examples (**n** and **p**) and quantification of three independent experiments, with control at timepoint 0 set to 1 (**o** and **q**). Circle indicates average, and error bars represent standard deviation; n = 3 biological replicates. Two-way ANOVA and Sidak's post-hoc test. **r**, OPP incorporation in RagA-, RagB- or luciferase-knockdown EFO21 cells treated with the indicated amino-acid concentrations. The experiment was repeated twice with three biological replicates each. Line indicates average, and error bars represent standard deviation; n = 6 biological replicates. Two-way ANOVA and Sidak's post-hoc test. Gating strategy in Extended Data Fig. 10m. Exact P values are shown in the graphs. NS, not significant. Source numerical data and unprocessed blots are available in source data.

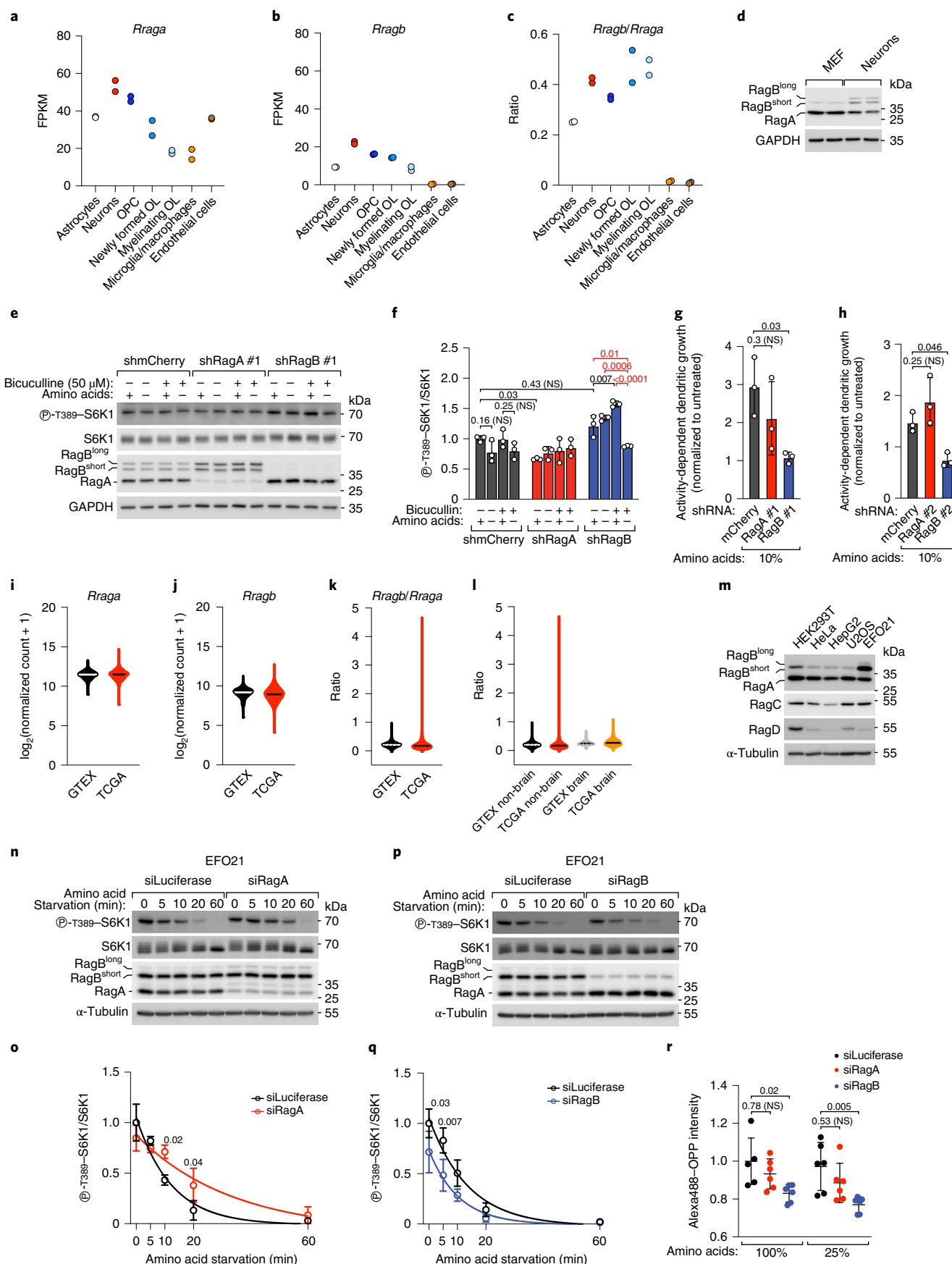

interface of GATOR1 when non-GTP bound (Fig. 2i–k) made us reason that a substantial pool of non-GTP-bound RagB[long] could act as a 'sponge' that binds GATOR1 on the GAP interface and titrates it away from RagA, RagB[short] or the small pool of GTP-bound RagB[long] in a cell. In this way, RagB[long] would act as a GATOR1 inhibitor. This possibility is consistent with the RagA/B proteins being stoichiometrically in great excess compared with the GATOR1 subunits (RagA/B, 58,467.8 protein copies per cell; DEPDC5, 649.6; Nprl2, 11,268.6; Nprl3, 9,860.6; data from HeLa cells)[38]. Indeed, we found that non-GTP-bound RagB[long] harbouring the T54N mutation outcompetes GTP-locked RagA for Nprl2/3 binding (Fig. 4d,e). By acting as a GATOR1 inhibitor, RagB[long] should confer increased resistance to nutrient starvation when co-expressed with the other RagA/B isoforms. Indeed, co-expression of either wild-type RagB[long] or the RagB[longT54N] mutant together with RagA caused less inactivation of mTORC1 upon amino-acid starvation compared with expression of RagA alone (Fig. 4f,g). As the RagB[longT54N] mutant cannot bind mTORC1, this is consistent with RagB[long] acting via GATOR1 inhibition rather than direct mTORC1 activation.

We next tested if RagB[long] can inhibit GATOR1 GAP activity towards RagA also in vitro through this mechanism. We first confirmed that also in vitro RagB[long] binds the GAP interface of GATOR1 more strongly than RagA or RagB[short] using purified GDP-loaded Rag dimers and a GATOR1 complex containing DEPDC5[Y775A] to disrupt the inhibitory interface (Extended Data Figs. 7b and 9e,f). Next, to test if RagB[long] can inhibit GATOR1 specifically via the GAP interface, we purified RagB[longT54N], which interacts strongly with the GAP interface of GATOR1 but not its inhibitory interface (Fig. 2f–k), as a dimer with GTP-locked RagC[Q120L] (Fig. 4h) to mimic the Rag conformation probably occurring in vivo owing to inter-subunit crosstalk[15]. Consistent with direct GATOR1 inhibition, the GAP activity of GATOR1 towards RagA was lower in the presence of equimolar amounts of RagB[longT54N] (Fig. 4h,i) but not of the analogous RagB[shortT54N] (Fig. 4j). This indicates that inhibition of GATOR1 via GAP-interface binding is a specific feature of RagB[long], and that RagB[short] instead only inhibits GATOR1 via the inhibitory interface (Fig. 3 and Extended Data Fig. 7).

In sum, we find that RagB[long] inhibits GATOR1 by acting as a 'sponge' for its GAP interface to potentiate signalling through the other Rag isoforms (Fig. 4k).

**RagB[short] and RagB[long] additively inhibit GATOR1.** Since RagB[short] and RagB[long] inhibit GATOR1 activity through distinct mechanisms—RagB[short] via DEPDC5 binding and RagB[long] via GAP-interface binding—we predicted that co-expression of both isoforms together with RagA, as physiologically observed in the brain, would have additive effects. To this end, we transfected RagABKO cells with equal total amounts of RagA, RagA + RagB[short], or RagA + RagB[short] +RagB[long]. Cells transfected with all three isoforms had total Rag protein levels similar to cells transfected with only RagA (Extended Data Fig. 9g), thus allowing us to see differences in RagA versus RagB function rather than levels. In line with our prediction, we observed that, although complete and prolonged amino-acid starvation caused comparable mTORC1 inhibition in all cases, co-expression of all three RagA/B isoforms enabled cells to maintain elevated mTORC1 activity when amino-acid levels were reduced to 50%, 25% or 10% of normal cell culture levels, compared with cells expressing only RagA or RagA + RagB[short] (Fig. 5a–i). Analogously, in a time course of complete amino-acid starvation, cells expressing all three RagA/B isoforms inactivated mTORC1 more slowly than cells expressing only RagA (Extended Data Fig. 9h,i).

To test if similar effects can also be observed in a cell line that endogenously expresses all three RagA/B isoforms, we turned to the mouse neuroblastoma Neuro-2a cell line. As expected, knockout of RagA, which leaves only the more GATOR1-resistant RagB

isoforms in the cells, caused mTORC1 activity to be lower and largely insensitive to amino-acid removal (Extended Data Fig. 9j,k), similar to what was observed in HEK293T cells (Fig. 1k,l). In contrast, knockout of the RagB isoforms, which leaves only RagA in the cells, reduced mTORC1 activity but at the same time caused the cells to remain amino-acid sensitive (Extended Data Fig. 9l,m), paralleling the differences seen between HEK293T cells expressing all three RagA/B isoforms versus only RagA (Fig. 5i).

In sum, these results indicate that RagB[short] and RagB[long] inhibit GATOR1 additively to render mTORC1 more resistant to nutrient depletion, according to the scheme in Fig. 5j. This may cause different tissues in the body, which express different ratios of RagA, RagB[short] and RagB[long], to regulate mTORC1 activity differently upon amino-acid restriction.

**RagB maintains high mTORC1 in neurons during starvation.** RagB[short] and RagB[long] are abundantly expressed in the mammalian brain (Fig. 1a,b). Interrogation of a transcriptomic dataset of the main cell types of the brain[39] revealed that RagB is expressed in neurons, astrocytes and oligodendroglial lineage cells, but not in microglia or endothelial cells (Fig. 6a–c). Moreover, the ratio of RagB to RagA is highest in neurons and oligodendroglial lineage cells. We confirmed that RagB[short] and RagB[long] are highly expressed in neurons also at the protein level as compared with mouse embryonic fibroblasts (MEFs) isolated in parallel from the same mouse embryos (Fig. 6d). In contrast, Rheb and its GAP TSC2, relaying growth factor signalling to mTORC1, did not differ strongly between neurons and MEFs (Extended Data Fig. 10a).

On the basis of the mechanism outlined above, expression of all three RagA/B isoforms should lead to relative resistance of mTORC1 activity to nutrient deprivation, analogous to the RagABKO cells reconstituted to express all three RagA/B isoforms (Fig. 5a–i). Indeed, we found that mTORC1 activity in neurons is rather resistant to amino-acid removal. A time course of amino-acid starvation at day in vitro (DIV) 5 showed slower dynamics of mTORC1 inactivation in neurons as compared with MEFs, with S6K phosphorylation being largely retained after 30 min of amino-acid starvation (Extended Data Fig. 10b,c). In more mature neurons (DIV 10), mTORC1 activity became largely resistant to amino-acid removal despite complete and prolonged starvation for 24 h and simultaneous stimulation of synaptic activity with bicuculline to increase their metabolic rate (Fig. 6e,f and Extended Data Fig. 10d,e).

We then tested the contribution of the RagA/B isoforms to this resistance. Knockdown for RagB, but not RagA, caused a significant drop in S6K phosphorylation in response to combined amino-acid starvation and bicuculline treatment (Fig. 6e,f and Extended Data Fig. 10f,g), suggesting that high RagB levels in neurons are at least partially responsible for their resistance to nutrient depletion and that this effect depends on neuronal activity, possibly because it increases protein synthesis and thus amino-acid usage[40]. Growth of dendritic arbours, key for neuronal function, is controlled by extracellular cues such as synaptic activity and by intracellular signalling cascades, including mTORC1, which in this context has been mostly studied downstream of growth factor signalling[41–45]. We therefore asked whether nutrient sensing through the RagA/B proteins could also be involved in this process by assessing dendrite growth in a standardized medium containing a low amino-acid concentration (10% of MEM)[46,47] and upon synaptic stimulation with bicuculline. Consistent with the changes in mTORC1 activity detected by western blot, knockdown of RagB, but not RagA, using two independent sets of short hairpin RNAs (shRNAs) blunted the synaptic-activity-dependent stimulation of dendritic growth (Fig. 6g,h).

**RagB maintains high mTORC1 in tumour cells during starvation.** Somatic inactivating mutations in all GATOR1 components

have been reported in small subsets of glioblastomas and ovarian cancers[16]. Additionally, 17% of follicular lymphomas display RagC mutations that cause partial mTORC1 resistance to low nutrients[48], indicating that disruption of the nutrient-sensing mechanism of mTORC1 is a recurring event in various cancer types. Analogously, elevated expression of the more GATOR1-resistant RagB isoforms could be an alternative route to maintaining mTORC1 activity under the low nutrient conditions experienced by cancer cells. Consistent with this hypothesis, a comparison of RagA and RagB (short and long) transcript levels in normal (GTEX) and cancer samples (TCGA) revealed that, although variable when considered separately (Fig. 6i,j), the ratio RagB/RagA was substantially increased in a subset of cancer samples (Fig. 6k). This increase did not depend on a potential overrepresentation of brain samples in the TCGA collection, since an identical trend could be appreciated also when considering only non-brain cancers (Fig. 6l). Among the samples that showed a RagB/RagA ratio greater than 0.5, we found disparate cancer primary sites and, interestingly, an enrichment in acute myeloid leukaemia samples as compared with the whole TCGA collection (Extended Data Fig. 10h,i). Interestingly, cancer samples with low RagB/RagA ratios also had lower expression of each of the three GATOR1 subunits as compared with samples with RagB/RagA ratios greater than 0.5 (Extended Data Fig. 10j–l), consistent with the hypothesis that high RagB or low GATOR1 expression could be alternative mechanisms to confer nutrient stress resistance to cancer cells. To validate these findings experimentally, we interrogated the Cancer Cell Line Encyclopedia for cell lines with high expression of the RagB isoforms and found the EFO21 ovarian cancer cell line as the top-scoring one. EFO21 cells have high levels of RagB^short protein compared with RagA in comparison with other normal or cancer cell lines, together with low but detectable RagB^long expression (Fig. 6m). We then tested the functional relevance of this profile of RagA/B expression by knocking down RagA or the RagB isoforms. As expected, depletion of the RagB isoforms lowered mTORC1 activity both under nutrient-rich conditions and during amino-acid starvation (Fig. 6p,q), while depletion of RagA (leaving only RagB in the cell) caused more persistent mTORC1 activity (Fig. 6n,o). Consistent with these results, knockdown of RagB in EFO21 cells caused lower protein synthesis rate, detected via incorporation of fluorescently labelled O-propargyl-puromycin (OPP), especially when exposed to lower amino-acid concentrations (Fig. 6r).

In sum, these results indicate that high expression of the RagB isoforms, as observed physiologically in neurons and pathologically in some cancers, confers partial resistance to low nutrients and thus contributes to determining cell-specific dynamics of mTORC1 responses.

## Discussion

On the basis of their high sequence similarity, RagA and RagB have been assumed to be functionally equivalent. The findings presented here, together with those in an accompanying study[34], show that this is not the case. RagA and the two RagB isoforms have distinct profiles of interaction with Raptor and GATOR1, which correspond to different levels of mTORC1 activity and result in different responses to nutrient deprivation. First, RagB^long interacts only weakly with Raptor as compared with RagA and RagB^short and thus cannot sustain strong lysosomal accumulation and mTORC1 activity in nutrient-rich conditions, when expressed alone. Second, RagB^short and RagB^long, but not RagA, are able to counteract the activation of GATOR1 upon nutrient removal by inhibiting this complex through two distinct mechanisms, which cause persistent mTORC1 activity. When all isoforms are co-expressed, the two inhibitory mechanisms provided by the RagB isoforms potentiate mTORC1 activity and slow down the dynamics of its inactivation when nutrients are scarce. High expression of the two RagB isoforms, together with RagA, is observed physiologically in neurons, where it helps

maintain mTORC1 activity during amino-acid starvation. We finally show that cancer cells can use this mechanism to their advantage by upregulating the RagB isoforms.

The inhibition of GATOR1 by RagB^short depends on the interaction between this Rag isoform and the GATOR1 subunit DEPDC5. Previously, an inhibitory interaction between RagA/B and GATOR1 via DEPDC5 was described in cells endogenously expressing both RagA and RagB^short and named 'inhibitory mode' to distinguish it from the interaction via Nprl2/3 that executes the GAP activity[17]. Our results indicate that the inhibitory mode is probably a specific feature of RagB^short binding to DEPDC5 and that, while RagA also interacts with DEPDC5, this interaction is compatible with normal GATOR1 activity. Thus, the inhibitory effect detected in cells expressing both RagA and RagB^short might actually reflect just the contribution of RagB^short-mediated inhibition of GATOR1. This finding is rather unexpected since RagA and RagB^short are likely to have very similar interfaces for DEPDC5 binding. Although no structure of RagB^short in complex with GATOR1 nor of RagB^short alone is yet available, the N-terminal extension of RagB^short is expected to be in the vicinity of the DEPDC5–switch I interface on the basis of a published structure of GATOR1 in complex with RagA[17]. Hence it is possible that this N-terminal extension of RagB^short acts on DEPDC5. Consistent with this, we observed that removal of the N-terminal extension restores the same affinity for GATOR1 as RagA and results in complete mTORC1 inactivation upon nutrient starvation as in RagA-expressing cells (Extended Data Fig. 3g–j). The N-terminal extension of RagB^short might then force an orientation of the GATOR1 complex where the Nprl2/3 subunits are less favourably placed for the execution of the GAP activity on the nearby RagA/B GTPases (Fig. 3i).

RagB^long had been previously described as a brain-specific splicing isoform of RagB of unknown function[11]. Here we show that RagB^long has very peculiar features that distinguish it from the other two RagA/B isoforms and from the other Rag GTPases altogether: (1) it has low affinity for GTP; (2) it interacts poorly with Raptor; (3) it interacts strongly with the Nprl2/3 subunits of GATOR1 when not bound to GTP. On this basis, we propose that its primary function is to act as a 'sponge' that obstructs the GAP-competent interface of GATOR1 from the other RagA/B isoforms, thereby enhancing mTORC1 activity. The low affinity of RagB^long for GTP is therefore a key aspect of its function, as it enables RagB^long to interact strongly with Nprl2/3 while the other RagA/B isoforms are GTP bound, and it allows RagB^long to be insensitive to GATOR1 without hyperactivating mTORC1. RagB^long has close parallels to the so-called pseudo-GTPases, a group of atypical GTPases defined by their inability to bind GTP or, alternatively, to hydrolyse it[49]. While the low GTP binding of pseudo-GTPases usually depends on mutations in the G1 motif interacting with the phosphates or in the G4 and G5 motifs interacting with the guanosine ring, in the case of RagB^long this is probably achieved through insertion of the sequence encoded by exon 4 of Rragb in its switch I region, which might alter the GTP binding pocket, together with providing a high-affinity interface for Nprl2/3 binding. Future studies will be needed to better understand which consequences this sequence imposes on the structure of RagB^long and to correlate such structural features with the functional peculiarities of this RagA/B isoform.

Both RagB isoforms are highly expressed in neurons. We find here that mTORC1 activity in neurons is exceptionally resistant to nutrient depletion and that this effect depends at least in part on their high RagB levels. What is the physiological function of the resistance of neuronal mTORC1 to nutrient starvation? At the organismal level, the various organs are metabolically wired to privilege those fulfilling vital functions such as the brain. Thus, starvation induces autophagy in most organs within 24 h, including liver, pancreas, kidney, skeletal muscle and heart, but not in the brain[50]. Autophagy-derived substrates, such as nucleosides, amino acids

and lipids are then metabolized in the liver to produce glucose and ketone bodies necessary to feed the brain[51]. As mTORC1 is a potent inhibitor of autophagy, resistance of neuronal mTORC1 to nutrient starvation could then support the preferential upregulation of autophagy in the peripheral tissues and the subsequent re-routing of nutrients to the brain. Additionally, mTORC1 is involved in a variety of crucial neuron-specific functions, including dendritic growth[43–45], synaptic plasticity[4], axon guidance and both developmental and adult neurogenesis[4,52,53]. High expression of the RagB isoforms in neurons could therefore lend more robustness to these critical processes, by ensuring that oscillations in nutrient levels do not immediately result in corresponding changes in mTORC1 activity. Consistent with this interpretation, we find that the stimulation of dendritic growth by synaptic activity is blunted upon knockdown of the RagB isoforms when neurons are exposed to low amino-acid concentrations.

Germline monoallelic loss-of-function mutations in all three GATOR1 subunits have been recognized as the most frequent genetic cause of focal epilepsies, accounting for almost 10% of the cases[54]. Moreover, bi-allelic loss-of-function mutations in some subunits of the KICSTOR complex, anchoring GATOR1 to the lysosomal surface[55,56], have been associated with a broad range of neurodevelopmental disorders, including mental retardation and developmental and epileptic encephalopathy[57–64]. The propensity of such whole-body mutations to specifically give rise to neurological manifestations seems at odds with the ubiquitous importance of the mTORC1 signalling axis. As a comparison, mutations in the TSC complex lead to skin lesions such as facial angiofibromas, kidney angiomyolipomas and pulmonary lymphangiomyomatosis, alongside neurologic disorders[65]. The inhibition of GATOR1 by the RagB isoforms that we describe here helps explain this discrepancy, as a decrease in GATOR1 function is expected to have the most dramatic consequences in cells expressing higher amounts of these two isoforms. Genetic or pharmacological inhibition of the RagB isoforms might then be a particularly appealing therapeutic intervention, as it could partially restore GATOR1 function selectively in the brain.

Components of the nutrient-sensing machinery are often mutated in cancer. A priori, it is unclear whether one would expect it to be beneficial or deleterious for cancer cells to properly sense their nutrient status. For instance, one could imagine that the ability to sense the lack of nutrients and thereby downregulate anabolic processes might be important for cancer cells to avoid metabolic catastrophe and ensure viability. By contrast, one might imagine that having constitutively high activation of mTORC1 might give a growth and proliferation advantage to cancer cells. This logic may depend on the type of tumour in consideration, and on the location of cells within the tumour: leukaemia cells are likely to have constitutively high nutrient availability and may therefore benefit more from constitutively active nutrient sensing, whereas cells at the centre of a solid tumour that is poorly perfused might survive better with an intact nutrient stress response. Hence, it is interesting to look at what has been found from cancer sequencing. DEPDC5, a component of the GATOR1 complex, is frequently mutated in gastrointestinal stromal tumours, leading to constitutively high mTORC1 activity[66]. All three components of the GATOR1 complex are mutated in a variety of carcinomas, including hepatocellular carcinoma, bladder carcinoma and breast carcinoma[16]. Likewise, recurrent gain-of-function mutations in RagC that increase mTORC1 binding and activation were found in follicular lymphoma[48]. Together, these data indicate that constitutive activation of mTORC1 is advantageous for some cancer cells despite the cost of impaired nutrient sensing. The data we present here fit this general trend. We find that many tumour cells shift their relative balance of RagA/B expression towards RagB and away from RagA. Our data indicate that this leads to stronger and more constitutive activation of mTORC1, thereby providing cancer cells with a growth advantage.

Altogether, we provide here evidence that the functions of RagA and RagB have substantially diverged during evolution, with RagB assuming the role of a GATOR1 decoy substrate and inhibitor that enables elevated mTORC1 activity in a tissue-specific manner.

## Online content

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

## Methods

All experiments conform to the relevant regulatory standards of the German Cancer Research Center and have been approved by the ethical authorities, Regierungspräsidium Karlsruhe, Germany (G-196/17). Further information on research design, statistics and technical information is available in the Nature Research Reporting Summary linked to this article.

**Antibodies.** The following primary antibodies were used: RagA (recognizing also both RagB isoforms) (Cell Signaling Technology #4357), RagC (Cell Signaling Technology #9480), RagD (Cell Signaling Technology #4470), phospho-S6K1 T389 (Cell Signaling Technology #9205), S6K1 (Cell Signaling Technology #2708), phospho-TFEB S211 (Cell Signaling Technology #37681), TFEB (Cell Signaling Technology #37785), phospho-4EBP1 S65 (Cell Signaling Technology #9451), 4EBP1 (Cell Signaling Technology #9452), FLAG tag (Sigma-Aldrich #F7425, 1:2500), α-tubulin (Sigma-Aldrich #T9026, 1:5000), mTOR (Cell Signaling Technology #2983, 1:200 for immunofluorescence, 1:1,000 for western blots), Raptor (Cell Signaling Technology #2280), LAMP2 (Hybridoma Bank #H4B4, 1:50 for immunofluorescence, 1:500 for western blots), calreticulin (Cell Signaling Technology #12238), VDAC (Cell Signaling Technology #4661), HA tag (Cell Signaling Technology #2367, used to detect HA-tagged proteins in whole-cell lysates), HA tag (Cell Signaling Technology #3724, used to detect HA-tagged proteins in immunoprecipitates), LAMTOR1/p18 (Cell Signaling Technology #8975), GAPDH (Cell Signaling Technology #2118), myc tag (Cell Signaling Technology #2278), TSC2 (Cell Signaling Technology #4308), Rheb (Abnova #H00006009-M01), DEPDC5 (Abcam #ab185565, used to screen DEPDC5KO clones), calnexin (Enzo #ADI-SPA-960-D). All antibodies were diluted 1:1,000 unless indicated otherwise. The following secondary antibodies were used: anti-rabbit HRP (Jackson ImmunoResearch #111-035-003, used 1:10,000), anti-mouse HRP (Jackson ImmunoResearch #115-035-003, used 1:10,000), anti-rabbit Alexa488 (Life Technologies #A11008, used 1:500), anti-mouse TRITC (Jackson ImmunoResearch #715-025-151, used 1:200).

**Cell lines and amino-acid treatments.** HEK293T (ATCC #CRL-3216), Neuro-2a (ATCC #CCL-131), U2OS (ATCC #HTB-96), HeLa (ATCC #CCL-2) and HepG2 (ATCC #HB-8065) cells were cultured in DMEM (Life Technologies #41965-062) containing 25 mM glucose, 4 mM L-glutamine, 1× penicillin–streptomycin (Life Technologies #15140-122) and 10% FBS. HEK293 cells (Stratagene #240073) for recombinant adeno-associated viral vector (rAAV) production were cultured in DMEM supplemented with 10% FBS, non-essential amino acids (Life Technologies #11140-035; 1:100), sodium pyruvate (Life Technologies #11360-039; 1:100) and 50 U ml$^{-1}$ penicillin–streptomycin. EFO21 cells (DMSZ #ACC 235) were cultured in RPMI 1640 (Life Technologies #52400-025) containing 1× MEM non-essential amino acids (Life Technologies #11140035), 1 mM sodium pyruvate (Life Technologies #11360070), 2 mM L-glutamine (Life Technologies #25030024), 1× penicillin–streptomycin (Life Technologies #15140-122) and 20% FBS. All cell lines were kept in 5% $CO_2$ at 37 °C and split every other day or when subconfluent using trypsin–EDTA 0.25% (Life Technologies #25200-056). All cell lines were tested for mycoplasma to exclude contamination. No cell line authentication was performed.

For amino-acid starvation of cell lines, homemade DMEM containing 25 mM glucose was prepared according to the formulation of the commercial medium but omitting all amino acids, and completed by addition of 10% dialysed FBS (dFBS) (Life Technologies #A33820-01) (−aa medium). For amino-acid stimulation, the medium was supplemented with 1× MEM non-essential amino acids (Life Technologies #11140035), 1× MEM amino acids (Sigma-Aldrich #M5550) and 2 mM L-glutamine (Life Technologies #25030024) (+aa medium). Before any amino-acid treatment, cells were first incubated in +aa medium for 1 h.

**Plasmids.** The coding sequence of RagB$^{long}$ was PCR-amplified from EFO21 cDNA using Phusion High-Fidelity Polymerase (New England Biolabs #M0530) and cloned into pcDNA3.1(+)-based vectors. The coding sequences of metap2, rap2a, FLCN and FNIP2 were PCR-amplified from HEK293T cDNA using Phusion High-Fidelity Polymerase and cloned into pRK5- or pcDNA3.1(+)-based vectors. pRK5-based plasmids coding for S6K1 (Addgene #100509) (ref. [21]), Raptor (Addgene #8513) (ref. [67]), DEPDC5 (Addgene #46327) (ref. [16]), Nprl2 (Addgene #99709) (ref. [15]) and Nprl3 (Addgene #46330) (ref. [16]) were from David Sabatini. Plasmids coding for RagA, RagB$^{short}$, RagC and RagD were previously described[33]. When needed, tags were exchanged by subcloning the inserts into pRK5- or pcDNA3.1(+)-based plasmids containing the tag of interest. The pSpCas9(BB)-2A-Puro (px459) plasmid for CRISPR–Cas9 gene editing (Addgene #62988) (ref. [68]) was a gift from Feng Zhang. The LJC5–Tmem192–3xHA plasmid used for lyso-IPs (Addgene #102930) is from ref. [69]. Point mutations were introduced through PCR-based site-directed mutagenesis. All plasmids were verified by Sanger sequencing.

**Generation of knockout cell lines, transient and stable plasmid transfections, and siRNA knockdown.** Knockout cell lines were generated through CRISPR–Cas9 editing using px459 plasmids expressing gRNAs from the Brunello library[70] (human cell lines) or from CHOPCHOP (https://chopchop.cbu.uib.no/)[71] (mouse cell lines) (Supplementary Table 1). Single clones were screened by western blot

for loss of the protein of interest and then confirmed by Sanger sequencing of the target locus after PCR amplification (Supplementary Table 1) and TOPO cloning (Life Technologies #450640).

For transient plasmid transfections, $4 \times 10^6$ HEK293T cells were seeded in 10-cm dishes. One day later, cells were co-transfected with pRK5 or pcDNA3/pcDNA3.1(+)-based plasmids using PEI (Polysciences #23966-1) at a 3:1 ratio with DNA in the following amounts: 30 ng FLAG-S6K1, 5/25/100 ng of each HA-tagged GATOR1 subunit or of HA-tagged wild-type or Y775A-mutated DEPDC5 for titration experiments, 200 ng of each HA-tagged GATOR1 subunit in rescue experiments, 600 ng RagA, 300 ng RagA and 300 ng RagB$^{short}$, or 200 ng RagA, 200 ng RagB$^{short}$ and 200 ng RagB$^{long}$ when comparing the various Rag combinations in Fig. 5a–i and Extended Data Fig. 9h,i; 600 ng RagA, 500 ng RagA and 100 ng RagB$^{long}$, 520 ng RagA and 80 ng RagB$^{longT54N}$ when comparing the various Rag combinations in Fig. 4f,g; 400 ng in all other cases. Empty pcDNA3.1(+) plasmid was added to the transfection mix to reach a total of 10 μg DNA per 10-cm dish. To assess mTORC1 activity specifically in transfected cells upon transient transfection, FLAG–S6K1 was co-transfected and immunoprecipitated. For stable transfections, the coding sequences of the proteins of interest were cloned or subcloned in a modified pcDNA3.1(+) plasmid where the neomycin cassette was exchanged with a puromycin cassette, transfected using Lipofectamine 2000 (Life Technologies #11668500), and grown in the presence of 1 μg ml$^{-1}$ puromycin (Sigma-Aldrich #P9620) until a resistant polyclonal cell population emerged. For knockdown experiments, $3 \times 10^6$ EFO21 cells were reverse transfected in 15-cm dishes with 15 nM pools of four siRNAs against human RagA (siGENOME, Horizon Discovery #M-016070-01-0005), human RagB (siGENOME, Horizon Discovery #M-012189-01-0005) or Renilla luciferase as control (siGENOME, Horizon Discovery #P-002070-01-50). Three days after transfection, cells were trypsinized and seeded in six-well plates for experiments.

**Cell lysis, immunoblots and immunoprecipitations.** For immunoblot experiments, subconfluent cells in six-well plates were washed once with ice-cold DPBS and lysed in 100 μl lysis buffer (40 mM HEPES pH 7.4, 1% Triton X-100, 10 mM β-glycerophosphate, 10 mM sodium pyrophosphate, 30 mM NaF, 2× Roche Complete protease inhibitor cocktail with EDTA, 1× Roche PhosSTOP phosphatase inhibitors). Lysates were cleared by max speed centrifugation at 4 °C in a tabletop centrifuge and processed immediately for immunoblots or stored at −80 °C until needed. The same amount of cell lysate was loaded on multiple membranes, one for each protein of interest, including loading controls.

For immunoprecipitations, cells in 10-cm dishes were lysed in 1 ml EDTA-free lysis buffer supplemented with magnesium (40 mM HEPES pH 7.4, 5 mM $MgCl_2$, 1% triton X-100, 10 mM β-glycerophosphate, 10 mM sodium pyrophosphate, 30 mM NaF and 1× Roche Complete EDTA-free protease inhibitor cocktail). Lysates were cleared by max-speed centrifugation at 4 °C and processed immediately. Then, 2–3 mg protein per sample in 1 ml total volume was incubated with 10 μl (packed volume) pre-washed anti-FLAG M2 affinity gel beads (Sigma-Aldrich #A2220), 20 μl anti-HA magnetic beads (ThermoFisher #88837) or 10 μl anti-DYKDDDDK affinity resin beads (ThermoFisher #A36803) (for FLAG-S6K1 immunoprecipitations) for 2 h with gentle rotation. Immunoprecipitates were washed three times with lysis buffer supplemented with 500 mM or 200 mM NaCl (for FLAG–S6K1 immunoprecipitation) and then eluted in 25 μl 2× Laemmli.

Immunoblots were quantified using Image Lab (Biorad). The band intensity of each sample was normalized to the sum of all samples in one experiment, to allow comparisons between different experiments. In immunoprecipitation experiments, quantifications represent the ratio between co-immunoprecipitating and immunoprecipitated protein in the IP after normalization for the co-immunoprecipitating protein in the whole-cell lysate (normalized to α-tubulin).

**Lyso-IP.** Lyso-IP experiments were performed following previously described methods[69]. Briefly, $12 \times 10^6$ control or knockout HEK293T cells were seeded in 15-cm dishes and transiently transfected 24 h later with 6 μg of plasmid coding for TMEM192–3xHA. Two days after transfection, the medium was exchanged with fresh medium 1 h before cell collection. Cells were scraped in 1 ml ice-cold PBS containing protease and phosphatase inhibitors, centrifuged at 1,000g for 2 min at 4 °C in a tabletop centrifuge, resuspended in 950 μl PBS containing protease and phosphatase inhibitors, and gently homogenized with 20 strokes in a 5-ml douncer. Then, 25 μl cell suspension was transferred to a new tube before homogenization and lysed with 70 μl 1% Triton lysis buffer to generate whole-cell lysate samples. Intact cells were pelleted at 1,000g for 2 min, and the supernatant was incubated with 150 μl pre-washed anti-HA magnetic beads (ThermoFisher #88837) for 3 min with gentle rotation. Immunoprecipitated lysosomes were washed three times with PBS containing protease and phosphatase inhibitors and subsequently lysed in 100 μl 1% Triton lysis buffer.

**Immunofluorescence.** A total of $1.2 \times 10^5$ HEK293T cells were seeded on fibronectin-coated 12-mm glass coverslips. One day later, cells were washed once in DPBS, fixed for 15 min in 4% formaldehyde in PBS (ThermoFisher #28908), permeabilized for 10 min with 0.2% Triton X-100 in PBS, blocked 1 h with 0.25% BSA in PBS, incubated overnight with primary antibodies, incubated 1 h with

fluorophore-conjugated secondary antibodies, counterstained with 5 µg ml⁻¹ DAPI (Applichem #A1001) and then mounted on glass slides using Vectashield (Vector Labs #H-1000). Coverslips were imaged with a confocal microscope (Leica TCS SP8) using a 63× objective. A total of 16 random fields of view from two coverslips per condition were acquired. mTOR co-localization analysis was performed using CellProfiler. The mTOR staining was segmented using an Otsu-based adaptive thresholding algorithm to separate the bright spots of lysosomal mTOR from the cytosolic one. Similarly, the LAMP2 staining was segmented to identify LAMP2-positive objects. Finally, the percentage of the area of the LAMP2-positive objects covered by the lysosomal mTOR was calculated.

**RNA extraction and reverse-transcription qPCR.** Total RNA was extracted from cultured cells with TRIzol (Life Technologies #15596018) as per the manufacturer's instructions. Two micrograms of RNA per sample was reverse transcribed with Maxima H Minus reverse transcriptase (ThermoFisher #EP0753) and an oligo-dT primer. Expression of RagA, RagB, RagC and RagD was determined by qPCR with isoform-specific primers (Supplementary Table 1) using a SYBRGreen-based master mix (primaQUANT CYBR 2x qPCR SYBRGreen Master Mix with LOW ROX, Streinbrenner, #SL-9913). The absolute number of copies in each sample in Extended Data Fig. 1g was extrapolated from a standard curve created using a serial dilution of defined amounts of plasmids containing the coding sequences of each Rag GTPase. The relative abundance of RagA and RagB in Extended Data Fig. 10f,g was determined with the $2^{-\Delta\Delta Ct}$ method using Rpl13a as loading control.

**Purification of RagA/B GTPases from bacteria.** Dual expression pETDuet-1 plasmids coding for untagged RagA or RagB$^{short}$ and N-terminally His-tagged RagC$^{S75N}$ were transformed into Rosetta (Sigma-Aldrich #70953) competent cells. After overnight pre-culture, bacterial suspensions were diluted 1:100 in 1.6 L 2× YT medium supplemented with 2% glucose, carbenicillin (100 µg ml⁻¹) and chloramphenicol (34 µg ml⁻¹), further grown at 37 °C until reaching $OD_{600}$ 0.7, and then induced with 0.5 mM IPTG for 17 h at 37 °C. Bacteria were pelleted and resuspended in 120 ml lysis buffer (50 mM HEPES pH 7.4, 100 mM NaCl, 2 mM MgCl₂, 0.05% Triton X-100, 10 mM imidazole, 2 mM dithiothreitol (DTT), 0.5 mM PMSF, 100 µM ATP, 100 µM GDP, 100 µg ml⁻¹ lysozyme, 10 µg ml⁻¹ DNAse I, 1× Roche Complete protease inhibitor cocktail without EDTA and 1× FastBreak Cell Lysis Reagent (Promega)). The lysate was incubated with gentle shaking at room temperature for 1 h, followed by an additional hour at 4 °C, and then cleared by centrifugation at 23,000g for 30 min at 4 °C. All subsequent steps were performed at 4 °C, unless otherwise indicated. The cleared lysate was mixed with Ni-NTA agarose beads (400 µl slurry per 30 ml lysate), shaking for 1.5 h at 4 °C, followed by three washes with wash buffer (50 mM HEPES pH 7.4, 300 mM NaCl, 2 mM MgCl₂, 0.05% Triton X-100, 30 mM imidazole, 2 mM DTT and 100 µM ATP). The bound proteins were then eluted by incubating the beads with elution buffer (50 mM HEPES pH 7.4, 50 mM NaCl, 2 mM MgCl₂, 0.05% Triton X-100, 250 mM imidazole, 2 mM DTT and 100 µM ATP) for 30 min. The purity of the Rag heterodimers was refined with two additional purification steps. First, anion exchange was performed by applying the eluate from the previous step to a HiTrap Q HP (Sigma-Aldrich #GE17-1153-01) column with an isocratic flow pump at a rate of 1 ml min⁻¹, followed by washing with 5 column volumes of buffer A (50 mM HEPES pH 7.4, 50 mM NaCl, 2 mM MgCl₂, 0.05% Triton X-100, 2 mM DTT and 100 µM ATP) and elution in 15 column volumes of a linear salt gradient from buffer A to buffer B (50 mM HEPES pH 7.4, 500 mM NaCl, 2 mM MgCl₂, 0.05% Triton X-100, 2 mM DTT and 100 µM ATP). The fractions containing the Rag heterodimers were then pooled together, buffer-exchanged and concentrated to a 2-ml volume of buffer E (50 mM HEPES pH 7.4, 100 mM KOAc and 2 mM DTT) with 30 kDa MWCO PES concentrators, and finally loaded onto a HiPrep 16/60 Sephacryl S-200-HR (Sigma-Aldrich #GE17-1166-01) column for gel filtration. The fractions containing the Rag heterodimers were then pooled together, concentrated, snap-frozen after addition of glycerol (20% final concentration), aliquoted and stored at −80 °C. The final yield is approximately 1 mg Rag heterodimer per litre of bacterial culture.

**Purification of RagA/B GTPases and GATOR1 from HEK293T cells.** pcDNA3.1-based plasmids coding for untagged RagA, RagB$^{shortT54N}$ or RagB$^{long}$ (wild type or T54N) and FLAG-tagged RagC (S75N or Q120L) were co-transfected at a 1:1 ratio in RagABKO HEK293T cells previously seeded in 20 15-cm dishes using PEI. Forty-eight hours after transfection, cells were scraped in 60 ml ice-cold lysis buffer (40 mM HEPES pH 7.4, 100 mM NaCl, 5 mM MgCl₂, 1% Triton X-100, 100 µM ATP and 1× Roche Complete protease inhibitor cocktail without EDTA) and homogenized with 20 strokes of a Dounce homogenizer on ice. After clearing through centrifugation for 1 h at 23,000g at 4 °C, the lysate was incubated with 400 µl (packed volume) of anti-FLAG M2 affinity gel beads shaking for 3 h, followed by three washes with lysis buffer supplemented with 300 mM NaCl. After exchanging buffer to buffer E (50 mM HEPES pH 7.4, 100 mM KOAc and 2 mM DTT), the bound proteins were eluted overnight at 4 °C by adding 250 µg ml⁻¹ FLAG peptide (Sigma-Aldrich #F3290). The eluate was then concentrated in 30 kDa MWCO PES columns, supplemented with glycerol (20% final concentration), aliquoted, snap-frozen and stored at −80 °C. The GATOR1 complex was purified following a similar procedure. pRK5-based plasmids

coding for FLAG–DEPDC5 (wild type or Y775A), HA-Nprl2 and HA-Nprl3 were co-transfected at a 1:2:2 ratio, respectively, in control HEK293T cells. After immunoprecipitation as above, the anti-FLAG beads were buffer-exchanged to CHAPS buffer (50 mM HEPES pH 7.4, 150 mM NaCl, 2 mM MgCl₂ and 0.1% CHAPS) and the GATOR1 complex was eluted overnight at 4 °C by adding 250 µg ml⁻¹ FLAG peptide. The eluate was then concentrated in 100 kDa MWCO PES columns, supplemented with glycerol (10% final concentration) and 2 mM DTT, aliquoted, snap-frozen and stored at −80 °C.

**Malachite-green GTPase assay.** The Rag GTPase activity was assayed using a colourimetric malachite-green based reaction that detects the phosphate released upon GTP hydrolysis, according to previously published protocols[72]. Briefly, aliquots of the Rag dimers and GATOR1 were thawed on ice, diluted in dilution buffer (30 mM Tris–HCl pH 7.4, 0.1% CHAPS and 100 mM NaCl) to 5× the final concentrations in the assay, and then mixed together in 1× GTPase buffer (10 mM, 2 mM MgCl₂, 0.05% CHAPS, 10 mM NaCl, 100 µM DTT and 300 µM GTP) at the final concentrations of 1 µM for the Rags and 50 nM to 2 µM for GATOR1 in a 20 µl volume. For the experiments in Extended Data Fig. 7d,e, the Rag heterodimers were first immobilized on 50 µg magnetic beads (Dynabeads His-Tag, Life Technologies #10103D). The reaction was started by incubating the samples at 30 °C. After 3 h, the reaction was terminated with 5 µl 0.5 M EDTA. The colourimetric signal was generated by adding 75 µl of the malachite-green solution (1 mg ml⁻¹ malachite-green oxalate and 10 mg ml⁻¹ ammonium molybdate tetrahydrate, in 1 M HCl), incubated 5 min at room temperature, and then immediately read with a microplate reader (SPECTROstar Omega, BMG LABTECH) at 660 nm absorbance. Samples without Rags nor GATOR1, with Rags but no GATOR1, and with GATOR1 but no Rags were analysed in parallel to allow for background subtraction and to ensure that the GTPase activity detected was not due to contaminating GTPases. The amount of phosphate per sample was then calculated using a standard curve of NaH₂PO₄•H₂O in the range 0–50 µM. The amount of phosphate generated by GATOR1-stimulated GTP hydrolysis was then calculated by subtracting the background signal of the reaction with only the Rag heterodimers.

**In vitro GATOR1–Rag interaction.** Six micrograms of purified GATOR1 was bound on anti-HA magnetic beads (ThermoFisher #88837) for 2 h in CHAPS buffer (40 mM HEPES pH 7.4, 150 mM NaCl, 2 mM MgCl₂ and 0.1% CHAPS) followed by three washes with CHAPS buffer supplemented with 500 mM NaCl. Three micrograms of each Rag heterodimer was then added and incubated overnight with the GATOR1-containing beads in CHAPS buffer supplemented with 200 µM GDP. The same amount of Rag dimers was incubated in parallel with empty beads as background control. On the following day, the beads were washed three times with CHAPS buffer supplemented with 500 mM NaCl and the GATOR1–Rag complexes were eluted in 2× Laemmli buffer.

**GTP pull-down.** HEK293T cells were transiently transfected with plasmids coding for HA-tagged GTP-locked RagA/B isoforms and RagC$^{S75N}$. Forty-eight hours later, cells were lysed in 1 ml GTP-stripping buffer (40 mM HEPES pH 7.4, 20 mM EDTA, 0.3% CHAPS, 10 mM β-glycerophosphate, 10 mM sodium pyrophosphate, 30 mM NaF and 1× Roche Complete EDTA-free protease inhibitor cocktail) to remove all bound GTP. After clearing, lysates were passed through 10 kDa MWCO PES columns (GE Healthcare, #28-9322-96) and buffer-exchanged into EDTA-free, high-magnesium buffer (40 mM HEPES pH 7.4, 10 mM MgCl₂, 0.3% CHAPS, 10 mM β-glycerophosphate, 10 mM sodium pyrophosphate, 30 mM NaF and 1× Roche Complete EDTA-free protease inhibitor cocktail). In parallel, duplicate samples were passed through 10 kDa MWCO PES columns, but kept in GTP-stripping buffer, as negative control. Equal protein amounts (2–3 mg) from buffer-exchanged and negative-control lysates were then incubated with 10 µl (packed volume) pre-washed γ-amino-hexyl-GTP beads (Jena Bioscience #AC-117S) for 2 h with gentle rotation. GTP-bound proteins were then washed three times with wash buffer (40 mM HEPES pH 7.4, 1% Triton X-100, 500 mM NaCl, 10 mM β-glycerophosphate, 10 mM sodium pyrophosphate, 30 mM NaF and 1× Roche Complete EDTA-free protease inhibitor cocktail) containing either 5 mM MgCl₂ or 20 mM EDTA and then eluted in 30 µl 2× Laemmli.

**Extraction of mouse tissues and preparation of lysates, isolation of fibroblasts (MEFs) and primary cortical neurons from mouse embryos and amino-acid treatments.** Mice were housed in pathogen-free conditions, 12-h day/night cycle, at 21 °C temperature and 50–60% relative humidity. Food and water were available ad libitum. For mouse tissue analysis, C57B6/J control female postnatal-day 60–90 adult mice were killed with isofluorane, and the tissues of interest were dissected, snap-frozen in liquid nitrogen and stored at −80 °C until needed. To prepare tissue lysates, the frozen tissues were ground on dry ice in Eppendorf tubes using plastic pestles and lysed by adding 500 µl lysis buffer (40 mM HEPES pH 7.4, 1% Triton X-100, 10 mM β-glycerophosphate, 10 mM sodium pyrophosphate, 30 mM NaF, 2× Roche Complete protease inhibitor cocktail with EDTA and 1× Roche PhosSTOP phosphatase inhibitors) and then processed as described above for cell lysates. MEFs were prepared from six to eight embryonic-day 15 mouse embryos from time-mated C57B6/J mice. After removal of the heads,

from which neurons were prepared in parallel, and visceral organs, the embryos were finely minced with a scalpel and then digested in 15 ml trypsin–EDTA 0.25% containing 130 mg ml⁻¹ DNAse I (Roche #10104159001) in a water bath at 37 °C. Trypsin was inactivated by addition of 20 ml DMEM containing 10% FBS, and the cell suspension was then distributed in two 10-cm dishes. Twenty-four hours later, cells were trypsinized and re-seeded at a 1:10 ratio. To prepare the primary cortical neurons, the cortices were dissected, trypsinized and physically dissociated. After washing in HBSS, $5 \times 10^5$ cells were plated onto poly-L-lysine-coated 3.5-cm dishes (Life Technologies #21430-012) supplemented with 2 mM L-glutamine (Life Technologies #25030-024), 0.6% glucose and 10% heat-inactivated horse serum, and kept in 5% $CO_2$ at 36.5 °C. After 4 h, the medium was exchanged to MEM supplemented with 1 mM sodium pyruvate (Sigma-Aldrich #P2256), 2 mM L-glutamine (Life Technologies #25030-024), 0.6% glucose, 0.22% sodium bicarbonate, 0.1% egg albumin (Sigma-Aldrich #A5503), N2 supplement (Life Technologies #17502048) and B27 supplement (Life Technologies #17504044). Twenty-four hours later, the medium was replaced with fresh medium and cells were further cultured until DIV 5. For amino-acid treatments of primary embryonic neurons, media were prepared using homemade, amino-acid-free DMEM with 1 mM sodium pyruvate (Life Technologies #11360070), 1× N2 supplement (Life Technologies #17502048) and 1× B27 supplement (Life Technologies #17504044), without the addition of serum, and supplemented with 1× MEM non-essential amino acids, 1× MEM amino acids and 2 mM L-glutamine for the +aa condition. For amino-acid treatments of MEFs, homemade, amino-acid free DMEM was completed completed by addition of 10% dFBS (Life Technologies #A33820-01) (−aa medium), and supplemented with 1× MEM non-essential amino acids, 1× MEM amino acids and 2 mM L-glutamine (+aa medium).

**Production of rAAV.** Serotype 1/2 rAAV particles were produced and purified as described previously[73,74].

**shRNA-mediated knockdown and amino-acid treatments in primary forebrain cultures from post-natal mice.** Primary dissociated hippocampal and cortical cultures from postnatal-day 0 C57BL/6NCrl mice (Charles River Laboratories) of either sex were prepared and maintained in 12- or 24-well plates according to established protocols[46,47]. Briefly, cells were plated at a density of $1 \times 10^5$ cells cm⁻² and grown until DIV 8 in Neurobasal-A medium (Life Technologies #10888022) supplemented with B27, 0.5 mM L-glutamine, 1% rat serum (Biowest #S2150) and 50 U ml⁻¹ penicillin–streptomycin. Cytosine arabinoside (Sigma-Aldrich #C1768; 2.8 µM) was added on DIV 3 to prevent proliferation of glial cells. For knockdown experiments, cells were infected ≥6 h later with $1 \times 10^{10}$ viral particles ml⁻¹ of serotype 1/2 rAAVs driving the expression of shRNAs targeting RagA or RagB, or mCherry as a control (Supplementary Table 1). A medium change was performed on DIV 8 to standard medium consisting of a 9:1 mixture of buffered saline solution (10 mM HEPES, pH 7.4, 114 mM NaCl, 26.1 mM NaHCO₃, 5.3 mM KCl, 1 mM MgCl₂, 2 mM CaCl₂, 30 mM glucose, 1 mM glycine, 0.5 mM C₃H₃NaO₃ and 0.001% phenol red) and MEM (Life Technologies #21090), supplemented with insulin (7.5 µg ml⁻¹), transferrin (7.5 µg ml⁻¹) and sodium selenite (7.5 ng ml⁻¹) (ITS Liquid Media Supplement, Sigma-Aldrich #I3146), 50 U ml⁻¹ penicillin–streptomycin and 0.9× MEM amino acids (Sigma-Aldrich #M5550) to yield a 100% amino-acid concentration. On DIV 9, cells were treated with standard medium containing homemade, amino-acid-free MEM (according to Life Technologies #21090) and supplemented with 1× or 0.1× MEM amino acids for the 100% or 10% amino-acid concentrations, respectively, or not supplemented with amino acids for the 0% amino-acid concentration. Where indicated, the GABA_A receptor antagonist bicuculline (ENZO Life Sciences #ALX-550-515; 50 µl) was added to induce action potential bursting. Then, 22–26 h later, cells were washed once with PBS and lysed in 1% Triton lysis buffer (40 mM HEPES pH 7.4, 1% Triton X-100, 10 mM β-glycerophosphate, 10 mM sodium pyrophosphate, 30 mM NaF, 2× Roche Complete protease inhibitor cocktail with EDTA and 1× Roche PhosSTOP phosphatase inhibitors).

**Morphometric analysis of hippocampal neurons.** Hippocampal neurons were transfected with plasmids coding for shRNAs against RagA or RagB, or mCherry as a control, on DIV 8 using Lipofectamine 2000, as described[75]. All constructs carry an additional expression cassette for GFP which allows the monitoring and assessment of neuronal structure. After transfection, neurons were maintained at 37 °C and 5% $CO_2$ in standard medium consisting of a 9:1 mixture of buffered saline solution and MEM, as described above, and placed in the IncuCyte Live-Cell Analysis System to monitor neurite outgrowth using a 20× objective to acquire images of the entire surface of the well every 6 h. Twenty-four hours after transfection, neurons were treated with 50 µM bicuculline to induce activity-dependent dendritic remodelling and imaged for 48 h. Neurite length of GFP-expressing neurons was analysed using IncuCyte NeuroTrack Software Module. The neurite length values were plotted over time, and the area under the curve was computed to extrapolate one value per experimental condition. Values derived from bicuculline-treated samples were then normalized to the respective control-treated samples to yield activity-dependent dendritogenesis.

**OPP incorporation assay.** A total of $5 \times 10^5$ EFO21 cells per well were seeded in six-well plates 3 days after siRNA transfection. The following day, cells were washed once with DPBS and incubated with DMEM supplemented with 10% dFBS and containing either 1× MEM non-essential amino acids, 1× MEM amino acids and 2 mM L-glutamine (100% amino acids) or four times less amino acids (25% amino acids) for 4.5 h. Then, 20 µM OPP reagent (Jena Bioscience #NU-931-05) was added for 30 min. Cells were subsequently washed with DPBS, trypsinized and fixed with ice-cold 70% ethanol for 30 min at −20 °C, followed by three washes in PBS containing 0.5% Tween-20. The incorporated OPP was then labelled with the Alexa488 Fluor Picolyl azide using the Click-iT Plus OPP Protein synthesis assay kit (Life Technologies #C10456), as per the manufacturer's instructions. Samples were run in a Guava easyCyte HT flow cytometer (Millipore) and analysed using FlowJo (v10). The cell population of interest was identified plotting FSC-H versus SSC-H, singlets gated by plotting FSC-H versus FSC-A, and the mean intensity of the Alexa488 signal within the singlets population was used to quantify the extent of OPP incorporation.

**Statistics and reproducibility.** Statistical analysis was performed using GraphPad Prism (9.0.2) and Microsoft Excel (16.16.27). GTEX and TCGA datasets were analysed using the Xena browser[76]. Two-tailed, unpaired *t*-test or one-way or two-way analysis of variance (ANOVA) followed by post-hoc tests were used to determine significance as indicated in the figure legends. In time-course or titration experiments, curves were fit with non-linear regression analysis following a two-phase decay, three-phase decay, sigmoidal or hyperbola model. Data were assumed to be homoschedastic and normally distributed, although this was not formally tested. No statistical method was used to pre-determine sample size. Sample sizes were chosen on the basis of standard practice in the field. All experiments were repeated at least twice with independent sets of biological samples. Quantification and statistics were derived from $n = 3$ independent experiments, unless specified in the legends. No data were excluded from the analyses. The experiments were not randomized. The Investigators were not blinded to allocation during experiments and outcome assessment.

**Reporting summary.** Further information on research design is available in the Nature Research Reporting Summary linked to this article.

## Data availability

Data on RagA/B expression in brain cell types were extracted from the RNA-seq dataset published by the Barres group[39] (GEO accession number GSE52564). Data from the GTEX and TCGA databases used in Fig. 6 and Extended Data Fig. 10 were accessed through the Xena browser[76] (xenabrowser.net). All other data supporting the findings of this study are available from the corresponding author on reasonable request. Source data are provided with this paper.

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

## Acknowledgements

We are grateful to H. Bading, Department of Neurobiology and Interdisciplinary Center for Neurosciences, Heidelberg University, for providing the opportunity to carry out in his laboratory the RagA/RagB knockdown experiments in mouse cortical neurons. This work was funded in part via a DFG SFB873 grant to A.M.-V. G.F. was supported by an EMBO Long-Term Fellowship (ALTF 755-2018).

## Author contributions

G.F., S.M., S.K., A.M.H, M.R., D.C.I., J.-P.Q., N.t.B. and D.M. performed experiments. G.F., S.M., S.K., A.M.H, M.R., D.C.I., J.-P.Q., D.M., A.M.-V. and A.A.T. designed experiments and analysed data. G.F., A.M.-V. and A.A.T. wrote the manuscript.

## Funding

## Competing interests

The authors declare no competing interests.

## Additional information

**Extended data** is available for this paper at https://doi.org/10.1038/s41556-022-00977-x.

**Correspondence and requests for materials** should be addressed to Aurelio A. Teleman.

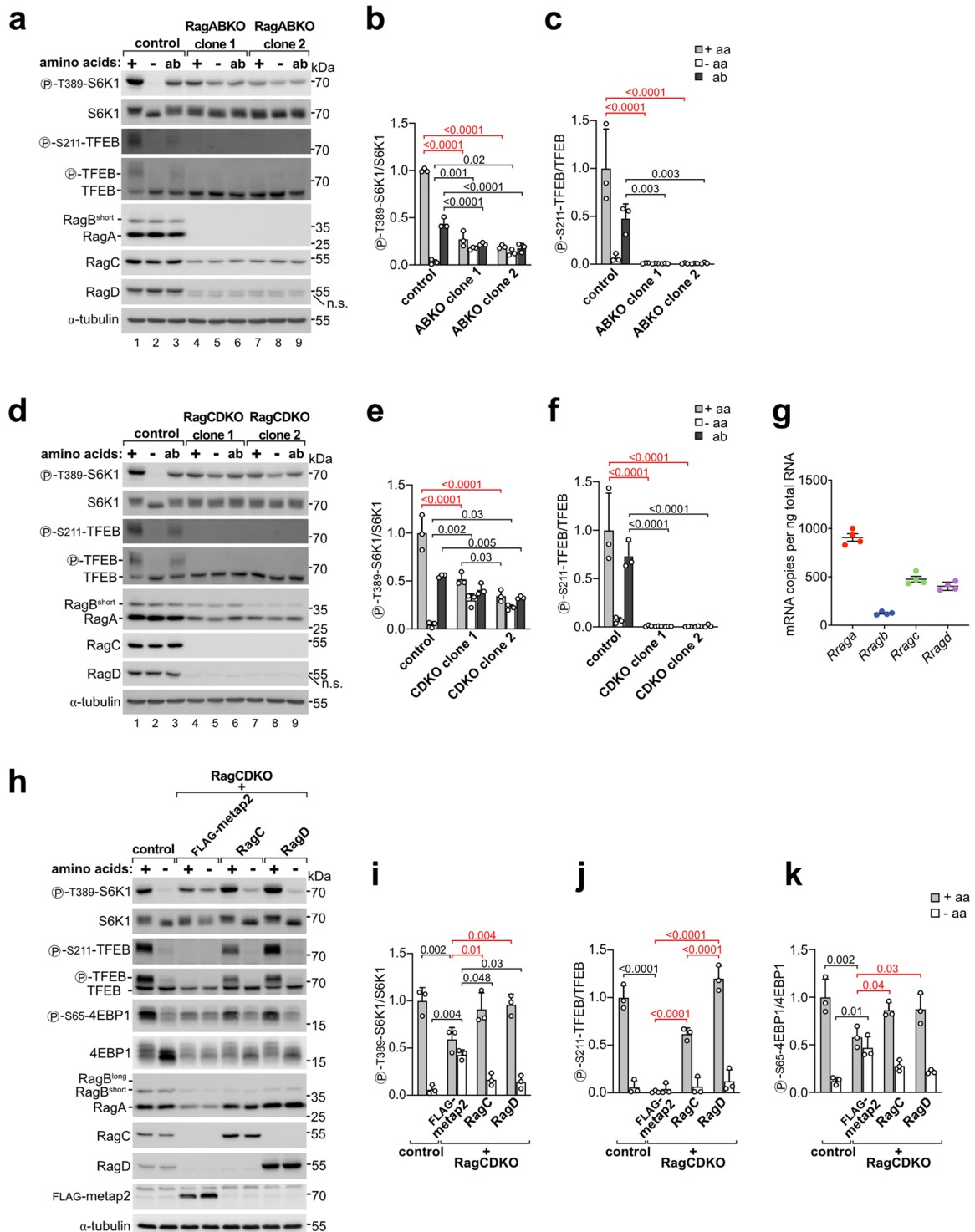

**Extended Data Fig. 1 | mTORC1 activity in RagABKO and RagCDKO cells. (a-f)** S6K1 and TFEB phosphorylation in control and RagA/B-double knockout (a-c) or RagC/D-double knockout (d-f) HEK293T cells in nutrient rich conditions, after amino-acid starvation for 1h, or after starvation for 1h followed by amino-acid addback for 15 min ('ab'). n.s.=non-specific band. (a,d) Representative example. (b-c,e-f) Quantification of 3 independent experiments, with unstarved control cells set to 1. Bar Height=average, error bars=standard deviation, n=3 biological replicates. Two-way ANOVA and Sidak's post-hoc test. **(g)** Absolute quantification of Rag levels by qPCR in HEK2983T cells. Line=average, error bars=standard deviation, n=4 biological replicates. **(h-k)** S6K1, TFEB, and 4EBP1 phosphorylation in control or RagCDKO cells stably transfected with a control protein (FLAG-metap2) or with the indicated Rag isoforms. Cells were incubated in amino-acid rich medium ('+') or starved of amino acids for 30 minutes ('−'). (h) Representative example. (i-k) Quantification of 3 independent experiments, with unstarved control cells set to 1. Bar Height=average, error bars=standard deviation, n=3 biological replicates. Two-way ANOVA and Sidak's post-hoc test. -aa: amino-acid free DMEM+10% dFBS. +aa: -aa medium supplemented with 1x amino acids. Exact p values are shown in the graphs. Source numerical data and unprocessed blots are available in source data.

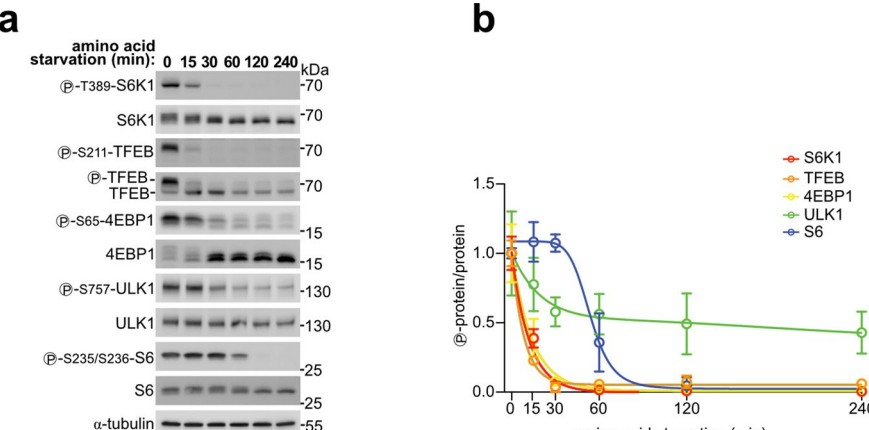

**Extended Data Fig. 2 | Response of mTORC1 targets to amino-acid removal. (a, b)** Phosphorylation of the mTORC1 targets S6K1, TFEB, 4EBP1, ULK1, and the S6K targets S6 in control cells upon amino-acid starvation (amino-acid free DMEM + 10% dFBS) for the indicated time points. (a) Representative example. (b) Quantification of 3 independent experiments, with unstarved cells set to 1. Circle=average, error bars=standard deviation, n = 3 biological replicates. Source numerical data and unprocessed blots are available in source data.

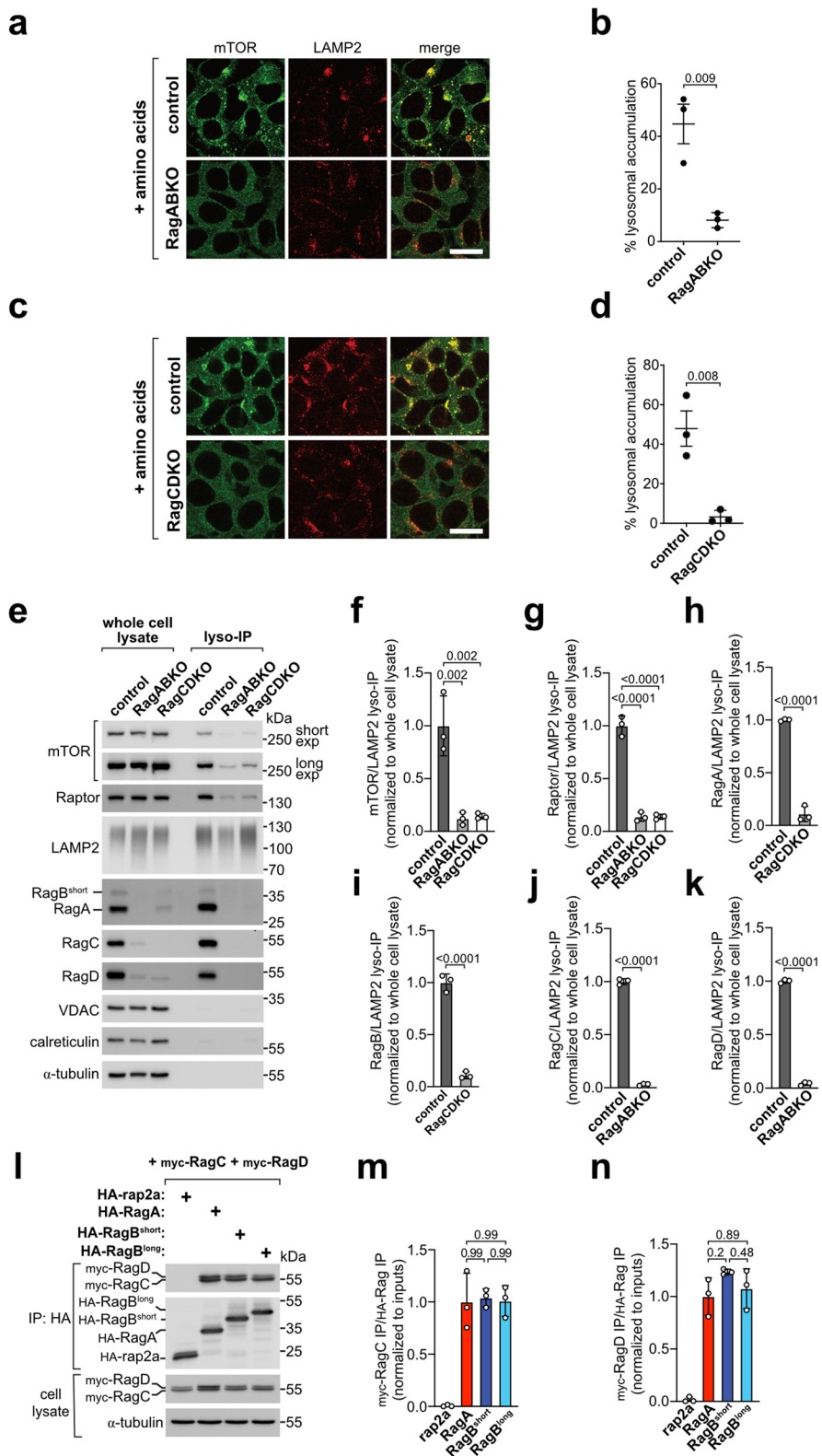

**Extended Data Fig. 3 | See next page for caption.**

**Extended Data Fig. 3 | mTOR localization in RagABKO and RagCDKO cells. (a-d)** mTOR and LAMP2 immunofluorescence in control and RagABKO or RagCDKO cells in amino-acid rich conditions. (a,c) Representative examples. (b-d) Quantification of 3 independent experiments as percentage of LAMP2 signal overlapping with the mTOR signal. Each data point represents the average of multiple fields of view of one replicate experiment. Line=average, error bars=standard deviation, n = 3 replicates. Two-tailed, unpaired t-test. Scale bar: 20 µm. **(e-k)** Lysosomal immunopurification (lyso-IP) from control, RagABKO, and RagCDKO cells in amino-acid replete conditions. LAMP2: lysosomal marker. Markers of other organelles are shown as control: VDAC (mitochondria), calreticulin (ER), α-tubulin (cytosol). (e) Representative example. (f-k) Quantification of 3 independent experiments, with control cells set to 1. Bar Height=average, error bars=standard deviation, n = 3 biological replicates. One-way ANOVA and Tukey's post-hoc test (f-g) or two-tailed, unpaired t-test (h-k). **(l-n)** CoIP of HA-tagged RagA, RagB[short], or RagB[long] with myc-tagged RagC and RagD. (l) Representative example. (m-n) Quantification of 3 independent experiments, with the HA-RagA condition set to 1. Bar Height=average, error bars=standard deviation, n = 3 biological replicates. One-way ANOVA and Tukey's post-hoc test. Exact p values are shown in the graphs. Source numerical data and unprocessed blots are available in source data.

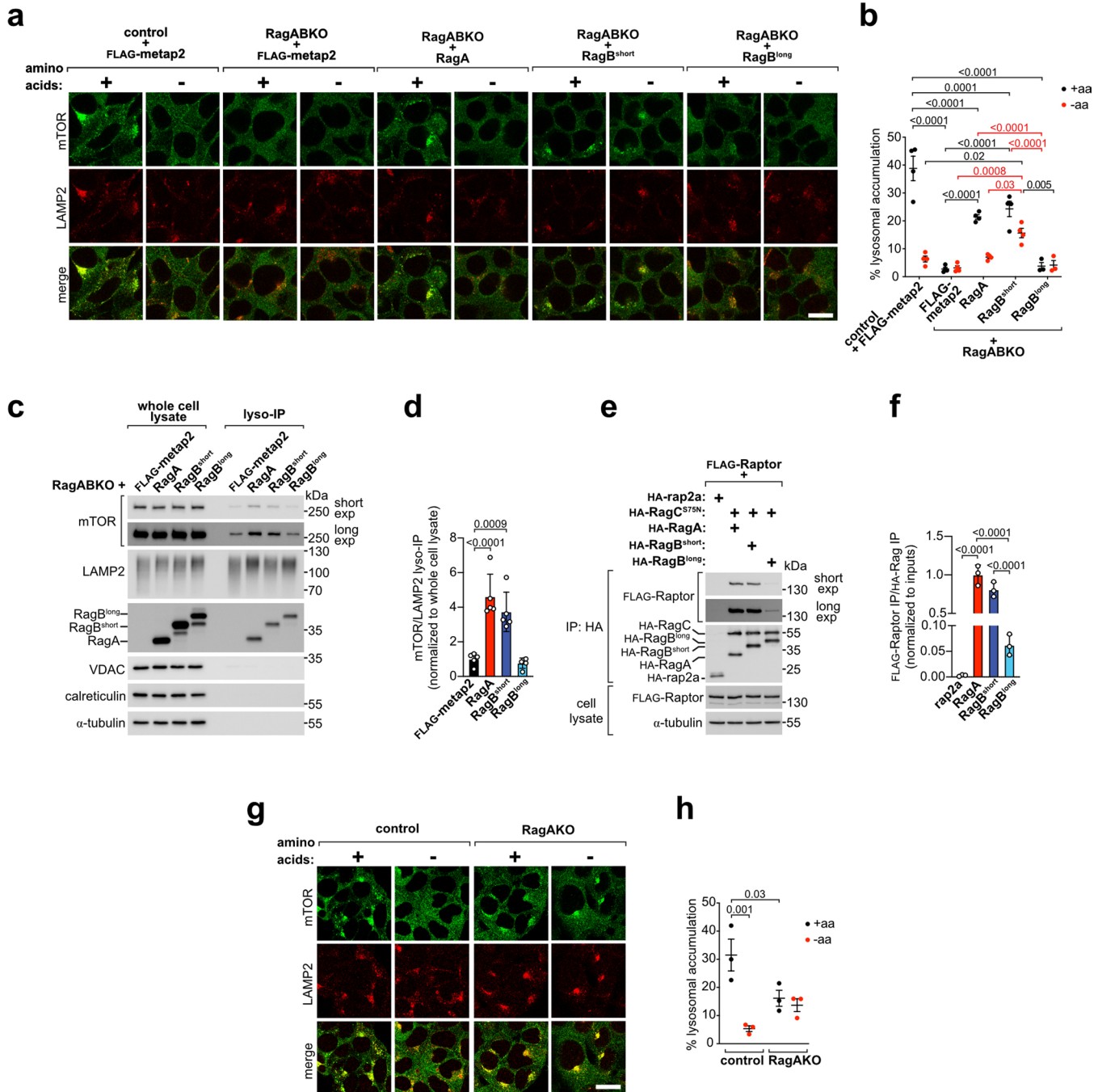

**Extended Data Fig. 4 | mTOR persists on lysosomes during starvation in cells expressing only Rag. (a, b)** mTOR and LAMP2 immunofluorescence in control (FLAG-metap2) or RagABKO cells stably transfected with RagA/B paralogues and incubated in amino-acid rich medium or starved of amino acids for 30 min. (a) Representative example. (b) Quantification of 3 independent experiments with all genotypes and 1 additional experiment with all genotypes except RagABKO + RagB^long. Data are expressed as percentage of LAMP2 signal overlapping with the mTOR signal, each data point represents the average of multiple fields of view from one independent experiment. Line=average, error bars=standard deviation, n = 3-4 replicates. Two-way ANOVA and Sidak's post-hoc test. Scale bar=20 μm. **(c, d)** Lysosomal immunopurification (lyso-IP) from RagABKO cells stably transfected with FLAG-metap2 or the indicated RagA/B paralogues in amino-acid replete conditions. LAMP2: lysosomal marker. Markers of other organelles are shown as control: VDAC (mitochondria), calreticulin (ER), α-tubulin (cytosol). (c) Representative example. (d) Quantification of 5 independent experiments, with FLAG-metap2 cells set to 1. Bar Height=average, error bars=standard deviation, n = 5 biological replicates. One-way ANOVA and Tukey's post-hoc test. **(e, f)** coIP of Raptor with wild-type-RagA/B•RagC^S75N dimers. (e) Representative example. (f) Quantification of 3 independent experiments, with the RagA condition set to 1. Bar Height=average, error bars=standard deviation, n = 3 biological replicates. One-way ANOVA and Tukey's post-hoc test. **(g, h)** mTOR and LAMP2 immunofluorescence in control or RagAKO cells. Cells were incubated in amino-acid rich medium or starved of amino acids for 30 min. (g) Representative example. (h) Quantification of 3 independent experiments. Data are expressed as percentage of LAMP2 signal overlapping with the mTOR signal, each data point represents the average of multiple fields of view from one replicate experiment. Line=average, error bars=standard deviation, n = 3 biological replicates. Two-way ANOVA and Sidak's post-hoc test. Scale bar=20 μm. -aa: amino-acid free DMEM+10% dFBS. +aa: -aa medium supplemented with 1x amino acids. Exact p values are shown in the graphs. Source numerical data and unprocessed blots are available in source data.

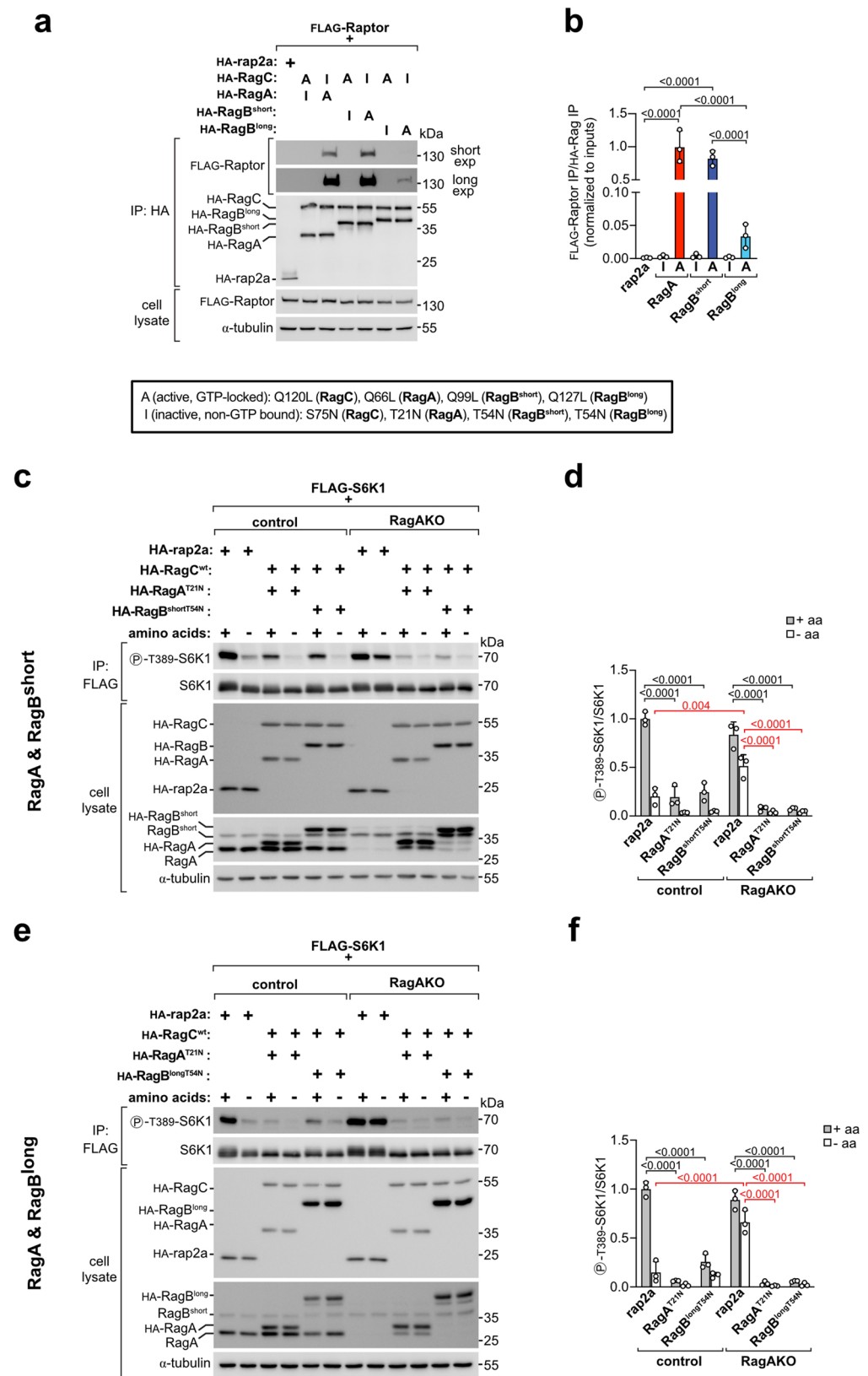

Extended Data Fig. 5 | See next page for caption.

**Extended Data Fig. 5 | Inactive RagA/B rescue persistent mTORC1 activity in RagAKO cells. (a, b)** coIP of Raptor with Rag dimers harboring the indicated mutations. (a) Representative example. (b) Quantification of 3 independent experiments, with the active mutant of RagA set to 1. Bar Height=average, error bars=standard deviation, n = 3 biological replicates. One-way ANOVA and Tukey's post-hoc test. **(c-f)** Persistent mTORC1 activity in RagAKO cells during starvation is rescued by transient transfection of non-GTP bound RagA, RagB<sup>short</sup> (c-d), or RagB<sup>long</sup> (e-f). Cells were treated with amino-acid rich medium or starved of amino acids for 30 min before lysis. (c,e) Representative examples. (d,f) Quantification of 3 independent experiments, with unstarved HA-rap2a-transfected control cells set to 1. Bar Height=average, error bars=standard deviation, n = 3 biological replicates. Two-way ANOVA and Sidak's post-hoc test. -aa: amino-acid free DMEM + 10% dFBS. +aa: -aa medium supplemented with 1x amino acids. Exact p values are shown in the graphs. Source numerical data and unprocessed blots are available in source data.

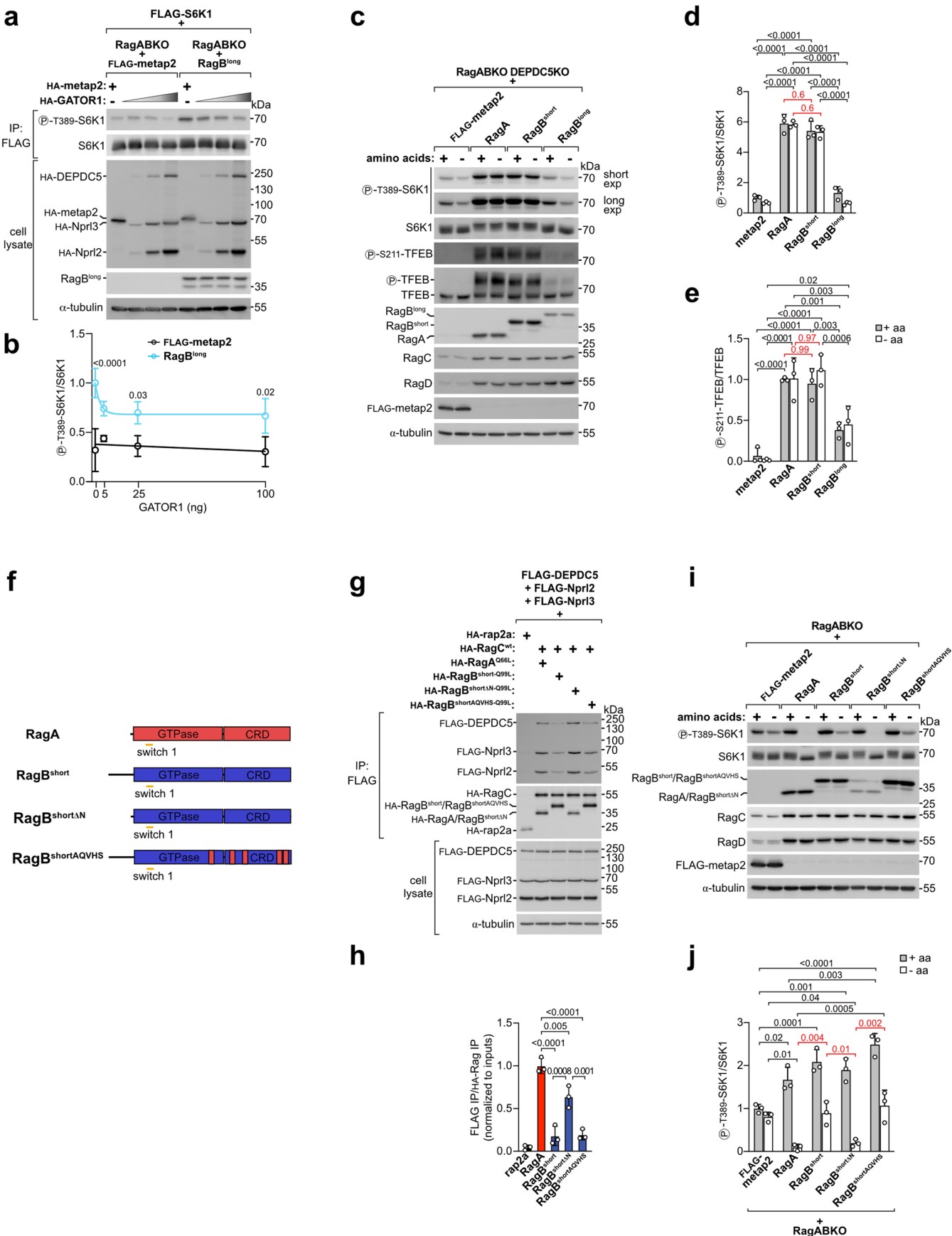

**Extended Data Fig. 6 | See next page for caption.**

**Extended Data Fig. 6 | Mechanistic determinants of RagA vs RagB<sup>short</sup> functional differences. (a, b)** RagABKO cells expressing RagB<sup>long</sup> or a control protein (FLAG-metap2) were transiently transfected with increasing amounts of GATOR1 plasmids (5 ng, 25 ng, or 100 ng of each GATOR1 subunit) or metap2 (100 ng) as negative control. (a) Representative example. (b) Quantification of 3 independent experiments, with RagB<sup>long</sup>-expressing cells transfected with metap2 set to 1. Circle=average, error bars=standard deviation, n = 4 biological replicates. Two-way ANOVA and Sidak's post-hoc test. **(c-e)** mTORC1 activity is comparable between RagABKO/DEPDC5KO stably expressing RagA or RagB<sup>short</sup> and treated with amino-acid rich or amino-acid free medium for 30 min. (c) Representative example. (d-e) Quantification of 3 independent experiments, with unstarved control (FLAG-metap2) cells set to 1. Bar Height=average, error bars=standard deviation, n = 3 biological replicates. Two-way ANOVA and Sidak's post-hoc test. **(f)** Scheme of the RagA/B constructs used. RagB<sup>shortΔN</sup>: deletion of amino acids 2 − 34; RagB<sup>shortAQVHS</sup>: substitution of 5 amino acids in the RagB<sup>short</sup> sequence with the corresponding amino acids in the RagA sequence (S191A, E229Q, A258V, Q341H, C342S). **(g, h)** coIP of GATOR1 with the RagA, RagB<sup>short</sup>, or the RagB<sup>short</sup> mutants described in (f). (g) Representative example. (h) Quantification of 3 independent experiments, with the RagA condition set to 1. Bar Height=average, error bars=standard deviation, n = 3 biological replicates. One-way ANOVA and Tukey's post-hoc test. **(i-j)** S6K1 phosphorylation in RagABKO cells stably expressing a control protein (FLAG-metap2) or RagA, RagB<sup>short</sup>, or the two RagB<sup>short</sup> mutants described in (f) and treated with amino-acid rich or amino-acid free medium for 30 min before lysis. (i) Representative example. (j) Quantification of 3 independent experiments, with unstarved control (FLAG-metap2) cells set to 1. Bar Height=average, error bars=standard deviation, n = 3 biological replicates. Two-way ANOVA and Sidak's post-hoc test. -aa: amino-acid free DMEM + 10% dFBS. +aa: -aa medium supplemented with 1x amino acids. Exact p values are shown in the graphs. Source numerical data and unprocessed blots are available in source data.

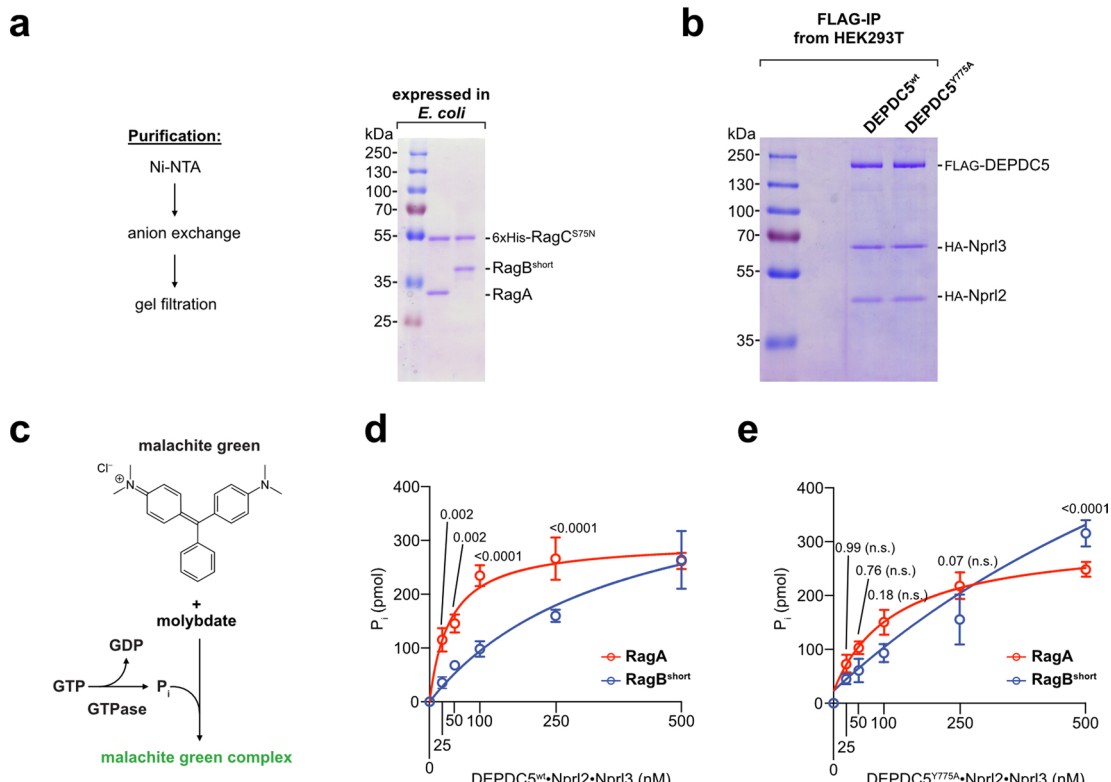

**Extended Data Fig. 7 |** *in vitro* **GTPase assays with purified RagA/B proteins. (a)** Purification strategy and Coomassie staining of RagA and RagB[short] purified as dimers with His-tagged RagC[S75N] (1 µg dimer per lane) from *E. coli*. The experiment was repeated once. **(b)** Coomassie staining of the GATOR1 complex purified from HEK293T cells as trimer of wild-type or mutant (Y775A) FLAG-DEDPC5 and HA-tagged Nprl2/3 (1 µg complex per lane). The experiment was repeated once. **(c)** Scheme depicting the malachite-green based reaction used to assay the Rag GTPase activity. The inorganic phosphate released upon hydrolysis of GTP forms a complex with molybdate and malachite green that causes the malachite green to change from yellow to blue-green. **(d, e)** GTPase activity of RagA/B dimerized to RagC[S75N] (1 µM) and mixed with the indicated concentrations of GATOR1 containing wild-type (d) or Y775A-mutant (e) DEPDC5 after immobilization of RagA/B on His-tag beads. Data from three (d) or four (e) independent experiments were plotted and a curve was fit with non-linear regression analysis following a hyperbola model. Circle=average, error bars=standard deviation, n = 3 (d) or 4 (e) replicates. Two-way ANOVA and Sidak's post-hoc test, p values are indicated above the circles. n.s., not significant. Source numerical data are available in source data.

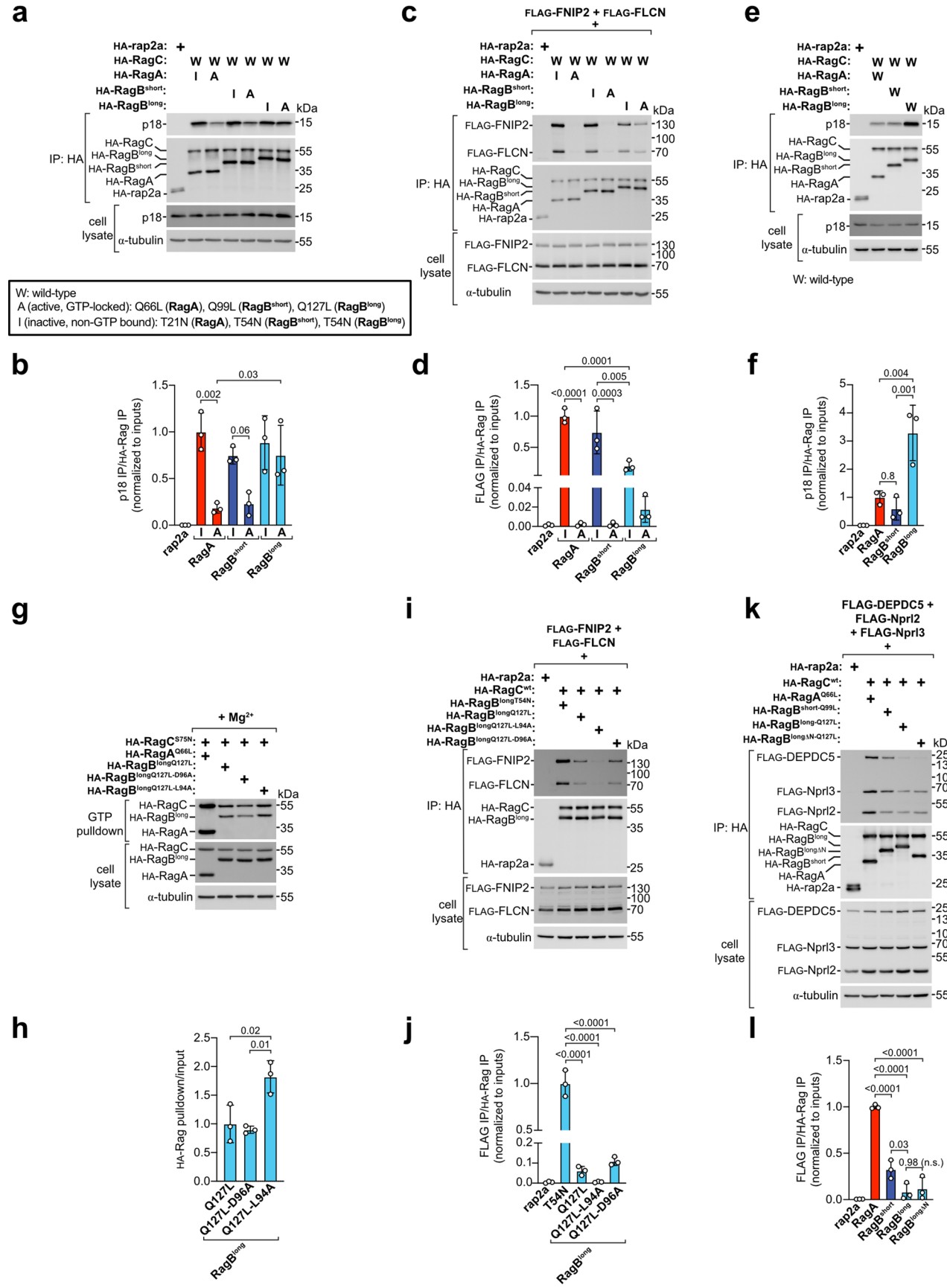

**Extended Data Fig. 8 | See next page for caption.**

**Extended Data Fig. 8 | RagB<sup>long</sup> interactions are consistent with low GTP binding. (a**, **b)** coIP of endogenous p18 with Rag dimers harboring the indicated mutations. (a) Representative example. (b) Quantification of 3 independent experiments, with the inactive mutant of RagA set to 1. Bar Height=average, error bars=standard deviation, n = 3 biological replicates. One-way ANOVA and Tukey's post-hoc test. **(c, d)** coIP of the folliculin complex (FLCN, FNIP2) with Rag dimers harboring the indicated mutations. (c) Representative example. (d) Quantification of 3 independent experiments, with the inactive mutant of RagA set to 1. Bar Height=average, error bars=standard deviation, n = 3 biological replicates. One-way ANOVA and Tukey's post-hoc test. **(e, f)** coIP of endogenous p18 with wild-type RagA/B isoforms. (e) Representative example. (f) Quantification of 3 independent experiments, with the RagA condition set to 1. Bar Height=average, error bars=standard deviation, n = 3 biological replicates. One-way ANOVA and Tukey's post-hoc test. **(g, h)** GTP-pull down assay comparing RagA, RagB<sup>long</sup>, and two mutants of RagB<sup>long</sup>. (g) Representative example. (h) Quantification of 3 independent experiments, with RagB<sup>Q127L</sup> set to 1. Bar Height=average, error bars=standard deviation, n = 3 biological replicates. One-way ANOVA and Tukey's post-hoc test. **(i, j)** coIP of the folliculin complex (FLCN, FNIP2) with RagB<sup>long</sup> mutants. (i) Representative example. (j) Quantification of 3 independent experiments, with the RagB<sup>longT54N</sup> condition set to 1. Bar Height=average, error bars=standard deviation, n = 3 biological replicates. One-way ANOVA and Tukey's post-hoc test. **(k)** coIP of GATOR1 with RagA, RagB<sup>short</sup>, RagB<sup>long</sup>, or a RagB<sup>long</sup> mutant lacking the N-terminal extension (RagB<sup>longΔN</sup>). (k) Representative example. (l) Quantification of 3 independent experiments, with the RagA condition set to 1. Bar Height=average, error bars=standard deviation, n = 3 biological replicates. One-way ANOVA and Tukey's post-hoc test. Exact p values are shown in the graphs. n.s., not significant. Source numerical data and unprocessed blots are available in source data.

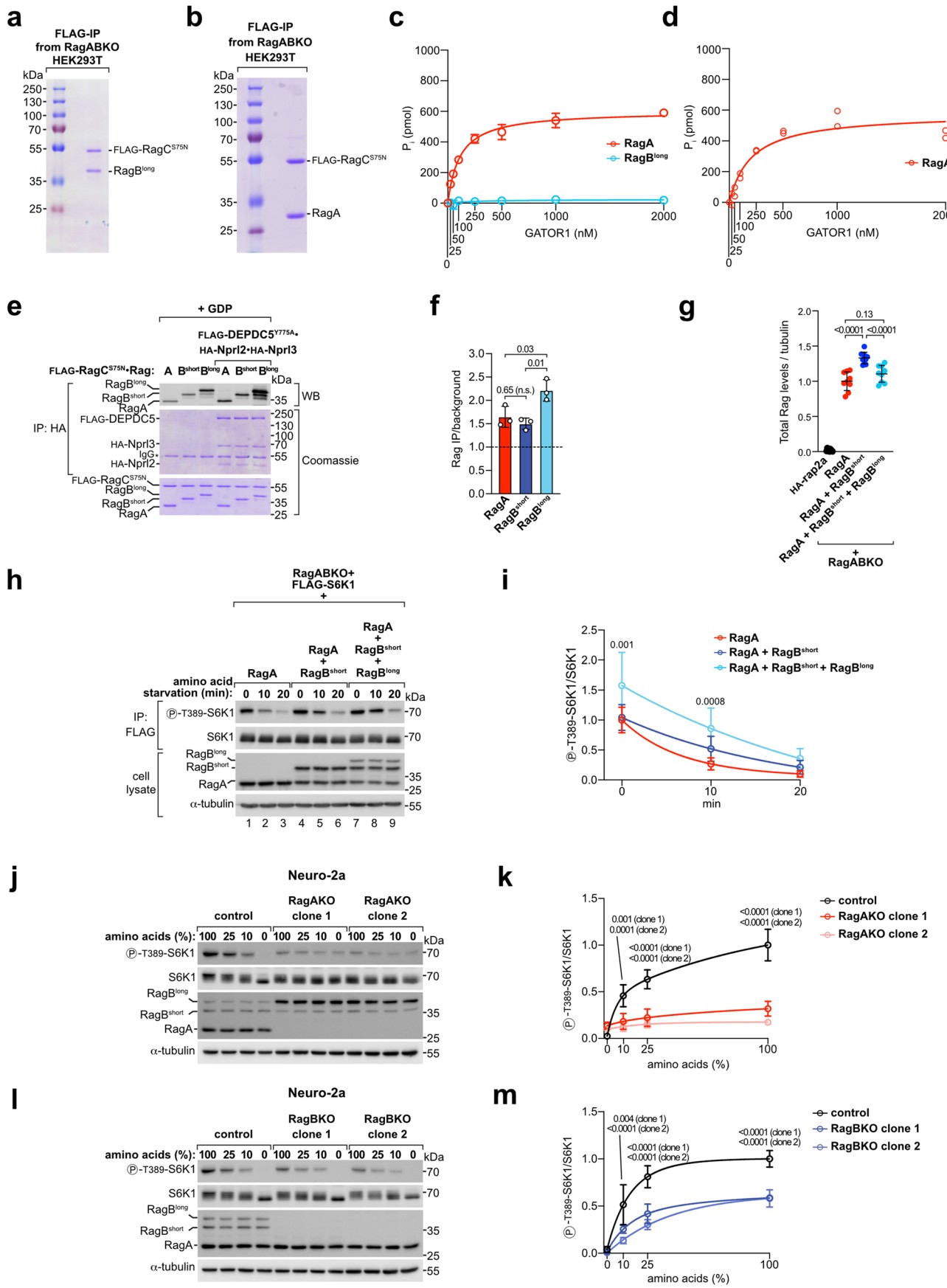

**Extended Data Fig. 9 | See next page for caption.**

**Extended Data Fig. 9 | RagB^long acts as a 'sponge' for the GAP interface of GATOR1. (a, b)** Coomassie staining of RagB^long (a) or RagA (b) purified from RagABKO HEK293T cells as dimers with FLAG-tagged RagC^S75N (1 μg dimer per lane). The experiment was repeated once. **(c, d)** GTPase activity of RagA or RagB^long proteins dimerized to RagC^S75N (1 μM) and mixed in solution with the indicated concentrations of GATOR1. RagB^long was purified from mammalian cells and RagA was purified from either (c) bacteria or (d) mammalian cells. Data from three (c) or two (d) independent experiments were plotted and a curve was fit with non-linear regression analysis following a hyperbola model. Circle=average (c) or individual replicates (d), error bars=standard deviation. **(e, f)** *in vitro* interaction of purified Rag dimers with GATOR1 (containing DEPDC5^Y775A) in the presence of GDP. (e) Representative example. (f) Quantification of 3 independent experiments normalized for the background in the empty beads control. Bar Height=average, error bars=standard deviation, n=3 replicates. One-way ANOVA and Tukey's post-hoc test. **(g)** Quantification of the total RagA/B protein levels in Fig. 5a-i, with the RagA condition set to 1. Line=average, error bars=standard deviation, n=9 biological replicates. One-way ANOVA and Tukey's post-hoc test. **(h-i)** S6K1 phosphorylation in RagABKO cells transiently transfected with RagA, RagA and RagB^short, or all three RagA/B isoforms during starvation (amino-acid free DMEM+10% dFBS) for the indicated time points. (h) Representative example. (i) Quantification of six independent experiments, with RagA-expressing cells at time point 0 set to 1. Circle=average, error bars=standard deviation, n=6 biological replicates. Two-way ANOVA and Tukey's post-hoc test. **(j-m)** Amino-acid titration in Neuro-2a cells knockout for RagA (j-k) or RagB (l-m) treated with the indicated amino-acid concentrations for one hour. (j,l) Representative examples. (k.m) Quantifications of 3 independent experiments, with unstarved control cells set to 1. Circle=average, error bars=standard deviation, n=3 biological replicates. Two-way ANOVA and Tukey's post-hoc test. Exact p values are shown in the graphs. n.s., not significant. Source numerical data and unprocessed blots are available in source data.

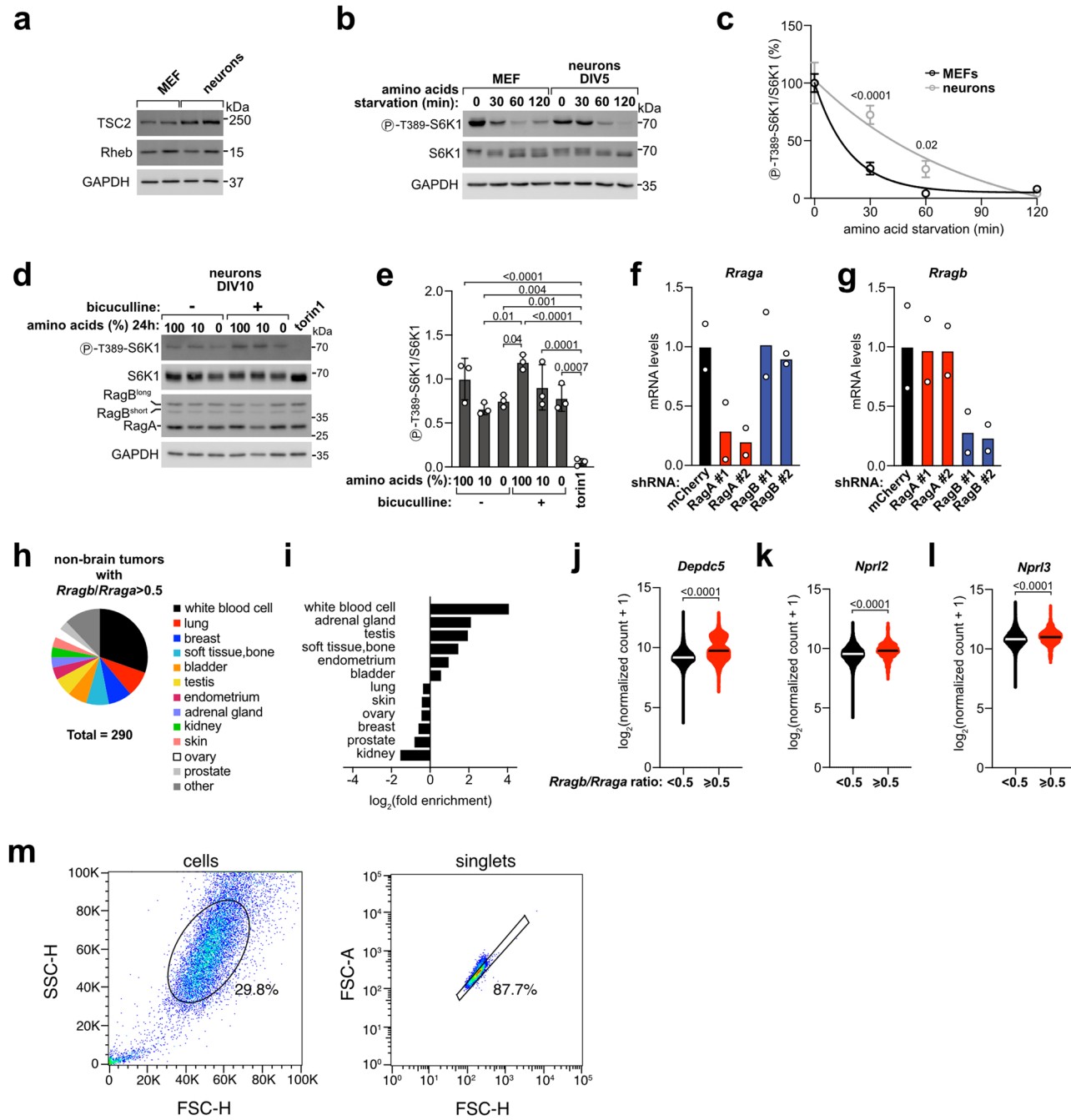

**Extended Data Fig. 10 | RagB determines mTORC1 resistance to starvation in neurons and cancer cells. (a)** Rheb and TSC2 expression in mouse cortical neurons at DIV 5 as compared to mouse embryonic fibroblasts (MEF). Two biological replicates per cell type are shown. GAPDH: loading control. **(b, c)** S6K1 phosphorylation in MEF or neurons (DIV 5) during amino-acid starvation for the indicated time points. (b) Representative example. (c) Quantification of 3 independent experiments, with the unstarved conditions set to 100%. Circle=average, error bars=standard deviation, n=3 biological replicates. Two-way ANOVA and Sidak's post-hoc test. **(d, e)** DIV 10 mouse cortical neurons were incubated in medium containing the indicated amino-acid concentrations (in amino-acid free MEM 1:10 in buffered saline solution, see methods) with/without bicuculline (50 μM) for 24 hours or torin1 (250 nM) for 30 minutes, as control. (d) Representative example. (e) Quantification of 3 independent experiments, with 100% aa/-bicuculline neurons set to 1. Bar Height=average, error bars=standard deviation, n=3 biological replicates. One-way ANOVA and Tukey's post-hoc test. **(f, g)** qPCR of *Rraga* (f) and *Rragb* (g) in DIV 10 neurons infected with viruses expressing a control shRNA (shmCherry) or two sets of shRNAs targeting RagA or RagB. Data are shown as fold change compared to the shmCherry condition after normalization to *Rpl13a*. Bar Height=average, n=2 biological replicates. **(h)** Primary site distribution of the TCGA cancer samples (excluding brain cancer samples) with a high *Rragb/Rraga* ratio (≥0.5) (n=290 biological replicates). other=primary sites representing individually <3%. **(i)** Primary site enrichment of the TCGA cancer samples (excluding brain cancer samples) with a high *Rragb/Rraga* ratio (≥0.5) (n=290 biological replicates) as compared to all non-brain TCGA cancer samples (n=9841 biological replicates). Only primary sites representing ≥3% of the TCGA cancer samples with a high *Rragb/Rraga* ratio were considered. **(j-l)** Violin plot of the mRNA levels of the three GATOR1 subunits *Depdc5* (j), *Nprl2* (k), and *Nprl3* (l) in cancer (TCGA) samples with low (<0.5) or high (>0.5) *Rragb/Rraga* ratio. Two-tailed, unpaired t-test, n=10210 (*Rragb/Rraga* < 0.5) or 325 (*Rragb/Rraga* > 0.5) biological replicates. **(m)** Gating strategy for the OPP incorporation experiment shown in Fig. 6r. Exact p values are shown in the graphs. Source numerical data and unprocessed blots are available in source data.

# Reporting Summary

## Statistics

For all statistical analyses, confirm that the following items are present in the figure legend, table legend, main text, or Methods section.

| n/a | Confirmed | |
|---|---|---|
| ☐ | ☒ | The exact sample size (*n*) for each experimental group/condition, given as a discrete number and unit of measurement |
| ☐ | ☒ | A statement on whether measurements were taken from distinct samples or whether the same sample was measured repeatedly |
| ☐ | ☒ | The statistical test(s) used AND whether they are one- or two-sided<br>*Only common tests should be described solely by name; describe more complex techniques in the Methods section.* |
| ☒ | ☐ | A description of all covariates tested |
| ☐ | ☒ | A description of any assumptions or corrections, such as tests of normality and adjustment for multiple comparisons |
| ☐ | ☒ | A full description of the statistical parameters including central tendency (e.g. means) or other basic estimates (e.g. regression coefficient) AND variation (e.g. standard deviation) or associated estimates of uncertainty (e.g. confidence intervals) |
| ☐ | ☒ | For null hypothesis testing, the test statistic (e.g. *F*, *t*, *r*) with confidence intervals, effect sizes, degrees of freedom and *P* value noted<br>*Give P values as exact values whenever suitable.* |
| ☒ | ☐ | For Bayesian analysis, information on the choice of priors and Markov chain Monte Carlo settings |
| ☒ | ☐ | For hierarchical and complex designs, identification of the appropriate level for tests and full reporting of outcomes |
| ☒ | ☐ | Estimates of effect sizes (e.g. Cohen's *d*, Pearson's *r*), indicating how they were calculated |

*Our web collection on statistics for biologists contains articles on many of the points above.*

## Software and code

Policy information about availability of computer code

| Data collection | Xena browser (xenabrowser.net) |
|---|---|
| Data analysis | Image Lab 5.2.1<br>Microsoft Excel 16.16.27<br>CellProfiler 3.1.5<br>GraphPad Prism 9.0.2<br>Xena browser (xenabrowser.net)<br>FlowJo (v10) |

For manuscripts utilizing custom algorithms or software that are central to the research but not yet described in published literature, software must be made available to editors and reviewers. We strongly encourage code deposition in a community repository (e.g. GitHub). See the Nature Portfolio guidelines for submitting code & software for further information.

## Data

Policy information about availability of data

All manuscripts must include a data availability statement. This statement should provide the following information, where applicable:
- Accession codes, unique identifiers, or web links for publicly available datasets
- A description of any restrictions on data availability
- For clinical datasets or third party data, please ensure that the statement adheres to our policy

Source data have been provided in Source Data. Data on RagA/B expression in brain cell types were extracted from the RNAseq dataset published by the Barres

group (GEO accession number: GSE52564). Data from the GTEX and TCGA databases used in Fig. 6 and Extended Data Fig. 10 were accessed through the Xena browser (xenabrowser.net). All other data supporting the findings of this study are available from the corresponding author on reasonable request.

# Field-specific reporting

Please select the one below that is the best fit for your research. If you are not sure, read the appropriate sections before making your selection.

☒ Life sciences   ☐ Behavioural & social sciences   ☐ Ecological, evolutionary & environmental sciences

For a reference copy of the document with all sections, see nature.com/documents/nr-reporting-summary-flat.pdf

# Life sciences study design

All studies must disclose on these points even when the disclosure is negative.

| | |
|---|---|
| Sample size | Samples sizes were chosen based on standard practice in the field (e.g. PMID: 18497260, 20381137, 26972053, 23723238, 24529379) |
| Data exclusions | No data were excluded. |
| Replication | All experiments were repeated two to three times with independent biological replicates. All attempts at replication were successful. For immunofluorescence and biochemical experiments, one representative experiment is shown. |
| Randomization | No randomization was used to allocate samples in experiments comparing different cell lines. When comparing different treatments within the same cell line, samples were assigned randomly to control or treatment group. |
| Blinding | No blinding was used since experiments required frequent intervention by investigators to maintain cell lines, thus precluding effective blinding. |

# Reporting for specific materials, systems and methods

We require information from authors about some types of materials, experimental systems and methods used in many studies. Here, indicate whether each material, system or method listed is relevant to your study. If you are not sure if a list item applies to your research, read the appropriate section before selecting a response.

### Materials & experimental systems

| n/a | Involved in the study |
|---|---|
| ☐ | ☒ Antibodies |
| ☐ | ☒ Eukaryotic cell lines |
| ☒ | ☐ Palaeontology and archaeology |
| ☐ | ☒ Animals and other organisms |
| ☒ | ☐ Human research participants |
| ☒ | ☐ Clinical data |
| ☒ | ☐ Dual use research of concern |

### Methods

| n/a | Involved in the study |
|---|---|
| ☒ | ☐ ChIP-seq |
| ☐ | ☒ Flow cytometry |
| ☒ | ☐ MRI-based neuroimaging |

## Antibodies

| | |
|---|---|
| Antibodies used | The following primary antibodies were used:<br>RagA (recognizing also both RagB isoforms) (Cell Signaling Technology #4357)<br>RagC (Cell Signaling Technology #9480)<br>RagD (Cell Signaling Technology #4470)<br>phospho-S6K1 T389 (Cell Signaling Technology #9205)<br>S6K1 (Cell Signaling Technology #2708)<br>phospho-TFEB S211 (Cell Signaling Technology #37681)<br>TFEB (Cell Signaling Technology #37785)<br>phospho-4EBP1 S65 (Cell Signaling Technology #9451)<br>4EBP1 (Cell Signaling Technology #9452)<br>FLAG tag (Sigma-Aldrich #F7425, 1:2500)<br>alpha-tubulin (Sigma-Aldrich #T9026, 1:5000)<br>mTOR (Cell Signaling Technology #2983, 1:200 for immunofluorescence, 1:1000 for western blots)<br>Raptor (Cell Signaling Technology #2280)<br>LAMP2 (Hybridoma Bank #H4B4, 1:50 for immunofluorescence, 1:500 for western blots)<br>calreticulin (Cell Signaling Technology #12238)<br>VDAC (Cell Signaling Technology #4661)<br>HA tag (Cell Signaling Technology #2367, used to detect HA-tagged proteins in whole cell lysates)<br>HA tag (Cell Signaling Technology #3724, used to detect HA-tagged proteins in immunoprecipitates)<br>LAMTOR1/p18 (Cell Signaling Technology #8975) |

GAPDH (Cell Signaling Technology #2118)
myc tag (Cell Signaling Technology #2278)
TSC2 (Cell Signaling Technology #4308)
Rheb (Abnova #H00006009-M01)
DEPDC5 (Abcam #ab185565, used to screen DEPDC5KO clones)
calnexin (Enzo #ADI-SPA-960-D).

All antibodies were diluted 1:1000 unless indicated otherwise.

The following secondary antibodies were used:
anti-rabbit HRP (Jackson ImmunoResearch #111-035-003, used 1:10000)
anti-mouse HRP (Jackson ImmunoResearch #115-035-003, used 1:10000)
anti-rabbit Alexa488 (Life Technologies #A11008, used 1:500)
anti-mouse TRITC (Jackson ImmunoResearch #715-025-151, used 1:200)

| Validation | RagA, RagC, RagD: validated for WB in previous publications and by this study through corresponding knockout cell lines (Extended Data Fig. 1a,d)
phospho-S6K1 T389 and S6K1: validated for WB by the manufacturer (cells treated with insulin) and by this study through treatments that affect mTORC1 activity (e.g. Fig. 1e)
phospho-TFEB S211 and TFEB: validated for WB by the manufacturer (cells treated with torin1) and by this study through treatments that affect mTORC1 activity (e.g. Fig. 1e)
phospho-4EBP1 S65 and 4EBP1: validated for WB by the manufacturer (cells treated with insulin) and by this study through treatments that affect mTORC1 activity (e.g. Fig. 1e)
FLAG tag, HA tag (#2367 and #3724), and myc tag: validated for WB by the manufacturer (cells overexpressing tagged proteins) and by this study through overexpression of tagged proteins (e.g. Fig. 2g,j)
alpha-tubulin: validated for WB by the manufacturer using independent antibody validation
mTOR: validated for immunostaining in Sancak Y et al., Science, 2008 (PMID:18497260), for WB in Sherman et al., J Neurosci, 2012 (PMID: 22302821)
Raptor: validated for WB in Rogala et al., Science, 2019 (PMID: 31601708)
LAMP2: validated for WB in Majer F, Gene, 2012 (PMID: 22365987)
calreticulin: validated for WB in Wolfson et al., Nature, 2017 (PMID: 28199306)
VDAC: validated for WB in Li et al., Sci Rep, 2018 (PMID: 29572489)
LAMTOR1/p18: validated for WB in Son et al., Cell Metab, 2019 (PMID: 30197302)
GAPDH: validated for WB in Mungrue et al., J Immunol, 2009 (PMID: 19109178)
TSC2: validated for WB by the manufacturer using a TSC2-null cell line
Rheb: validated for WB by the manufacturer using recombinant protein
DEPDC5: validated for WB in De Fusco et al., Neurobiol. Dis., 2020 (PMID: 32113911)
calnexin: validated for WB in Gardner et al., Arch Biochem Biophys, 2000 (PMID: 10871059) |
|---|---|

# Eukaryotic cell lines

Policy information about cell lines

| Cell line source(s) | HEK293T: ATCC #CRL-3216
HEK293: Stratagene #240073
Neuro-2a: ATCC #CCL-131
U2OS: ATCC #HTB-96
HeLa: ATCC #CCL-2
HepG2: ATCC #HB-8065
EFO-21: DMSZ #ACC 235 |
|---|---|
| Authentication | None. |
| Mycoplasma contamination | No mycoplasma contamination was detected by qPCR. |
| Commonly misidentified lines (See ICLAC register) | None of these cell lines were used. |

# Animals and other organisms

Policy information about studies involving animals; ARRIVE guidelines recommended for reporting animal research

| Laboratory animals | C57B6/J time-mated 2-3 months old female mice (E15): isolation of MEFs and neurons
C57B6/J 2-3 months old female mice: tissue extraction
C57BL/6NCrl P0 pups: isolation of neurons |
|---|---|
| Wild animals | No wild animals were used. |
| Field-collected samples | No samples were field-collected. |
| Ethics oversight | All animal experiments conform to the relevant regulatory standards of the German Cancer Research Center and have been approved by the ethical authorities, Regierungspräsidium Karlsruhe, Germany. |

Note that full information on the approval of the study protocol must also be provided in the manuscript.

# Flow Cytometry

## Plots

Confirm that:

☒ The axis labels state the marker and fluorochrome used (e.g. CD4-FITC).

☒ The axis scales are clearly visible. Include numbers along axes only for bottom left plot of group (a 'group' is an analysis of identical markers).

☒ All plots are contour plots with outliers or pseudocolor plots.

☒ A numerical value for number of cells or percentage (with statistics) is provided.

## Methodology

| | |
|---|---|
| Sample preparation | 20 µM OPP reagent (Jena Bioscience #NU-931-05) was added for 30 minutes to the cells. The cells were subsequently washed with DPBS, trypsinized, and fixed with ice-cold 70% ethanol for 30 minutes at -20C degrees, followed by three washes in PBS supplemented with 0.5% Tween-20. The incorporated OPP was then labeled with the Alexa488 Fluor Picolyl azide using the Click-iT Plus OPP Protein synthesis assay kit (Life Technologies #C10456), as per manufacturer's instructions. |
| Instrument | Guava easyCyte HT flow cytometer (Millipore) |
| Software | FlowJo (v10) |
| Cell population abundance | 29.8% (87.7% singlets) |
| Gating strategy | The cell population of interest was identified plotting FSC-H vs SSC-H, singlets gated by plotting FSC-H vs FSC-A, and the mean intensity of the Alexa488 signal within the singlets population was used to quantify the extent of OPP incorporation. |

☒ Tick this box to confirm that a figure exemplifying the gating strategy is provided in the Supplementary Information.

