## [Peer Review File · Nature Cell Biology]

Peer Review Information

Journal: Nature Cell Biology

Manuscript Title: Brain-enriched RagB isoforms regulate the dynamics of mTORC1 activity through GATOR1 inhibition

Corresponding author name(s): Aurelio Teleman

Reviewer Comments & Decisions:

Decision Letter, initial version:

Dear Dr Teleman,

Thank you for submitting your manuscript "Brain-enriched RagB isoforms regulate the dynamics of mTORC1 activity via GATOR1 inhibition", to Nature Cell Biology, and thank you for your patience with the peer review process. It has now been seen by 3 referees, who are experts in neuronal cell biology and mTORC/trafficking (referee 1); mTORC (referee 2); and mTORC biology (referee 3), and whose comments are pasted below. In light of their advice, we regret that we cannot offer to publish the study in Nature Cell Biology.

As you will see, although the reviewers found this work interesting, they raised serious concerns that question the strength of the data and of the novel conclusions that can be drawn at this stage. The reviewers had consistent concerns regarding the strength of the evidence, sharing similar important technical and experimental concerns. Another major limit for NCB is the lack of analyses of the physiological functions of RagB isoforms in the brain. We agree with Reviewer #1 (a neuroscientist) that this line of investigation is needed to support the advance and increase our understanding of the tissue-specific regulation of mTORC1 activity.

Significant additional experimentation would be needed to address the reviews. We feel that this amount of work is greater than what would be reasonable to request in a standard revision period.

Given our interest in this area, we would however be open to the possibility of considering a revised manuscript that would fully address the referee concerns. However, any decision to re-review such a revised study would depend on the strength of the revisions and the published literature at the time of resubmission.

We are very sorry that we could not be more positive on this occasion, but we thank you for the opportunity to consider this work. I hope that the reviews will be helpful as you consider how to move forward with the work. Thank you again for considering NCB for this work.

With kind regards,
Melina

Melina Casadio, PhD
Senior Editor, Nature Cell Biology
ORCID ID: <https://orcid.org/0000-0003-2389-2243>

Reviewers' comments:

Reviewer #1 (Remarks to the Author):

In this Ms Figlia et al analyze how mTORC1 maybe differentially regulated by distinct isoforms or RagA vs RagB. Based on the observation that different isoforms of RagA and RagB are differentially expressed in various tissues including brain, they use HEK293T KO cell lines to reconstitute the effects of RagA and the short or long isoforms of RagB on mTORC1 signaling (readout by S6K1 phosphorylation) and the effects of amino acid starvation. Evidence is provided, mostly based on single immunoblots, that the activities of RagBshort and RagBlong are more resistant to amino acid starvation compared to RagA. It is shown that RagA or RagBshort are sufficient for mTORC1 recruitment to lysosomes, while RagBlong is not. This phenotype correlates with a comparably poor ability of RagBlong to interact with Raptor/ mTORC1, even when loaded with GTP. Further cell-based assays that probe the levels of pS6K1 activation suggest that RagBshort and RagBlong may be somewhat resistant to inactivation via the Rag GAP complex GATOR1 that is activated upon amino acid depletion. Further mapping experiments suggest distinct mechanisms by which RagB isoforms could modulate GATOR1 activity: From co-immunoprecipitation experiments in cells and pulldowns it is concluded that RagBlong has low affinity for GTP and for GATOR1, but displays comparably (i.e. higher than RagA or RagBshort) strong interactions with the GAP interface of GATOR1 and, thereby, can inhibit GAP activity. Moreover, RagBshort is shown to bind to the DEPDC5 subunit of GATOR1, an interaction that was previously established to be inhibitory. In a final dataset it is shown that neurons and some tumor cells shift the balance of RagA/B expression towards RagB, a feature that may allow them to sustain an active pool of mTORC1 even under low amino acid supply. In support of this possibility Figlia et al show that loss of RagA in a neuroblastoma cell line renders these cells relatively resistant to amino acid starvation. A similar phenotype (based on single immunoblots) is seen in EFO21 tumor cells. Based on these results the authors propose a function for brain-enriched RagB isoforms in the tissue-specific regulation of mTORC1 signaling via GATOR1.

The question how mTORC1 activity is regulated in different cell types or tissues is of high relevance and interest to the community. The claimed differential function of distinct RagA vs RagB isoforms is novel and potentially interesting. My enthusiasm for the present Ms is, however, dampened by the lack of quantitative data and the limited insight the paper provides into the physiological functions of the different Rag isoforms in native cells or tissues. Moreover, some of the data appear overinterpreted. Thus, the study in my view does not rise to the standard or significance of a Nat Cell Biol paper.

Major issues:

1. The paper suffers from lack of quantification and missing experimental replicates that greatly limit my enthusiasm for this study, in spite of the general interest of the topic and the quality of data

2provided. Essentially all key conclusions appear to be based on single immunoblots throughout the paper, e.g. Figure 1d,e,h,i; Figure 2a-f (d represents single data points); Figure 3b-e,h,i; Figure 4b-f; Figure 5c-e; Figure 6d-h, o, q and with only single data points shown for key data in panels g,i,p,r. Similar concerns hold for supplemental data figures 1-5.

Multiple experimental replicates are required and quantifications from these independent experiments ought to be analyzed statistically.

2. Similar to my point above, all quantifications regarding "rates", e.g. S4, are based on single data points. This is insufficient. Moreover, the use of the term "affinity" in the context of RagA/B association with Raptor and GATOR1 requires the experimental determination of Kds, not a simple CoIP experiment using overexpressed proteins.

3. Several statements do not seem to be sufficiently supported by the data. E.g. "Indeed, we observed higher mTORC1 activity in nutrient-rich conditions in RagABKO cells transfected with RagBlong in addition to RagA and RagBshort (Figure 5c)." Looking at the WB I see no real evidence to support this, while the effects of RagB co-expression on pS6K1 levels/ mTORC1 activity in response to full or partial amino acid starvation appear to be minor. Also, the authors do not seem to consider that co-expression of all three Rag isoforms leads to elevated total Rag levels, which is expected to result in higher mTORC1 activity, not necessarily as a consequence of a particular isoform. In Figure 4b, expression levels of the various Rags are different, e.g. RagBlong is higher expressed in the negative control. Thus, the conclusion that RagBlong interacts with GTP "almost at background levels" cannot be made.

4. The data in support of the proposed mechanism of RagBshort vs long modulation of GATOR1 function remain very indirect. To probe the interaction of RagA/B isoforms with GATOR1, the authors co-immunoprecipitate Rags with either the full GATOR1 complex (by overexpressing DEPDC5, Nprl2, Nprl3) or Nprl2 and Nprl3 in DEPDC5 KO cells (Figure 3b,c). This approach is problematic for several reasons: (1) Expression levels of Rags differ (e.g. RagBshort > RagA). (2) The influence of other proteins on complex composition cannot be excluded. (3) The absence of DEPDC5 might influence binding of Nprl2/3 to the Rags. Two sets of experiments should be conducted to tackle these issues: First, the interaction of different Rag isoforms with GATOR1 needs to be characterized in more detail using purified proteins.

Second, to support the claim that RagB but not RagA can inhibit GATOR1-GAP function via binding to DEPDC5 GAP activity assays using purified proteins (similar to Shen et al., Nature 2018) should be carried out. These assays need to include GDP loaded RagBlong to show decreased GAP activity of GATOR1 towards RagA and directly address the proposed GATOR1 GAP inhibition by RagBlong.

5. Statistics in Figures 1g, S1f,h and S2c are based on fields of view. This can hardly be considered independent biological experiments. Statistical analysis should be performed on at least 3 independent experimental replicates and tested for significance.

With respect to Fig 1g, it is unclear what exactly is meant by mTOR signal overlapping with LAMP2 Pearson correlation would be one way to assess this.

6. The manuscript falls short of providing compelling physiological data to support key functions of RagB isoforms in the brain or primary tissue. For example, mTORC1 hyperactivity is strongly implicated in epilepsy. Is RagB overexpressed in some inherited forms of epilepsy? Given that the metabolism of neurons and astrocytes is intimately linked, most importantly via the glutamine/ glutamate cycle that impinges on mTORC1 activity, it is surprising that no efforts are made to

understand the physiological relevance of the claimed mechanism and the roles of the different RagA/B isoforms in this context.

What are the consequences of RagB inactivation in primary neurons with respect to mTORC1 signaling, cell and neurite outgrowth, and neuronal excitability (all known to be regulated by mTORC1)?

7. The ratio of RagB/A in Neuro2a appears to be quite different from that in primary neurons (Fig. 6d-f), likely because these are tumor-derived. Hence, the conclusions derived from Neuro2a experiments cannot be transferred to primary neurons or to the brain.

8. Essentially all conclusions rest on pS6K1 levels as a readout for mTORC1 activity. Key conclusions should be confirmed for other substrates such as p4E-BP1 or pULK1. This might be especially interesting in light of recent findings showing that mTOR substrate phosphorylation (for example S6K1 vs TFEB) is differentially controlled by different regulators of the pathway (Napolitano et al, *Ballabio Nature* 2020). It is conceivable that RagB affects only a subset of mTORC1 substrates but not others in a given tissue.

9. The authors often capitalize on GTP locked variants of the Rags. As an important control the interaction of the different WT RagA/B isoforms with the Ragulator GEF complex in cells should be assessed to make sure that the effects seen are not due to different modes of Ragulator-mediated activation.

Minor points:

10. LAMP2 staining seems to be affected by RagA/B or RagC/D loss in Fig. S1e,g.

11. Fig S1: pS6K1 levels of DKO clones are well above those of Ctrl in starved cells. How is this explained?

12. Blots for HA-tagged Rags in the lysate are often missing (e.g. Figure 2a, Figure 3b,c,h, Figure 4c,f, Figure S1d, Figure S3c) and should be added to the figure.

13. P-TFEB blots are of low quality (especially in Figures 1d,e, S1b,c). Better data are needed.

14. Do cortical neurons used for the experiments in Fig 6d,e contain astrocytes? Again, no serious quantifications of the time course of mTORC1 inactivation and, most importantly, its dependence on RagB isoforms are presented.

15. The order of panels should be consistent throughout the paper. Figure 5 b and c; Figure 6 e and f should be swapped

16. Figure 5c,d,e: please indicate that RagABKO cells were used.

17. The authors should explain why they overexpress FLAG-S6K1 in some experiments to assess pS6K1 levels but analyze the endogenous protein in others.

Reviewer #2 (Remarks to the Author):

4Figlia et al. investigate the role of RagB in mTORC1 resistance to nutrient starvation. Specifically, the authors look at the mechanisms involved in the neurons. There are multiple concerns prior to publication. However, the mechanistic data is pretty strong.

1. It is not clear why there is still high S6K1 phosphorylation in the RagAB KO and RagCD KO cell lines? One would anticipate a significant decrease. Is this a clonal effect? Does knockdown (siRNA) of RagA and RagB or RagC and RagD have a high level of S6K1 phosphorylation under nutrient deficient conditions?
2. Sup. Fig. E-H. Lyso-IP (PMID: 29074583) experiments should be performed here showing that Raptor and mTOR are not at the lysosome in RagABKO and RagCDKO cells. Also, in RagAB KO cells are RagC and Rag D still at the lysosome. In RagCD KO cells are RagA and RagB (short and long) still at the lysosome.
3. If RagB long can maintain mTORC1 activity but it's not at the lysosome, where is mTORC1 in the cell? Does RagB long interact with mTORC1 to the same extent as RagB short (Wild-type RagB forms)?
4. Fig. 1H. Deletion of RagA doesn't look like it increases RagB (short and long) expression as mentioned on Page 9.
5. Fig. 2C. The decrease looks the same when overexpressing GATOR1 in cells expressing RagA and RagB short. Fig 2D. Should have error bars with replicate blots.
6. Fig. 6D. Is mTORC1 signaling higher in neurons due to an elevated RagB? What about RagC/D? Fig. 6G. Include error bars of replicates.
7. Fig. 6E. Doesn't appear to see a change. Perhaps add graph of phosphorylation of S6K with error bars of the replicate.
8. Fig. 6I. Add error bars of replicates.
9. Fig. 6H-I. Doesn't appear to be a difference? Authors state say a decrease in mTORC1 activity after the knockdown of RagB. Error bars and replicates.
10. Fig. 6N. Would the authors expect an increase in RagC/D in EF021 cells?
11. Fig. 6Q-R. Do not see a difference between siLuciferase and siRagB in terms of mTORC1 activity? Error bars of replicates. Same with 6P.
12. What is the physiological reason to have higher levels of RagB short in EF021 cells?
13. For the cell lines used in Fig 6O-Q, have the authors thought about using them in a xenograft model to assess tumor size. In terms of biology, you would anticipate higher mTORC1/tumor size (in siRagA cells) compared to control cells. And lower mTORC1/tumor size (in siRagB cells) compared to control cells.

Minor

1. Fig. 1D. is not in text.
2. RagB long should be included in Fig. 2B.
3. Manuscript hard to follow. Flow and organization

Reviewer #3 (Remarks to the Author):

Figlia et al. report that the RagBshort and RagB long isoforms have distinct mechanisms through which they regulate the activity of mTORC1, with RagBG short inhibiting Gator1 via DEPDC4, and RagB long inhibiting Gator1 via Nprl2/3. RagB and RagA are highly homologous (almost identical in the GTPase and CRD domains), and have been considered interchangeable in terms of their function. RagB has a 33 aa N-terminal extension that is absent in RagA, and RagB long has an additional 28 aa

5in the switch I region. Figlia et al. knocked out both RagA and RagB, and then re-expressed a single RagA or B isoform. They also did this with RagC/D. There are many strengths in this work, including one of the first attempts to define differences between RagA and RagB. Concerns, as detailed below, include the system itself (especially as it relates to the two RagB isoforms), the non-physiologic expression levels, and the lack of quantitative analysis of the many Western blots. It would appear that some conclusions are based on a single Western, rather than densitometric analyses of triplicate blots.

1. Significant concerns about the system itself weaken this work. These include the level of expression of the reconstituted cells vs. endogenous and the interpretation of cells in which RagB long is expressed alone, since it would appear from Figure 1B that expression of a single RagB isoform does not occur physiologically. It is also a concern that the HEK293 cells, used as a model here, express predominantly RagA, and therefore could lack some (unknown) factors required for RagB-dependent cellular responses. Supp Fig 1 shows that there is ~800 fold higher expression of RagA at the mRNA level. There is no evidence that RagB long is expressed by HEK293 cells. This leads to fundamental concerns about how to interpret these otherwise interesting data.

2. Figure 1D shows differential responses to Phospho-S6K and phospho-TFEB. It is surprising that levels of p-S6K are relatively high in the RagBKO cells; this is commented on but with no hypothesis for why this is observed – it is actually higher than cells with RagA re-expressed after 15 min of aa addback. Could this reflect a technical problem with this system? How do the levels of the overexpressed proteins compare with endogenous levels? No endogenous RagA or B appears on the Western, suggesting that levels could be much, much higher than physiologic levels.

3. The legend for this figure simply states that the cells were grown in “normal” media; the specific media should really be specified in the figure legend since this is critical to the interpretation (even if also in the methods section).

4. Figure 1D and E (and other similar blots) should include quantitation of the changes in p-S6K, based on triplicate blots. The changes appear to be rather modest compared with the knockout cells, and it is challenging to know whether there are any true differences between the isoforms. The changes in TFEB phosphorylation are impossible to interpret. It would be of interest to include phospho-S6 in this figure.

5. The last paragraph of the discussion about Figure 1 states that RagB long does not recruit mTOR to the lysosome as efficiently as the other isoforms but based on Fig 1G it does not recruit mTOR at the lysosomes at all.

6. The authors state at the beginning of the text section on Gator1 that “mTORC1 activity remains high in RagB expressing cells upon aa removal” but this is not really clear from Fig 1. In Fig 1D, the levels of activity (as judged by p-S6K) in RagB short and RagB long cells upon aa withdrawal appear similar to the cells expressing the Flag vector.

7. Regarding Fig 2A, the authors state that the three Rag isoforms “release mTORC1 similarly when in the GDP conformation” but the results in Fig 2A suggest that RagB long does not bind (or barely binds) Raptor even when in its active form.

8. In Fig 2C, it is surprising that the authors needed to IP S6K to detect its phosphorylation; in Figure

61 and throughout the literature, S6K phosphorylation is readily detected in the cell lysate. While the authors place a great deal of emphasis on the observation that “only the highest levels of GATOR1” can decrease ph-S6K in the RagB long cells, it is clear from Fig1 that these cells have levels of ph-S6K that are comparable to the vector control cells (Fig 1D). So, it does not seem surprising that Gator1 does not decrease ph-S6K in these cells. Again, it is concerning that the authors show quantitation (Fig 1D) that appears to be based on a single experiment and lacks statistical analysis.

9. Figure 2D seems to add little – here it is shown that GATOR1 overexpression decreases phospho-S6K in cells with RagA KO (that express a very small amount of RagB). The authors state that this is to test the hypothesis that “GATOR1 insensitivity is the cause of high mTORC1 activity in RagB expressing cells” but this does not make sense, since high mTORC1 is only seen in RagB long (Fig 2C/D).

10. The conclusion that differential resistance to GATOR1 is the “main functional distinction between RagB short and RagA” seems to be only partially supported by the data in Figure 2.

11. In Figure 3D, the authors tested mTORC1 activity (again by IP of S6K) with a mutant of DEPDC5 that does not bind the Rags.

**Although we cannot publish your paper, it may be appropriate for another journal in the Nature Portfolio. If you wish to explore the journals and transfer your manuscript please use our manuscript transfer portal. If you transfer to Nature journals or the Communications journals, you will not have to re-supply manuscript metadata and files. This link can only be used once and remains active until used.

All Nature Portfolio journals are editorially independent, and the decision on your manuscript will be taken by their editors. For more information, please see our manuscript transfer FAQ page.

Note that any decision to opt in to In Review at the original journal is not sent to the receiving journal on transfer. You can opt in to In Review at receiving journals that support this service by choosing to modify your manuscript on transfer. In Review is available for primary research manuscript types only.

**For Nature Research general information and news for authors, see <http://npg.nature.com/authors>.

Author Rebuttal to Initial comments

Reviewer #1

In this Ms Figlia et al analyze how mTORC1 maybe differentially regulated by distinct isoforms or RagA vs RagB. Based on the observation that different isoforms of RagA and RagB are differentially expressed in various tissues including brain, they use HEK293T KO cell lines to reconstitute the effects of RagA and the short or long isoforms of RagB on mTORC1 signaling (readout by S6K1 phosphorylation) and the effects of amino acid starvation. Evidence is provided, mostly based on single immunoblots, that the activities of RagBshort and RagBlong are more resistant to amino acid starvation compared to RagA. It is shown that RagA or RagBshort are sufficient for mTORC1 recruitment to lysosomes, while RagBlong is not. This phenotype correlates with a comparably poor ability of RagBlong to interact with Raptor/ mTORC1, even when loaded with GTP. Further cell-based assays that probe the levels of pS6K1 activation suggest that RagBshort and RagBlong may be somewhat resistant to inactivation via the Rag GAP complex GATOR1 that is activated upon amino acid depletion. Further mapping experiments suggest distinct mechanisms by which RagB isoforms could modulate GATOR1 activity: From co-immunoprecipitation experiments in cells and pulldowns it is concluded that RagBlong has low affinity for GTP and for GATOR1, but displays comparably (i.e. higher than RagA or RagBshort) strong interactions with the GAP interface of GATOR1 and, thereby, can inhibit GAP activity. Moreover, RagBshort is shown to bind to the DEPDC5 subunit of GATOR1, an interaction that was previously established to be inhibitory. In a final dataset it is shown that neurons and some tumor cells shift the balance of RagA/B expression towards RagB, a feature that may allow them to sustain an active pool of mTORC1 even under low amino acid supply. In support of this possibility Figlia et al show that loss of RagA in a neuroblastoma cell line renders these cells relatively resistant to amino acid starvation. A similar phenotype (based on single immunoblots) is seen in EFO21 tumor cells. Based on these results the authors propose a function for brain-enriched RagB isoforms in the tissue-specific regulation of mTORC1 signaling via GATOR1.

The question how mTORC1 activity is regulated in different cell types or tissues is of high relevance and interest to the community. The claimed differential function of distinct RagA vs RagB isoforms is novel and potentially interesting. My enthusiasm for the present Ms

8is, however, dampened by the lack of quantitative data and the limited insight the paper provides into the physiological functions of the different Rag isoforms in native cells or tissues. Moreover, some of the data appear overinterpreted. Thus, the study in my view does not rise to the standard or significance of a Nat Cell Biol paper.

We thank the reviewer for the positive comment that the discovery of this novel regulatory mechanism of mTORC1 is of high relevance and interest.

We have now added a significant amount of new data (90 figure panels) to address the issues raised here by our colleague. In particular, we added quantifications to all the immunoblots based on multiple biological replicates, confirming that all the results are statistically significant. We also added data describing the physiological consequences of RagA versus RagB in regulating neuron dendrite formation.

We would like to kindly point out, however, that the focus of this manuscript is the discovery of a novel mTORC1 regulatory mechanism, and the dissection of its molecular underpinnings. An in-depth characterization of the physiological consequences in primary cells and tissues goes beyond the scope of this manuscript. Indeed, this is generally uncommon in studies that focus on the molecular dissection of a novel mTORC1 regulatory mechanism, even when published in top journals. For instance, the discoveries of the Rag proteins published in Science (Sancak et al., Science, 2008), of Ragulator published in Cell (Sancak et al., Cell, 2010), of GATOR1 published in Science (Bar-Peled et al., Science, 2013) of CASTOR published in Cell (Chantranupong et al., Cell, 2016) all focused on understanding the new regulatory mechanism and tested functional but not physiological consequences. This is both because it is beyond doubt that mTORC1 signaling is physiologically relevant, and because understanding the physiological consequences constitutes an entire additional project that takes years.

Nonetheless, as requested, we now provide data on primary neurons showing that RagA knockdown versus RagB knockdown have the expected differential effect on mTORC1 activity (Fig. 6e-f) and on dendrite outgrowth (Fig. 6g). We hope the reviewer finds these results satisfactorily address the issues he raised.

Major issues:

1. The paper suffers from lack of quantification and missing experimental replicates that greatly limit my enthusiasm for this study, in spite of the general interest of the topic and the quality

of data provided. Essentially all key conclusions appear to be based on single immunoblots throughout the paper, e.g. Figure 1d,e,h,i; Figure 2a-f (d represents single data points); Figure 3b-e,h,i; Figure 4b-f; Figure 5c-e; Figure 6dh, o, q and with only single data points shown for key data in panels g,i,p,r. Similar concerns hold for supplemental data figures 1-5.

Multiple experimental replicates are required and quantifications from these independent experiments ought to be analyzed statistically.

Although we showed only one representative blot, we of course have multiple independent biological replicates of all our data, because we make sure they are reproducible before submitting a manuscript. We have now added these quantifications in the manuscript as:

Figures 1f, g, h, j, l, n, p,

Figures 2b, d, h, k

Figures 3b, d, f, h

Figures 4c, e, g

Figure 5b, d, f, h

Figure 6f, n, p

ED Fig. 1b, c, e, f, i, j, k

ED Fig. 2b

ED Fig. 3f, g, h, i, j, k, m, n

ED Fig. 4d, f

ED Fig. 5b, d, f

ED Fig. 6b, d, e, h, j

ED Fig. 8b, d, f, h, j, l

ED Fig. 9f, g, i, k, m

ED Fig. 10c, e

These quantifications support all our original conclusions and show that the effects are statistically significant.

2. Similar to my point above, all quantifications regarding "rates", e.g. S4, are based on single data points. This is insufficient.

We now provide replicates, quantifications, and statistical analyses for all timecourse experiments (Figures 4f-g, 6m-n, 6o-p, ED2a-b, ED9h-i, ED10b-c).

Moreover, the use of the term "affinity" in the context of RagA/B association with Raptor and GATOR1 requires the experimental determination of Kds, not a simple CoIP experiment using overexpressed proteins.

We thank the reviewer for pointing out that our use of the term "affinity" is not correct in this context. We have re-worded the text to say that RagA/B binds more or less GATOR1 or Raptor, which accurately describes what we see in the experiments.

3. Several statements do not seem to be sufficiently supported by the data. E.g. "Indeed, we observed higher mTORC1 activity in nutrient-rich conditions in RagABKO cells transfected with RagBlong in addition to RagA and RagBshort (Figure 5c)." Looking at the WB I see no real evidence to support this, while the effects of RagB co-expression on pS6K1 levels/ mTORC1 activity in response to full or partial amino acid starvation appear to be minor.

(This figure is now Fig 5a-h). We have now added quantifications to these blots which show that in medium containing 50% the normal amino acid concentration (Fig. 5a-b) S6K phosphorylation is 70% higher in cells expressing RagA + RagBshort + RagBlong compared to RagA only and this is highly significant ($p < 0.001$), while in medium with 25% amino acids (Fig. 5c-d) cells expressing all three RagA/B isoforms retain twice as much S6K phosphorylation compared to cells expressing RagA only (also very significant, with $p < 0.001$). An analogous effect, although milder, can also be observed at 10% the normal amino acid concentration, which we now also add to our revised manuscript (Fig. 5e-f). We now also

summarize the activity of mTORC1 in response to varying amino acid concentrations in a graph in Fig. 5i, where we combine all our experiments of amino acid titration. As correctly pointed out by the reviewer and also discussed in our manuscript, expression of all three RagA/B isoforms cannot however prevent mTORC1 inactivation upon complete and prolonged amino acid starvation

(Fig. 5g-h).

Also, the authors do not seem to consider that co-expression of all three Rag isoforms leads to elevated total Rag levels, which is expected to result in higher mTORC1 activity, not necessarily as a consequence of a particular isoform.

We thank the reviewer for bringing this up. Indeed, we did consider this aspect, which is why we kept the total amount of Rag plasmid the same in each condition and distributed this total amount amongst the various Rag isoforms. For this reason, the reviewer may note in Fig. 5a that RagA levels drop when we add in RagB isoforms. We now mention this explicitly in the main text and in the corresponding figure legend, and we have added a quantification of total Rag levels in ED Fig 9g which show that the total amount of Rag proteins in the RagA + RagBshort + RagBlong cells is equal to the level of RagA in the RagA-only expressing cells. Hence the phenotypes we observe derive from differences in function, not levels.

In Figure 4b, expression levels of the various Rags are different, e.g. RagBlong is higher expressed in the negative control. Thus, the conclusion that RagBlong interacts with GTP "almost at background levels" cannot be made.

(Now Figure 4b-c). To support this conclusion more convincingly, we now provide biological replicates and have quantified the amount of Rags in the pulldown normalized to the Rags in the whole cell lysate (after normalization for tubulin) as panel c of the same figure. The statistics show that RagB^{long} binds much less to GTP compared to RagA or RagB^{short} ($p < 0.001$).

4. The data in support of the proposed mechanism of RagBshort vs long modulation of GATOR1 function remain very indirect. To probe the interaction of RagA/B isoforms with GATOR1, the authors co-

12immunoprecipitate Rags with either the full GATOR1 complex (by overexpressing DEPDC5, Nprl2, Nprl3) or Nprl2 and Nprl3 in DEPDC5 KO cells (Figure 3b,c). This approach is problematic for several reasons:

Please see the detailed responses to the individual points below, nonetheless, as suggested by the reviewer, we now confirm the binding results and the proposed mechanism using purified proteins (ED Fig. 9e-f) and in vitro GAP assays (Fig. 4i-j, ED Fig. 7a-e, ED Fig. 9a-d). This shows that the mechanism is correct and direct.

(1) Expression levels of Rags differ (e.g. RagBshort > RagA).

We have now quantified all our coIP experiments (Fig. 2h,k; Fig. 4e; ED Fig. 3m-n; ED Fig. 4f; ED Fig. 5b; ED Fig. 6h; ED Fig. 8b,d, f, j, l), normalizing the coIPed protein for both the amount of the immunoprecipitated protein in the IP and for the levels of the co-IPing protein in the whole cell lysate. The results and our conclusions thus take into account potential differences in the levels of the Rags or their coimmunoprecipitating proteins.

(2) The influence of other proteins on complex composition cannot be excluded.

We would like to emphasize that the interactions between the Rags and GATOR1 are known to be direct, both on the DEPDC5 and Nprl2/3 side, as also confirmed by structural studies (Shen et al., Nature, 2018; Egri et al., Mol. Cell, 2022). Thus, the presence of such additional proteins would not invalidate any of our results, but simply mean that additional proteins are also at play. Nonetheless, we now show that we can recapitulate analogous effects in vitro with purified proteins (see the points below), indicating that the mechanisms we propose are indeed to a great extent direct.

(3) The absence of DEPDC5 might influence binding of Nprl2/3 to the Rags.

We use here an approach analogous to Shen et al., Nature, 2018, where the Nprl2/3 dimer was coIPed with the Rags using cell lysates from DEPDC5KO cells and was additionally shown to be competent for GAP activity in in vitro experiments in the absence of DEPDC5. Thus, the Nprl2/3-Rag interaction is known to be the most relevant one when considering the GAP interface of GATOR1. Nonetheless, to address the reviewer's concern, we now show also similar patterns of interaction between

Npr12/3 and the Rags in vitro using a purified GATOR1 complex containing the Y775A DEPDC5 mutant that fully reconstitutes the GATOR1 complex yet abolishes specifically binding through the inhibitory interface of GATOR1 (ED. Fig. 9e-f).

Two sets of experiments should be conducted to tackle these issues: First, the interaction of different Rag isoforms with GATOR1 needs to be characterized in more detail using purified proteins.

We now confirm the binding results and the proposed mechanism using purified proteins (ED Fig. 9e-f) and in vitro GAP assays (Fig. 4i-j, ED Fig. 7a-e, ED Fig. 9a-d).

Second, to support the claim that RagB but not RagA can inhibit GATOR1-GAP function via binding to DEPDC5 GAP activity assays using purified proteins (similar to Shen et al., Nature 2018) should be carried out. These assays need to include GDP loaded RagB^{long} to show decreased GAP activity of GATOR1 towards RagA and directly address the proposed GATOR1 GAP inhibition by RagB^{long}.

We thank the reviewer for suggesting these experiments, which we believe corroborated our proposed model. We now include these in vitro GAP activity data as Fig. 4i-j, ED Fig. 7a-e, and ED Fig. 9a-d.

ED Figure 7d-e shows that GATOR1 has reduced GAP activity towards RagB^{short} compared to RagA in a manner dependent on binding to the inhibitory interface of GATOR1. ED Figure 9a-d shows that GATOR1 has no detectable GAP activity towards RagB^{long}. As suggested by the reviewer, main Figure 4i-j shows that addition of non-GTP bound RagB^{long}, but not an analogous mutant of RagB^{short}, decreases the GAP activity of GATOR1 towards RagA. (Note that this mutant of RagB^{short} does not bind the inhibitory interface of GATOR1, and hence this only assays inhibition of GATOR1 through the GAP interface). Together, these in vitro data show that RagB^{short} inhibits GATOR1 via the inhibitory interface but not via the GAP interface, whereas RagB^{long} inhibits GATOR1 via the GAP interface.

In sum, these in vitro GAP assays recapitulate the regulation we see in vivo in the cell and confirm they are direct.

5. Statistics in Figures 1g, S1f,h and S2c are based on fields of view. This can hardly be considered independent biological experiments. Statistical analysis should be performed on at least 3 independent experimental replicates and tested for significance.

We have now changed our statistics according to the reviewer's suggestion, so that each data point now represents the average of multiple fields of view from one independent experiment, and several independent experiments are shown (ED Fig.

3b,d; ED Fig. 4b, h). The original conclusions are confirmed.

With respect to Fig 1g, it is unclear what exactly is meant by mTOR signal overlapping with LAMP2 Pearson correlation would be one way to assess this.

Even in nutrient-rich conditions, mTOR consists of two separate pools with very different spatial distribution: one highly concentrated on lysosomes and one diffusely cytosolic. In immunostainings, these two pools give rise to the typical appearance of very bright spots, corresponding to the high concentrations reached on the surface of lysosomes, on a "background" of much lower intensity signal coming from the cytosolic mTOR. Since Pearson's correlation coefficient does not fully capture this aspect of the mTOR biology, we performed an object-based colocalization analysis using CellProfiler. We segmented the mTOR staining using an Otsu-based adaptive thresholding algorithm to separate the bright spots of lysosomal mTOR from the cytosolic one. Using a similar approach, we then segmented the LAMP2 staining to identify LAMP2-positive objects. Finally, we calculated the percentage of the area of the LAMP2-positive objects covered by the lysosomal mTOR. We have now explained this quantification approach more clearly in the corresponding material and methods paragraph.

6. The manuscript falls short of providing compelling physiological data to support key functions of RagB isoforms in the brain or primary tissue. For example, mTORC1 hyperactivity is strongly implicated in epilepsy. Is RagB overexpressed in some inherited forms of epilepsy? Given that the metabolism of neurons and astrocytes is intimately linked, most importantly via the glutamine/ glutamate cycle that impinges on mTORC1 activity, it is surprising that no efforts are made to understand the physiological relevance of the claimed mechanism and the roles of the different RagA/B isoforms in this context.

15What are the consequences of RagB inactivation in primary neurons with respect to mTORC1 signaling, cell and neurite outgrowth, and neuronal excitability (all known to be regulated by mTORC1)?

The reason we did not look at how the glutamine/glutamate cycle impinges on mTORC1 activity in this context is that we are studying the differential activity of RagA versus RagB proteins, whereas glutamine is known to regulate mTORC1 via a Ragindependent mechanism (Jewell et al., Science, 2015).

As mentioned above in the general comments, these issues raised by the reviewer are indeed interesting but not within the scope of our study, whose main focus is the dissection of the molecular mechanisms that distinguish the RagB isoforms from RagA in the mTORC1 pathway. It is well established that the regulation of mTORC1 is a physiologically relevant process in all cell types including neurons, where it is involved in differentiation from neural stem cells, circuit formation, synaptic plasticity, and neurite outgrowth, only to name a few, and a very large number of pathological conditions (Lipton and Sahin, Neuron, 2014). We thus anticipate that the mechanism we uncover here will be relevant for all these multiple aspects of neurobiology that have been linked to mTORC1 signaling. Clearly, however, probing how these mTORC1-dependent processes are affected by the mechanisms we uncover here cannot be done in the context of our study and will require many additional projects to be performed.

Nonetheless, to meet the reviewer's request, we have now analyzed dendritic growth in response to synaptic activation in primary neurons, as an example of an important physiological process affected by the mechanism we uncover, and found that, as expected, depletion of RagB, but not RagA, blunts this process. These data are now shown in Fig. 6g.

7. The ratio of RagB/A in Neuro2a appears to be quite different from that in primary neurons (Fig. 6d-f), likely because these are tumor-derived. Hence, the conclusions derived from Neuro2a experiments cannot be transferred to primary neurons or to the brain.

We now provide additional data on primary neurons in Fig. 6e-g and ED Fig. 10b-g, showing that

- mTORC1 in neurons is resistant to amino acid removal,
- this is in part RagB-dependent (i.e. RagB, but not RagA, knockdown causes mTORC1 activity to drop in a low amino acid condition), and

- as expected, when RagB, but not RagA, is knocked down this impairs dendritic growth, assayed in a standard medium that contains low amino acids –10% the concentration of MEM.

8. Essentially all conclusions rest on pS6K1 levels as a readout for mTORC1 activity. Key conclusions should be confirmed for other substrates such as p4E-BP1 or pULK1. This might be especially interesting in light of recent findings showing that mTOR substrate phosphorylation (for example S6K1 vs TFEB) is differentially controlled by different regulators of the pathway (Napolitano et al, Ballabio Nature 2020). It is conceivable that RagB affects only a subset of mTORC1 substrates but not others in a given tissue.

We now provide improved pTFEB blots, as requested by the reviewer in point 13, together with additional data on p4EBP1 in Fig. 1e-h and ED Fig. 1h-k.

Our data suggest that RagB exerts a more global effect on mTORC1 activity, based on:

- (1) persistent lysosomal localization upon amino acid starvation (ED Fig. 4a-b, g-h) and
- (2) similar patterns of phosphorylation of three typical mTORC1 substrates (S6K1, TFEB, 4EBP1) which drop rapidly upon amino acid starvation in control cells (ED Fig. 2a-b) but persist when RagB is expressed (Fig. 1e-h).

In contrast, we found that ULK1 phosphorylation in wild-type cells generally responds poorly to complete amino acid starvation (ED Fig. 2a-b), at least within the timeframe of our experiments, and therefore we did not test it further.

9. The authors often capitalize on GTP locked variants of the Rags. As an important control the interaction of the different WT RagA/B isoforms with the Ragulator GEF complex in cells should be assessed to make sure that the effects seen are not due to different modes of Ragulator-mediated activation.

The point raised by the reviewer here is a valid one, in that the functional differences between RagA/B could also be due to differential activation by their GEFs, such as Ragulator. We have now added the suggested experiment (ED Fig. 8e-f) which shows that RagA and RagB^{short} interact comparably with

Ragulator (p18), while RagB^{long} interacts more strongly than the other two Rag isoforms, consistent with its poor binding to GTP, since the Rag/Ragulator interaction is stronger when the Rags are not bound to GTP (Bar-Peled, Cell, 2012). Worth noting here is also that, as we show in the GAP assay in ED Fig. 9b, RagB^{long} does not hydrolyze GTP, so a GEF-dependent mechanism is anyway not relevant for this isoform. These data corroborate our findings and indicate it is unlikely that differential Ragulator binding is the underlying mechanism differentiating RagA from RagB function.

Minor points:

10. LAMP2 staining seems to be affected by RagA/B or RagC/D loss in Fig. S1e,g.

Indeed, it is possible that the distribution and/or size of lysosomes is altered in the double knockouts. The total mass of lysosomes however is not affected, based on the LAMP2 blots we performed as part of lyso-IP experiments (ED Fig. 3e, whole cell lysate samples). We also looked at mTOR localization on lysosomes using lyso-IPs and these results using a complementary method are consistent with the immunostaining results mentioned here.

11. Fig S1: pS6K1 levels of DKO clones are well above those of Ctrl in starved cells. How is this explained?

This is indeed a very interesting aspect. Despite their importance for mTORC1 activation, loss of all Rags results in only a partial drop in mTORC1 activity (ED Fig. 1a-f). This was already previously reported by us (Demetriades et al., Cell, 2014) and others (Jewell et al., Science, 2015). The residual mTORC1 activity has been attributed to an Arf1-dependent mechanism that allows mTOR to localize to lysosomes in the absence of the Rags. Consistent with this possibility, although we see an almost complete loss of mTOR localization on lysosomes in immunostainings, lyso-IP experiments indicate that a small amount of mTOR and Raptor is still retained on lysosomes also in double knockout cells (ED Fig. 3e-g, ED Fig. 4c-d). One possibility is that such Rag-independent mechanism is intrinsically less responsive to changes in amino acid levels. In addition to this possibility, the inactive conformation of the Rag GTPases has been shown to not only release mTORC1 from the lysosome, but also to actively recruit factors that inactivate mTORC1, such as the TSC complex (Demetriades et al., Cell, 2014). In the absence of all Rags, such mechanisms of mTORC1 inactivation would also be lost. Interesting in this respect is also a study showing that knockdown of the Rags can paradoxically rescue the low mTORC1 activity of cells depleted of GATOR2 (which inhibits GATOR1) (Yang et al., Dev. Cell, 2020).

12. Blots for HA-tagged Rags in the lysate are often missing (e.g. Figure 2a, Figure 3b,c,h, Figure 4c,f, Figure S1d, Figure S3c) and should be added to the figure.

We respectfully disagree. The amount of target protein coimmunoprecipitating with the Rags depends on the amount of immunoprecipitated Rags and on the level of the target protein available for binding in the whole cell lysate, but not on the Rag levels in the whole cell lysate. Along this line of reasoning, we do analyze the Rag levels in the whole cell lysate when we immunoprecipitate another protein and look at coimmunoprecipitating Rags, for instance in Fig. 4d-e.

13. P-TFEB blots are of low quality (especially in Figures 1d,e, S1b,c). Better data are needed.

We have now repeated these blots (Fig. 1e, ED Fig. 1h) and hope the reviewer is satisfied with their improved quality.

14. Do cortical neurons used for the experiments in Fig 6d,e contain astrocytes? Again, no serious quantifications of the time course of mTORC1 inactivation and, most importantly, its dependence on RagB isoforms are presented.

Since we did not use cytostatics in our experiments with DIV5 neurons, it is possible that contaminating astrocytes are present. However, given the rather early time point analyzed, they are likely to represent a very limited contamination that does not affect our conclusions. In contrast, we did use cytostatics (AraC) in our new experiments with DIV10 neurons to prevent glial overproliferation. We have now added the quantification of all our replicates of the time course with DIV5 neurons mentioned by the reviewer (now in ED Fig. 10b-c) and performed statistics. Due to inefficient knockdown in DIV5 neurons, we have instead addressed the aspect of RagB dependence using DIV10 neurons (Fig. 6e-f). All these data confirm our conclusions that (1) mTORC1 activity in neurons is strikingly resistant to amino acid removal - please note these are 24h amino acid starvations, rather than the usual 20 or 60 minute starvations used on the other cell lines - and (2) this is at least partly due to expression of the RagB isoforms.

15. The order of panels should be consistent throughout the paper. Figure 5 b and c; Figure 6 e and f should be swapped

We have tried to keep the order of the panels consistent in our revised figures, as suggested by the reviewer.

16. Figure 5c,d,e: please indicate that RagABKO cells were used.

(Now Fig. 5a-h). We have added this.

17. The authors should explain why they overexpress FLAG-S6K1 in some experiments to assess pS6K1 levels but analyze the endogenous protein in others.

We regret not making this clear enough in our original manuscript. We have used cotransfected FLAG-S6K1 to assess mTORC1 activity upon transient transfection of proteins, for instance GATOR1 in Fig. 2a-d or inactive Rags in ED Fig. 5c-f, while we used endogenous S6K1 in all other cases, i.e. knockout cells or stably-transfected cell lines, for instance in Fig. 1e-f, 1k-p. Our reasoning is that upon transient transfection not all cells are transfected, so by looking at the phosphorylation of the co-transfected FLAG-S6K1 we can have a more reliable indication of mTORC1 activity just in the transfected cells. In contrast, in knockout cells or stably transfected cells all cells are depleted of the protein of interest or overexpressing the protein of interest, respectively, so phosphorylation of endogenous S6K1 reflects the effect of the protein of interest on the mTORC1 pathway. This approach is commonly used in the mTOR field (see for instance: Sancak et al., Science, 2008; Menon et al., Cell, 2014; Shen et al. Mol. Cell, 2017; Rogala, Science, 2019). We have now adjusted our figure legends to make this more clear.

Reviewer #2

Figlia et al. investigate the role of RagB in mTORC1 resistance to nutrient starvation. Specifically, the authors look at the mechanisms involved in the neurons. There are multiple concerns prior to publication. However, the mechanistic data is pretty strong.

We thank the reviewer for appreciating our work. We have added a substantial amount of new data to the manuscript (90 figure panels) and hope to have addressed all the concerns raised by the reviewer. In particular, we have now further corroborated our mechanistic model using in vitro assays with purified proteins (Fig. 4h-j, ED Fig. 7a-e, ED Fig. 9a-f) and added data on potential physiological implications of the mechanisms we uncover (Fig. 6e-g, 6q). There were multiple figure panels where the reviewer pointed out that it was difficult to see the differences in mTORC1 activity we were describing. We have now added quantifications of multiple biological replicates to all of our immunoblots, which make the differences more visible, and confirm they are statistically significant. We hope the reviewer finds this revised manuscript significantly improved.

1. It is not clear why there is still high S6K1 phosphorylation in the RagAB KO and RagCD KO cell lines? One would anticipate a significant decrease. Is this a clonal effect? Does knockdown (siRNA) of RagA and RagB or RagC and RagD have a high level of S6K1 phosphorylation under nutrient deficient conditions?

We do not think this is a clonal effect, since we see it with both RagABKO and RagCDKO cells and with two distinct clones each. This observation was already previously reported by us (Demetriades et al., Cell, 2014) and others (Jewell et al., Science, 2015) in both Drosophila and human cells, with both knockdowns and knockouts. Hence it is a real phenotype of Rag loss-of-function cells. The presence of residual mTORC1 activity in double knockouts during amino acid starvation can be due to two possible mechanisms, either alternatively or in combination:

- (1) It has been shown that in the absence of all Rag proteins an Arf1-dependent mechanism takes over and allows mTOR to still partially localize to lysosomes (Jewell et al., Science, 2015), accounting for the residual mTORC1 activity in the double knockouts. Although we see an almost complete loss of mTOR localization on lysosomes in immunostainings, lyso-IP

21experiments indicate that indeed a small amount of mTOR and Raptor is still retained on lysosomes also in double knockout cells (ED Fig. 3e-g). It is possible that this Rag-independent mechanism does not respond to changes in amino acid levels as well or as rapidly as the Rag-dependent one;

- (2) In addition to this possibility, the inactive conformation of the Rag GTPases has been shown to not only release mTORC1 from the lysosome, but also to actively recruit factors that inactivate mTORC1, such as the TSC complex (Demetriades et al., Cell, 2014). In the absence of all Rags, such mechanisms of mTORC1 inactivation would then also be lost.

We have added these explanations to the Discussion.

2. Sup. Fig. E-H. Lyso-IP (PMID: 29074583) experiments should be performed here showing that Raptor and mTOR are not at the lysosome in RagABKO and RagCDKO cells. Also, in RagAB KO cells are RagC and Rag D still at the lysosome. In RagCD KO cells are RagA and RagB (short and long) still at the lysosome.

We thank the reviewer for this suggestion. Indeed, these experiments (ED Fig. 3e-k) have helped us clarify the origin of the residual mTORC1 activity in our double knockouts. As discussed in the point above, although we see an almost complete loss of mTOR localization on lysosomes by immunostainings, the lyso-IP experiments (which are more sensitive) show that a residual amount of mTOR and Raptor is still on the lysosomes. These results are in line with previous observations in MEFs that indicate that a Rag-independent mechanism can partially maintain mTORC1 on lysosomes when the Rags are knocked out (Jewell et al., Science, 2015). Additionally, we see that, in the absence of A/B, RagC/D do not localize on lysosomes and viceversa (ED Fig. h-k). Please note that in our quantifications we normalized the protein levels in the lyso-IP to their levels in the whole cell lysate – ie we take into account the fact that C/D levels are reduced in the whole cell lysate of A/B knockouts and viceversa.

3. If RagB long can maintain mTORC1 activity but it's not at the lysosome, where is mTORC1 in the cell? Does RagB long interact with mTORC1 to the same extent as RagB short (Wild-type RagB forms)?

To address this issue, we have performed both lyso-IP experiments with cells expressing RagA, RagB^{short}, or RagB^{long} (ED Fig. 4c-d) and colP experiments with Raptor and wild-type RagA/B (ED Fig. 4e-f) :

The lyso-IP experiments show that all three RagA/B isoforms are able to localize to the lysosome to a comparable extent. However, while the localization of mTOR on lysosomes is increased in cells expressing RagA or RagB^{short}, as expected, we do not see more mTOR on lysosomes in cells expressing RagB^{long}, as compared to the FLAG-metap2 negative control, consistent with our previous results. Nonetheless, some mTOR is present on lysosomes in RagABKO cells as well as RagABKO+

RagB^{long} cells.

The colP experiment shows that RagB^{long} is still competent to bind Raptor, although much more weakly than the other two isoforms, similar to what we saw before using mutant RagA/B (ED Fig. 5a-b). Hence, this weak binding is likely sufficient to cause a mild increase in mTORC1 activity.

4. Fig. 1H. Deletion of RagA doesn't look like it increases RagB (short and long) expression as mentioned on Page 9.

We thank the reviewer for pointing this out. Indeed, this is only visible in original Suppl Fig 2A (now Fig. 1i), so we have changed the figure citation to only reference this panel.

5. Fig. 2C. The decrease looks the same when overexpressing GATOR1 in cells expressing RagA and RagB short. Fig 2D. Should have error bars with replicate blots.

We thank the reviewer for pointing this out. We have now added quantifications of this blot (now in Fig. 2a-b), together with quantifications of all other blots in our manuscript, to make this more visible. Consistent with our other data, the difference between RagA and RagB^{short}-expressing cells in response to GATOR1 is significant only upon transfection of the lower amounts of GATOR1 (5ng, Fig. 2b). It is worth noting, that in this experiment we overexpress GATOR1 in cells that are already expressing endogenous GATOR1. Indeed, when we do a similar titration in cells knockout for DEPDC5, so as to reconstitute functional GATOR1 levels that go from 0 to maximum endogenous amount, the difference between RagA and RagBshort is more evident

(Fig. 3e-f).

6. Fig. 6D. Is mTORC1 signaling higher in neurons due to an elevated RagB? What about RagC/D?

We have now performed knockdowns of RagA vs RagB in neurons at day in vitro 10 (Fig. 6e-f) (knockdown at day in vitro 5 was technically problematic). Interestingly, we see that, at this later stage of maturation, neurons become even more resistant to amino acid starvation, with mTORC1 activity decreasing only slightly despite 24 hours of complete amino acid starvation (Fig 6e-f, ED Fig. 10d-e). Knockdown of RagB, but not RagA, causes mTORC1 activity to drop significantly upon starvation, as compared to the unstarved condition, if combined with synaptic activation through bicuculline, probably because protein synthesis – and thus amino acid usage – is higher when neurons are active (Fig. 6e-f). Other mechanisms are also likely to be involved, since the decrease in mTORC1 activity is not complete. Importantly, we show that this effect is physiologically relevant, because dendritic growth is blunted when neurons are depleted of RagB, but not RagA (Fig. 6g). In sum, the resistance of mTORC1 to amino acid removal in neurons is partly RagB-dependent and partly dependent on other mechanisms.

Regarding RagC vs D: Based on our data with HEK293T cells, RagC vs D appear to differentially affect the phosphorylation of only a subset of mTORC1 targets, namely TFEF, but not the global response of mTORC1 to amino acid starvation. Hence, it is unlikely to be causing this effect. The ratio of RagC vs D expression in brain also does not appear to be an outlier compared to other tissues (Fig. 1a).

Fig. 6G. Include error bars of replicates.

We have now obtained Neuro-2a cells knockout for either RagA or RagB, hence we replaced Fig. 6g in the original manuscript with analogous experiments using these knockouts, which can be found in ED Fig. 9j-m, along with their quantifications. As in the previous experiment using knockdowns, here too we see that loss of RagA, but not RagB, renders mTORC1 activity resistant to low amino acids, although the basal levels are lower than in controls, while RagB knockouts respond to amino acid restriction and have low mTORC1 activity both basally and during starvation.

7. Fig. 6E. Doesn't appear to see a change. Perhaps add graph of phosphorylation of S6K with error bars of the replicate.

24We have now quantified all our replicates of this experiment (now in ED Fig. 10b) and performed statistics (ED Fig. 10c), which renders the difference between MEFs and neurons more apparent.

We now also include assays on more mature neurons (DIV10) in Fig. 6e-f and ED Fig. 10d-e which show that mTORC1 activity in neurons is very resistant to amino acid removal, despite us starving them of amino acids for 24 hours (instead of the 20-60 minutes needed for other cell lines).

8. Fig. 6I. Add error bars of replicates.

We have now replaced this panel with analogous experiments using knockout instead of knockdown Neuro-2a cells. The new panels can be found in ED Fig. 9j-m. (The statistical analysis on the replicates of the knockdown cells also showed significance, but we felt the knockout cells were sufficient to make the point.)

9. Fig. 6H-I. Doesn't appear to be a difference? Authors state say a decrease in mTORC1 activity after the knockdown of RagB. Error bars and replicates.

Please see our response to Point 8 above. We have replaced these knockdown data with knockout data, including replicates, quantifications, and statistical analyses which make the differences more visible and confirm their significance.

10. Fig. 6N. Would the authors expect an increase in RagC/D in EFO21 cells?

We do not see a major change in RagC levels, while RagD levels are even reduced in EFO21 cells as compared to HEK293T cells (Fig. 6L).

11. Fig. 6Q-R. Do not see a difference between siLuciferase and siRagB in terms of mTORC1 activity? Error bars of replicates. Same with 6P.

25We have now quantified all our replicates for these experiments, which can now be found in Fig. 6m-p.

12. What is the physiological reason to have higher levels of RagB short in EFO21 cells?

We thank the reviewer for raising this issue. As tumor cells often experience low nutrient levels in their microenvironment, due to poor vascularization and increased utilization, relative resistance of mTORC1 to lower nutrients could allow cancer cells to maintain the anabolic pathways controlled by mTORC1 active and thus continue growing. Indeed, somatic mutations of the nutrient-sensing mechanisms of mTORC1 have been described in some glioblastoma and ovarian cancers (Bar-Peled, Science, 2013) and in up to 17% of follicular lymphomas (Okosun, Nat. Genet., 2016). This suggests that there could be a sort of threshold of nutrient levels below which the nutrient-sensing machinery inactivates mTORC1, but which would still be high enough to support growth if mTORC1 could be kept active. High expression of RagB could be an additional route that allows cancer cells to exploit this mechanism. Of note, like some of the tumors described to harbor mutations in GATOR1 components (BarPeled, Science, 2013), also EFO21 are ovarian cancer cells.

13. For the cell lines used in Fig 6O-Q, have the authors thought about using them in a xenograft model to assess tumor size. In terms of biology, you would anticipate higher mTORC1/tumor size (in siRagA cells) compared to control cells. And lower mTORC1/tumor size (in siRagB cells) compared to control cells.

This would indeed be a very interesting direction to follow up. Although in this revision we focused our analysis more on the physiological implications of RagB expression for neurons, we now also show that RagA vs RagB have different impacts on protein synthesis rates in EFO21 cells (Fig. 6q). Specifically, we see that knockdown of RagB decreases OPP incorporation, which could cause less tumor growth, as anticipated by the reviewer. Since we did a very large number of experiments to address the concerns regarding the molecular mechanism of this mTORC1 regulation, a follow-up study will be needed to look at the relevance for tumor growth in xenograft models.

Minor

261. Fig. 1D. is not in text.

We thank the reviewer for noticing that, we have now fixed it in our revised manuscript.

2. RagB long should be included in Fig. 2B.

We have now performed an analogous experiment using also RagB^{long}, which can be found in ED Fig. 5e-f.

3. Manuscript hard to follow. Flow and organization

We have tried to improve the flow and organization of our manuscript. We hope the reviewer finds the manuscript improved.

Reviewer #3

Figlia et al. report that the RagB^{short} and RagB^{long} isoforms have distinct mechanisms through which they regulate the activity of mTORC1, with RagB^{short} inhibiting Gator1 via DEPDC4, and RagB^{long} inhibiting Gator1 via Nprl2/3. RagB and RagA are highly homologous (almost identical in the GTPase and CRD domains), and have been considered interchangeable in terms of their function. RagB has a 33 aa N-terminal extension that is absent in RagA, and RagB^{long} has an additional 28 aa in the switch I region. Figlia et al. knocked out both RagA and RagB, and then re-expressed a single RagA or B isoform. They also did this with RagC/D. There are many strengths in this work, including one of the first attempts to define differences between RagA and RagB. Concerns, as detailed below, include the system itself (especially as it relates to the two RagB isoforms), the non-physiologic expression levels, and the lack of quantitative analysis of the many Western blots. It would appear that some conclusions are based on a single Western, rather than densitometric analyses of triplicate blots.

27We thank the reviewer for the supportive comments and constructive criticisms. We have now added a significant amount of new data (90 figure panels), including (1) *in vitro* GTPase assays using purified proteins that have corroborated our proposed mechanism, (2) physiological data on RagA vs RagB knockdown in primary neurons, which show that they have differential effects on dendrite growth, consistent with their effects on mTORC1 activity, and (3) quantifications of all the replicates of our blots and their statistical analysis. We hope these new data address satisfactorily all the reviewer's concerns.

1. Significant concerns about the system itself weaken this work. These include the level of expression of the reconstituted cells vs. endogenous and the interpretation of cells in which RagB long is expressed alone, since it would appear from Figure 1B that expression of a single RagB isoform does not occur physiologically. It is also a concern that the HEK293 cells, used as a model here, express predominantly RagA, and therefore could lack some (unknown) factors required for RagB-dependent cellular responses. Supp Fig 1 shows that there is ~800 fold higher expression of RagA at the mRNA level. There is no evidence that RagB long is expressed by HEK293 cells. This leads to fundamental concerns about how to interpret these otherwise interesting data.

As detailed below, our approach to dissecting these complex molecular mechanisms consists of two parts: first, studying each Rag in isolation to understand what it can/cannot do, and then secondly putting them together to understand what the combinations do:

- (1) First, we wanted to study the specific properties of each RagA/B isoform, which can be done only by taking them apart. We use therefore HEK293T cells expressing individual RagA/B isoforms as a mechanistic tool, not to draw conclusions about the physiological relevance of the effects we see. This is conceptually analogous to studying recombinant proteins in a test tube, except we do it in a cellular context with all the consequences of subcellular localization and the presence of all the other proteins in the cell. For this purpose, we undertook two complementary approaches. We either knocked out both RagA and RagB in HEK293T cells and then reconstituted the system with single Rag isoforms (which the reviewer mentions in his/her criticism), or we simply knocked out one single Rag isoform at a time in cells so as to leave only the other isoform, without any overexpression. These two approaches led to the same conclusions about the differences in RagA versus RagB function.

Importantly, we now also add to these data a substantial amount of new *in vitro* data with purified RagA/B and GATOR1 proteins (ED Fig. 7a-e, ED Fig. 9a-f, Fig. 4h-j) that show that the mechanisms we uncovered are direct, i.e. they do not require any celltype specific factor to be able to work. It is also worth emphasizing here that actually an important conclusion of our mechanistic characterization is that the primary function of RagB^{long} does indeed require that also the other Rags are present, because RagB^{long} interacts itself quite poorly with mTORC1, but rather works to enhance signaling through the other RagA/B isoforms by inhibiting GATOR1.

- (2) In the second part, following exactly the reviewer's logic, we study the relevance of the mechanisms we uncovered in the first part in a physiological context where all RagA/B isoforms are expressed together, either exogenously or endogenously. Specifically, we studied this
- in HEK239T cells simultaneously transfected with all three RagA/B isoforms (Fig. 5a-i),
 - in Neuro-2a cells (a neuroblastoma cell line that endogenously expresses all three isoforms, ED Fig. 9j-m),
 - in EFO21 cells (an ovarian cancer cell line that endogenously expresses high levels of RagB^{short} and a small amount of RagB^{long}, Fig. 6m-q), • and now in primary neurons, which endogenously express all three isoforms (Fig. 6d-g, ED Fig. 10b-e).

In all cases, we confirmed that the presence of the RagB isoforms makes cells more resistant to low amino acids, consistent with the effects we see in HEK293T cells and with the mechanism of GATOR1 inhibition. Importantly, we now also show with knockdown experiments that the resistance of neurons to low nutrients depends at least in part on RagB (Fig. 6e-f) and that depletion of RagB but not RagA blunts dendrite growth in primary neurons (Fig. 6g), consistent with the effects of RagB knockdown on mTORC1 activity and with mTORC1 being a positive regulator of this process.

Please note that there seems to be a misunderstanding in one of the reviewer's comments - based on the RNA data mentioned by the reviewer (now in ED Fig. 1g), in HEK293T cells RagA is expressed roughly 7 times more than RagB (907.4 vs 123.5 copies per ng RNA), not 800 times more.

Currently, we have also started studying RagA vs RagB in adult neural stem cells, which – like neurons – express all three RagA/B isoforms:

We have recently obtained neural stem cells knockout for RagA – thereby leaving only the RagB isoforms – and also in this case we see that mTORC1 activity becomes resistant to low amino acids:

Hence, we believe we have collected enough evidence to conclude that the core functions of RagA/B in the mTORC1 pathway do not depend on the cell type being studied, although it is certainly possible that some cell-specific factors could modulate the impact that such core functions have on the cell's physiology.

2. Figure 1D shows differential responses to Phospho-S6K and phospho-TFEB.

This has already been shown in (Napolitano et al., Nature 2020) that TFEB is phosphorylated by mTORC1 on the lysosome and therefore requires direct Rag binding to be recruited to the lysosome (hence it reads out both mTORC1 activity and Rag activity combined). In contrast, phosphorylation of S6K, which

is cytosolic, only reads out mTORC1 activity. For this reason, TFEB phosphorylation drops more strongly in the Rag knockout cells. Nonetheless, the key trends are the same, as can be seen in new Fig. 1e-h: both S6K and TFEB phosphorylation in the -aa condition are higher in the RagB-reconstituted cells compared to RagA-reconstituted cells (red stars Fig. 1f-g), and phosphorylation of both in the +aa condition is lower in the RagB^{long}-reconstituted cells compared to RagA and RagB^{short} (green stars Fig. 1f-g). Both results are statistically highly significant.

It is surprising that levels of p-S6K are relatively high in the RagABKO cells; this is commented on but with no hypothesis for why this is observed – it is actually higher than cells with RagA re-expressed after 15 min of aa addback. Could this reflect a technical problem with this system? How do the levels of the overexpressed proteins compare with endogenous levels? No endogenous RagA or B appears on the Western, suggesting that levels could be much, much higher than physiologic levels.

This does not appear to be a technical or a clonal effect, since we see it with both RagABKO and RagCDKO cells and with two distinct clones each.

The fact that basal pS6K is quite high in RagAB knockout cells is at first surprising based on a model mainly coming from the Sabatini lab that 1) the Rag proteins are the sole responsible mechanism for amino acid sensing in cells and 2) they only have an activating effect on mTORC1. However, this model is oversimplified and not completely correct. The elevated pS6K levels in the Rag knockout cells actually fit with what we (Demetriades et al., Cell, 2014) and others (Jewell et al., Science, 2015; Yang et al., Dev. Cell, 2020) have previously published:

- (1) we have observed a similar effect in both Drosophila and human cells also upon knockdown of the Rag proteins (Demetriades et al., Cell, 2014)
- (2) in Jewell et al., Science, 2015, MEFs double knockout for RagA and B were shown to still retain roughly 70% of mTORC1 activity, which the authors demonstrated was due to an Arf1-dependent mechanism that allows mTOR to still partially localize to lysosomes when Rag proteins are absent (Jewell et al., Science, 2015). Consistent with this, although we see a strong loss of mTOR localization on lysosomes in immunostainings (ED Fig. 3a-d), we have

now performed lyso-IP experiments that indicate that indeed a small amount of mTOR and Raptor is still retained on lysosomes also in our double knockout cells (ED Fig. 3e-g).

- (3) it was shown both in mammalian cells and in *Drosophila* that mTORC1 activity, which is low upon depletion of GATOR2 (a negative regulator of GATOR1), can be paradoxically increased by knocking down the Rag GTPases (Yang et al., *Dev. Cell*, 2020). This means that the Rag proteins must also be actively inhibiting mTORC1 in the -aa condition, and this inhibition is lost upon Rag loss-of-function (causing the elevated pS6K).

Indeed, we previously showed that in the -aa condition, the inactive conformation of the Rag GTPases not only release mTORC1 from the lysosome, but also actively recruit factors that inactivate mTORC1, such as the TSC complex (Demetriades et al., *Cell*, 2014). In the absence of all Rags, such mechanisms of mTORC1 inactivation are lost, causing elevated pS6K.

Hence, in sum, this is a real Rag loss-of-function phenotype. We have added these explanations to the Discussion.

Regarding the Rag levels: We have now repeated our blots with HEK293T cells expressing each RagA/B isoform (Fig. 1e-h) to include also a comparison with control cells, as requested by the reviewer. RagA/B levels are indeed higher than in control cells. However, we also see that the levels of RagC and RagD, which are decreased in RagABKO cells, are rescued back to control levels, and not higher, in cells reconstituted with each RagA/B isoform. This means that the levels of functional Rag heterodimers (A/B + C/D) is comparable between all these cell lines and control cells. Additionally, as discussed above, we also see analogous mTORC1 responses also when we knock out RagA vs RagB in cells that endogenously express all three RagA/B isoforms, without any overexpression.

3. The legend for this figure simply states that the cells were grown in "normal" media; the specific media should really be specified in the figure legend since this is critical to the interpretation (even if also in the methods section).

We thank the reviewer for pointing this out, we have now adjusted our figure legend accordingly.

4. Figure 1D and E (and other similar blots) should include quantitation of the changes in p-S6K, based on triplicate blots. The changes appear to be rather modest compared with the knockout cells, and it is challenging to know whether there are any true differences between the isoforms. The changes in TFEB phosphorylation are impossible to interpret. It would be of interest to include phospho-S6 in this figure.

We thank the reviewer for this comment. Generally, we now include quantifications of multiple independent biological replicates as well as statistical analysis for **all** immunoblots in the manuscript, including the ones mentioned here (now in Fig. 1e-h and ED Fig. 1h-k). This helps a lot to render the changes in mTORC1 activity more clear and to show that they are statistically significant. We hope the reviewer finds the manuscript substantially improved.

Regarding specifically the blots mentioned here, we have repeated these blots (now in Fig. 1e-h and ED Fig. 1h-k) to include also control cells. We also added blots of pTFEB/TFEB of improved quality, and blots for p4EBP1/4EBP1 as an additional readout of mTORC1 activity. All these show that cells expressing the RagB isoforms have higher mTORC1 activity in the -a.a. condition (red stars Fig. 1f-h) and that RagB^{long} does not reconstitute mTORC1 activity in the +a.a. condition to the same level as the other Rag isoforms (green stars, Fig. 1f-h). The statistical analyses show that these effects are statistically significant.

Although S6 is certainly another possible readout of mTORC1 activity, in a time course experiment with control cells (ED Fig. 2a-b) we see that its phosphorylation does not decrease much until one hour of complete amino acid starvation. Instead, phosphorylation of three direct mTORC1 substrates (S6K1, TFEB, 4EBP1) drops substantially already after 15 minutes of starvation. The slower dynamics of pS6 probably reflect the fact that S6 is not a direct substrate of mTORC1, but rather a substrate of an mTORC1 substrate (S6K). We prefer to use in this manuscript a shorter timepoint (30 min) to see direct effects on mTORC1 activity.

5. The last paragraph of the discussion about Figure 1 states that RagB long does not recruit mTOR to the lysosome as efficiently as the other isoforms but based on Fig 1G it does not recruit mTOR at the lysosomes at all.

This was indeed inaccurate wording, we thank the reviewer for pointing it out. Worth noting in this respect is that, although we do not see mTOR recruitment to the lysosome in immunostaining or in lysosomes (ED Fig. 4a-d), RagB^{long} seems to be still able to bind Raptor, albeit very weakly, as we see in colocalization experiments using either nucleotide-locked mutants or wild-type proteins (ED Fig. 4e-f, ED Fig. 5a-b).

6. The authors state at the beginning of the text section on Gator1 that “mTORC1 activity remains high in RagB expressing cells upon aa removal” but this is not really clear from Fig1. In Fig1D, the levels of activity (as judged by ph-S6K) in RagB short and RagB long cells upon aa withdrawal appear similar to the cells expressing the Flag vector.

We believe there could have been a misunderstanding here, caused by the fact that the original Figure 1D compared everything to RagABKO cells but didn't have control cells as a comparison (ie the Flag-vector cells were RagABKO). We now redid this Figure (now Fig. 1e-h) to include control cells, so the effects, as discussed in response to Point 4 above, should be clearly visible (red stars, Fig. 1f-h). In particular, cells expressing RagB^{short} or RagB^{long} have higher mTORC1 activity during amino acid starvation compared to control cells or cells expressing RagA.

7. Regarding Fig 2A, the authors state that the three Rag isoforms “release mTORC1 similarly when in the GDP conformation” but the results in Fig2A suggest that RagB long does not bind (or barely binds) Raptor even when in its active form.

(Now ED Fig. 5a-b). Indeed, as the reviewer points out, the main conclusion from this experiment is that in the GDP conformation none of the RagA/B isoforms is capable of binding Raptor, and hence of recruiting mTORC1 to the lysosome for activation. Whether RagB^{long} is releasing Raptor or was never binding it to start with is not so relevant here. We have now reworded this to read

“...none of the three RagA/B isoforms interacted with Raptor when not bound to GTP (ED Fig. 5a-b)...”

8. In Fig 2C, it is surprising that the authors needed to IP S6K to detect its phosphorylation; in Figure 1 and throughout the literature, S6K phosphorylation is readily detected in the cell lysate.

We regret not making this clear enough in our original manuscript. We have used cotransfected FLAG-S6K1 to assess mTORC1 activity upon transient transfection of proteins, for instance GATOR1 in Fig. 2a-d or inactive Rags in ED Fig. 5c-f, while we used endogenous S6K1 in all other cases, i.e. knockout cells or stably-transfected cell lines, for instance in Fig. 1e-f, 1k-p. Our reasoning is that upon transient transfection not all cells are transfected, so by looking at the phosphorylation of the co-transfected FLAG-S6K1 we can have a more reliable indication of mTORC1 activity just in the transfected cells. In contrast, in knockout cells or stably transfected cells all cells are depleted of the protein of interest or overexpressing the protein of interest, respectively, so phosphorylation of endogenous S6K1 reflects the effect of the protein of interest on the mTORC1 pathway. This approach is commonly used in the mTOR field (see for instance: Sancak et al., Science, 2008; Menon et al., Cell, 2014; Shen et al. Mol. Cell, 2017; Rogala, Science, 2019). We have now adjusted our figure legends to make this more clear.

While the authors place a great deal of emphasis on the observation that “only the highest levels of GATOR1” can decrease ph-S6K in the RagB long cells, it is clear from Fig1 that these cells have levels of ph-S6K that are comparable to the vector control cells (Fig 1D). So, it does not seem surprising that Gator1 does not decrease ph-S6K in these cells. Again, it is concerning that the authors show quantitation (Fig 1D) that appears to be based on a single experiment and lacks statistical analysis.

To address this issue, we have now provided quantifications of multiple biological replicates, as well as statistical analyses of all the blots mentioned here by the reviewer (Fig. 1e-h, Fig. 2a-b):

- Fig 1e-h now shows that indeed cells reconstituted with RagB^{long} have higher pS6K in the amino-acid starved condition compared to the control cells or cells reconstituted with RagA (red stars, Fig. 1f). This result is highly statistically significant ($p < 0.001$).
- Fig 2a-b shows low levels of GATOR1 (5ng Fig 2b) cause a stronger drop in pS6K in RagA-expressing cells compared to RagB^{short} expressing cells (red bar vs dark blue bar, $p < 0.05$). Compared to 0ng GATOR1, pS6K drops strongly with 5ng GATOR1 in the RagA cells (by roughly half, $p < 0.001$) whereas in the RagB^{long} cells pS6K does not show a statistically significant drop.

Additionally, we now add also an analogous GATOR1 titration experiment comparing FLAG-metap2 cells with RagB^{long}-expressing cells to make more clear the difference between these two cell lines (ED Fig. 6a-b).

We hope these new data and analyses make the results easier to observe.

9. Figure 2D seems to add little - here it is shown that GATOR1 overexpression decreases phospho-S6K in cells with RagA KO (that express a very small amount of RagB). The authors state that this is to test the hypothesis that "GATOR1 insensitivity is the cause of high mTORC1 activity in RagB expressing cells" but this does not make sense, since high mTORC1 is only seen in RagB long (Fig 2C/D).

(This figure is now Fig. 2c-d). We see that both RagB^{long} and RagB^{short} are resistant to GATOR1. This is supported by multiple pieces of data:

- pS6K levels are elevated in the -aa condition in RagB^{long} and RagB^{short} expressing cells (red stars, Fig. 1e-h).
- In vitro GAP assays show that RagB^{short} is resistant to GATOR1 due to binding of the GATOR1 inhibitory interface (ED Fig 7d-e)
- In vivo, RagB^{short} expressing cells are more resistant than RagA expressing cells to either GATOR1 titration (5ng in Fig. 2a-b, red stars) or to DEPDC5 titration (Fig. 3).

Hence, although the RagA KO cells express little RagB^{long} (main Fig. 1i), they do express RagB^{short}, which is why we wanted to test whether GATOR1 overexpression can rescue the phenotype.

10. The conclusion that differential resistance to GATOR1 is the "main functional distinction between RagB short and RagA" seems to be only partially supported by the data in Figure 2.

We now include quantifications of multiple biological replicates as well as statistical analyses to Figure 3 which make this conclusion more clear and confirm it is statistically significant. In Figure 3e-f one can see that GATOR1 reconstitution by expressing wildtype DEPDC5 causes a stronger drop in mTORC1 activity in RagA-expressing cells compared to RagB^{short}-expressing cells. This difference is gone in cells expressing mutant DEPDC5 which cannot bind the Rags specifically on the inhibitory interface (while the GAP

interface binding and GAP function of GATOR1 still work) (Fig. 3g-h, where there is no longer a statistically significant difference between the red and blue datapoints). This proves that the functional distinction between RagA and RagB^{short} is indeed caused via binding to the inhibitory interface of GATOR1.

Furthermore, the differences in mTORC1 activity between RagA- and RagB^{short} expressing cells are gone if we remove the GATOR1 subunit DEPDC5 (ED Fig. 6c-e) proving that the functional difference between RagA and RagB^{short} depends on GATOR1.

Nonetheless, we have now toned down this conclusion to “probably the main functional distinction”.

11. In Figure 3D, the authors tested mTORC1 activity (again by IP of S6K) with a mutant of DEPDC5 that does not bind the Rags.

Indeed – that is the point of this experiment. In this series of experiments (Fig. 3a-h, ED Fig. 7a-e) we ask whether the difference between RagA and RagBshort is due to binding to the inhibitory interface of GATOR1, which is mediated by its DEPDC5 subunit. While the Y775A mutation of DEPDC5 disrupts the binding through the inhibitory interface, it does not affect the binding to the Rags through the GAP interface of GATOR1, nor its GAP activity, both of which are mediated by the subunits Npr12/3 (Shen et al., Nature, 2018).

Decision Letter, first revision:

Dear Dr Teleman,

Thank you for your email asking us to reconsider our decision on your manuscript, "Brain-enriched RagB isoforms regulate the dynamics of mTORC1 activity via GATOR1 inhibition". We are always willing to hear the authors' perspective, but we must first prioritize decisions on new submissions. We appreciate your patience while we considered this appeal.

I have now discussed your manuscript and the referees' comments and your rebuttal in detail with my colleagues, and we would be willing to reconsider a revised manuscript provided the files below are provided with the revised manuscript (source files, reporting summary, and editorial policy checklist, see below), and that nothing similar is accepted for publication at Nature Cell Biology or published elsewhere in the meantime.

37Please note that, as we strive to limit all manuscripts to one round of major experimental revision, we will be looking for strong reviewer enthusiasm and support to move forward with the study.

Please pay close attention to our guidelines on statistical and methodological reporting (listed below) as failure to do so may delay the reconsideration of the revised manuscript. In particular please provide:

- a Supplementary Figure including unprocessed images of all gels/blots in the form of a multi-page pdf file. Please ensure that blots/gels are labeled and the sections presented in the figures are clearly indicated.
- a Supplementary Table including all numerical source data in Excel format, with data for different figures provided as different sheets within a single Excel file. The file should include source data giving rise to graphical representations and statistical descriptions in the paper and for all instances where the figures present representative experiments of multiple independent repeats, the source data of all repeats should be provided.

On resubmission please provide the completed Editorial Policy Checklist (found here <https://www.nature.com/documents/nr-editorial-policy-checklist.pdf>), and Reporting Summary (found here <https://www.nature.com/documents/nr-reporting-summary.pdf>). This is essential for reconsideration of the manuscript and these documents will be available to editors and referees in the event of peer review. For more information see below. Please also ensure that the presentation of statistical information in the revised submission complies with Nature Cell Biology's statistical guidelines (see below).

Please use the link below to submit the complete manuscript files and please include a point-by-point response to the complete reviewer comments, verbatim as provided in their reports.

[REDACTED]

Please let us know how you wish to proceed and when we can expect your revised manuscript. Thank you for your continuing interest in NCB.

With kind regards,

Melina

Melina Casadio, PhD
Senior Editor, Nature Cell Biology
ORCID ID: <https://orcid.org/0000-0003-2389-2243>

GUIDELINES FOR EXPERIMENTAL AND STATISTICAL REPORTING

REPORTING REQUIREMENTS – To improve the quality of methods and statistics reporting in our

38papers we have recently revised the reporting checklist we introduced in 2013. We are now asking all life sciences authors to complete two items: an Editorial Policy Checklist (found here <https://www.nature.com/documents/nr-editorial-policy-checklist.pdf>) that verifies compliance with all required editorial policies and a reporting summary (found here <https://www.nature.com/documents/nr-reporting-summary.pdf>) that collects information on experimental design and reagents. These documents are available to referees to aid the evaluation of the manuscript. Please note that these forms are dynamic 'smart pdfs' and must therefore be downloaded and completed in Adobe Reader. We will then flatten them for ease of use by the reviewers. If you would like to reference the guidance text as you complete the template, please access these flattened versions at <http://www.nature.com/authors/policies/availability.html>.

3rd June 2022

Dear Dr. Teleman,

Thank you for submitting your revised manuscript "Brain-enriched RagB isoforms regulate the dynamics of mTORC1 activity via GATOR1 inhibition" (NCB-T46214B). It has now been seen by the original referees and their comments are below. The reviewers find that the paper has improved in

39revision, and therefore we'll be happy in principle to publish it in Nature Cell Biology, pending minor revisions to comply with our editorial and formatting guidelines.

****Please note that the current version of your manuscript is in a PDF format; could you please email us a copy of the file in an editable format (Microsoft Word or LaTeX)? We can not proceed with PDFs at this stage. Thank you in advance for your attention to this point.****

After receiving this file, we will start performing detailed checks on your paper and will send you a checklist detailing our editorial and formatting requirements in about a week. Please do not upload the final materials and make any revisions until you receive this additional information from us.

Thank you again for your interest in Nature Cell Biology. Please do not hesitate to contact me if you have any questions.

Sincerely,

Melina

Melina Casadio, PhD
Senior Editor, Nature Cell Biology
ORCID ID: <https://orcid.org/0000-0003-2389-2243>

Reviewer #1 (Remarks to the Author):

The authors have done an impressive job in revising their Ms and in addressing all previous questions and concerns in full. I strongly support publication of the revised Ms in Nat Cell Biol.

Reviewer #2 (Remarks to the Author):

Addressed concerns

Reviewer #3 (Remarks to the Author):

all concerns have been thoroughly addressed

15th June 2022

Dear Dr. Teleman,

40Thank you for your patience as we've prepared the guidelines for final submission of your Nature Cell Biology manuscript, "Brain-enriched RagB isoforms regulate the dynamics of mTORC1 activity via GATOR1 inhibition" (NCB-T46214B). Please carefully follow the step-by-step instructions provided in the attached file, and add a response in each row of the table to indicate the changes that you have made. Please also check and comment on any additional marked-up edits we have proposed within the text. Ensuring that each point is addressed will help to ensure that your revised manuscript can be swiftly handed over to our production team.

We would like to start working on your revised paper, with all of the requested files and forms, as soon as possible (preferably within one week). Please get in contact with us if you anticipate delays.

In recognition of the time and expertise our reviewers provide to Nature Cell Biology's editorial process, we would like to formally acknowledge their contribution to the external peer review of your manuscript entitled "Brain-enriched RagB isoforms regulate the dynamics of mTORC1 activity via GATOR1 inhibition". For those reviewers who give their assent, we will be publishing their names alongside the published article.

Nature Cell Biology offers a Transparent Peer Review option for new original research manuscripts submitted after December 1st, 2019. As part of this initiative, we encourage our authors to support increased transparency into the peer review process by agreeing to have the reviewer comments, author rebuttal letters, and editorial decision letters published as a Supplementary item. When you submit your final files please clearly state in your cover letter whether or not you would like to participate in this initiative. Please note that failure to state your preference will result in delays in accepting your manuscript for publication.

Cover suggestions

As you prepare your final files we encourage you to consider whether you have any images or illustrations that may be appropriate for use on the cover of Nature Cell Biology.

Nature Cell Biology has now transitioned to a unified Rights Collection system which will allow our Author Services team to quickly and easily collect the rights and permissions required to publish your work. Approximately 10 days after your paper is formally accepted, you will receive an email in providing you with a link to complete the grant of rights. If your paper is eligible for Open Access, our Author Services team will also be in touch regarding any additional information that may be required to arrange payment for your article.

Please note that *Nature Cell Biology* is a Transformative Journal (TJ). Authors may publish their research with us through the traditional subscription access route or make their paper immediately open access through payment of an article-processing charge (APC). Authors will not be required to make a final decision about access to their article until it has been accepted. Find out more about Transformative Journals

Please use the following link for uploading these materials:
[REDACTED]

Best regards,

Nyx Hills
Staff
Nature Cell Biology

On behalf of

Melina Casadio, PhD
Senior Editor, Nature Cell Biology
ORCID ID: <https://orcid.org/0000-0003-2389-2243>

Reviewer #1:

Remarks to the Author:

The authors have done an impressive job in revising their Ms and in addressing all previous questions and concerns in full. I strongly support publication of the revised Ms in Nat Cell Biol.

Reviewer #2:

Remarks to the Author:

Addressed concerns

Reviewer #3:

Remarks to the Author:

all concerns have been thoroughly addressed

Author Rebuttal, first revision:

We thank the reviewers for their support.

Final Decision Letter:

Dear Dr Teleman,

I am pleased to inform you that your manuscript, "Brain-enriched RagB isoforms regulate the dynamics of mTORC1 activity through GATOR1 inhibition", has now been accepted for publication in Nature Cell Biology. Congratulations on this very nice study!

Thank you for sending us the final manuscript files to be processed for print and online production,

43and for returning the manuscript checklists and other forms. Your manuscript will now be passed to our production team who will be in contact with you if there are any questions with the production quality of supplied figures and text.

Please note that *Nature Cell Biology* is a Transformative Journal (TJ). Authors may publish their research with us through the traditional subscription access route or make their paper immediately open access through payment of an article-processing charge (APC). Authors will not be required to make a final decision about access to their article until it has been accepted. Find out more about Transformative Journals

If you have not already done so, we strongly recommend that you upload the step-by-step protocols used in this manuscript to the Protocol Exchange (www.nature.com/protocolexchange), an open online resource established by Nature Protocols that allows researchers to share their detailed experimental know-how. All uploaded protocols are made freely available, assigned DOIs for ease of citation and are fully searchable through nature.com. Protocols and Nature Portfolio journal papers in which they are used can be linked to one another, and this link is clearly and prominently visible in the online versions of both papers. Authors who performed the specific experiments can act as primary authors for the Protocol as they will be best placed to share the methodology details, but the Corresponding Author of the present research paper should be included as one of the authors. By uploading your Protocols to Protocol Exchange, you are enabling researchers to more readily reproduce or adapt the methodology you use, as well as increasing the visibility of your protocols and papers. You can also establish a dedicated page to collect your lab Protocols. Further information can be found at www.nature.com/protocolexchange/about

With kind regards,

Melina

Melina Casadio, PhD
Senior Editor, Nature Cell Biology
ORCID ID: <https://orcid.org/0000-0003-2389-2243>

** Visit the Springer Nature Editorial and Publishing website at www.springernature.com/editorial-and-publishing-jobs for more information about our career opportunities. If you have any questions please click here.**